

# Measuring varve thickness using µCT: a comparison with thin section

Marie-Eugénie Meusseunan Pascale Jamba[1,2*], Pierre Francus[1,2], Antoine Gagnon-Poiré[1,2], Guillaume St-Onge[2,3]

[1] Centre Eau Terre Environnement, Institut National de la Recherche Scientifique, Québec, Canada
[2] GEOTOP Montréal, Québec, H2X 3Y7, Canada
[3] Institut des sciences de la mer (ISMER), Université du Québec à Rimouski (UQAR) and Canada Research Chair in Marine Geology, Rimouski, QC G5L 3A1, Canada

*Correspondence to* : Marie-Eugénie Meusseunan Pascale Jamba[1] (Marie-Eugenie.Jamba@inrs.ca)

**Abstract.** X-ray micro-computed tomography (µCT) scans were performed on four varved sediment cores collected in Grand Lake (Labrador) and previously studied with thin sections. These scans allowed to investigate the possibility of using µCT as

a substitute for thin sections to carry out counts and thickness measurements of varved sediments. Comparing varve counts of these two methods, µCT counts are slightly higher than the ones made with thin sections. The difference in counts suggests that the petrographic study and a SEM analysis of a thin section remain necessary for determining the varve character of the laminae. Yet, µCT allows measurements in multiple directions, improving the robustness of the counts and allowing avoiding the manufacturing of continuous thin sections along sediment sequence.

As to the thickness measurement, the µCT analyses were made in two perpendicular directions. Not surprisingly, measurements made on the same cutting plane as the thin section are quite similar to the ones made on the latter. However, there are significant differences with measurements made on the perpendicular plane. This highlights the need to perform varve thickness measurements in at least two perpendicular directions for better estimates of varved sediment thicknesses. In addition, the study illustrates that µCT is an effective way to select the least deformed zones with parallel varves to carry out

the best possible thickness measurements.

## 1 Introduction

Analyses of sediment cores extracted from lakes and oceans have allowed a better understanding of climate change and the mode of natural climate variability in various regions of the globe through robust paleoclimate reconstructions. These reconstructions required detailed studies of the structure and texture of sedimentary facies. Among the most powerful

sedimentary facies for paleoclimatic reconstructions are varves, or annual laminations. The term varve is used to define a group of laminations that are deposited for one year (Kemp, 01 June 1996; Hughen, 2013; De Geer, 1912). These cyclic annual



sedimentation facies form in marine or lacustrine environments and can have a detrital, endogenous, or biogenic origin (Zolitschka et al., 2015; Schimmelmann et al., 2016).

The detrital varves are the object of study of this project. They often show regular alternation of light and dark beds of different

grain sizes, with millimetric thicknesses which are attributed to seasonal variations in detrital sediment supply (Ojala et al., 2012; Zolitschka et al., 2015).

The advantage of studying varves is that they are high-resolution sedimentary sequences that provide information on past abrupt environmental changes through variations in the structure, composition, and thickness of their distinct seasonal laminae (Ojala et al., 2012). Indeed, these variations of structures and the thickness of their seasonal laminae can be the result of several

independent factors such as melting of snow, landslide, etc. Yet, with a good understanding of the sedimentary components of varves and the mechanisms controlling their formation, they can be good paleoclimate indicators (Gagnon-Poiré et al., 2021; Amann et al., 2017; Palmer et al., 2019; Hardy et al., 1996). In addition, varved sediments contain their own chronology which can be converted into calendar years with exceptionally high temporal resolution. Ultimately, they are offering the possibility of calculating sediment flux rates (Ojala et al., 2012; Zolitschka et al., 2015; Emmanouilidis et al., 2020).

Thin sections are the most commonly used method to analyze varved sequences. The sediment core is sampled continuously by subsampling overlapping sediment blocks, which will then be freeze-dried and impregnated with epoxy resin, cut into blocks allowing the manufacturing of thin sections (Normandeau et al., 2019; Francus and Asikainen, 2001). Counting and measuring varves is then done by image analysis of thin sections and/or using SEM image in backscattered mode to improve the ability to define varve boundaries(Francus et al., 2008; Lapointe et al., 2019; Gagnon-Poiré et al., 2021; Soreghan and

Francus, 2004).

However, the sampling method as well as the multiple steps needed for the preparation of thin sections can induce sediment perturbations (refs). Finally starts to question of the representativeness of the sample analyzed : it has the limitations associated with its 2D view (Bendle et al., 2015), which make it difficult to estimate the true thickness of the varves. Yet, X-ray micro-computed tomography (µCT) is a tool that allows the observation of objects with a resolution of a few micrometers and has

the advantage of being a non-destructive technique. It allows the view of a volume instead of a plane, facilitating the study of the internal structures and orientation of a wide variety of geological objects (Cnudde and Boone, 2013; Voltolini et al., 2011; Bendle et al., 2015; Lisson et al., 2023; Cornard et al., 2023; Fabbri et al., 2024) .

This article aims to investigate the possibility of using X-ray micro-computed tomography (µCT) as an analytical tool to perform thickness measurements of varves in the frame of paleoclimatic studies. To do this, we tested the µCT method on a

sequence whose varves are easy to recognize, thick, not very disturbed and which have already been studied using the thin sections method. The objectives of this paper are to: (1) test if varve counts can be performed using µCT, (2) test if varve thickness can be retrieved from µCT images, (3) compare the thickness measurements obtained with those retrieved on thin section method, (4) evaluate the added value of µCT as an analysis tool for varved sediments.



## 2 Methods

### 2.1 Sample selection

The analyzed sediments come from Grand Lake, a 60 km long elongated lake, with a depth reaching 245 m (Trottier et al., 2020; Gagnon-Poiré et al., 2021). This fjord lake is located in a valley connected to the Lake Melville graben in the central Labrador region at 53°41'25.38" N, 60°32'6.53" W, approximately 15 m above sea level (Fig. 1) (Trottier et al., 2020; Gagnon-Poiré et al., 2021).

Grand Lake has two principal tributaries, the Naskaupi and Beaver Rivers. Four undisturbed sediment sequences (Fig. 2) were extracted in front of the deltas using a Uwitec gravity corer and are listed in Table 1. After recovery, these cores were cut into two halves. The first one was used for making thin sections, and their subsequent analysis with a scanning electron microscope was reported in Gagnon-Poiré et al. (2019). The second halves were analysed in this paper using a µCT scanner.

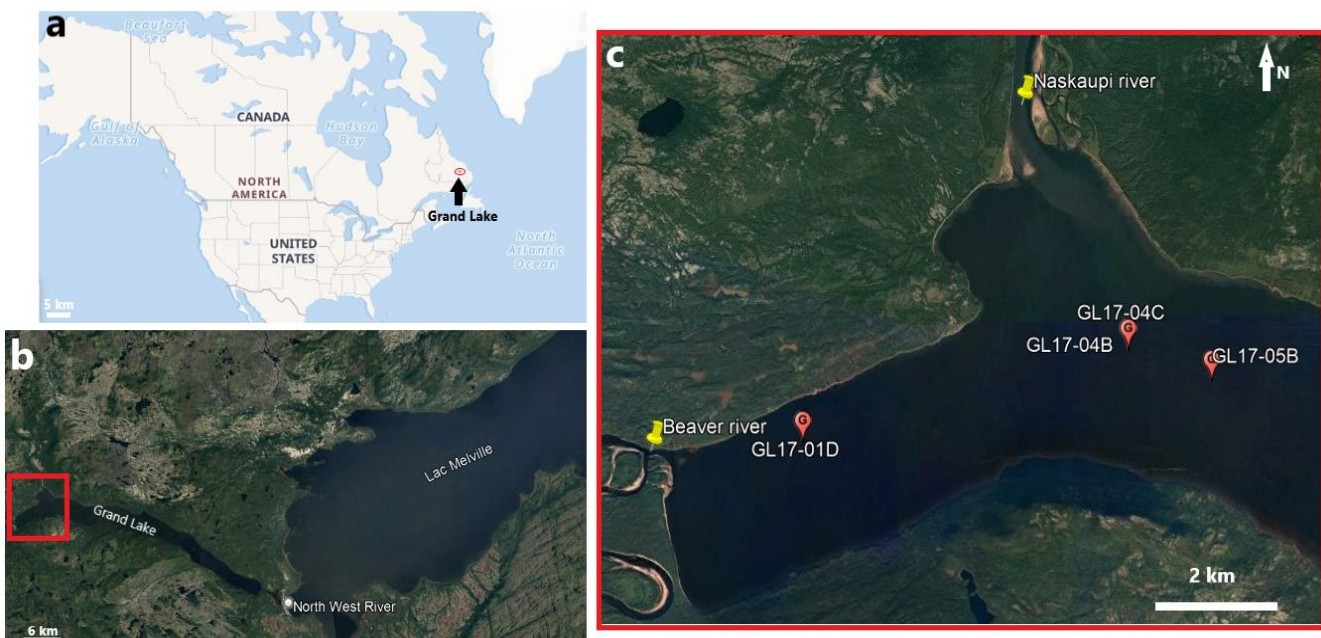

**Figure 1: Study area. (a) Location of Grand Lake encircled in red on the map of Canada. (b) Location of the study area. (c) Study area with the location of the principal tributaries of Grand Lake (yellow pins) and the sediment cores GL17-01D, GL17-04B, GL17-04C, GL17-05B (© Google Earth 2023).**





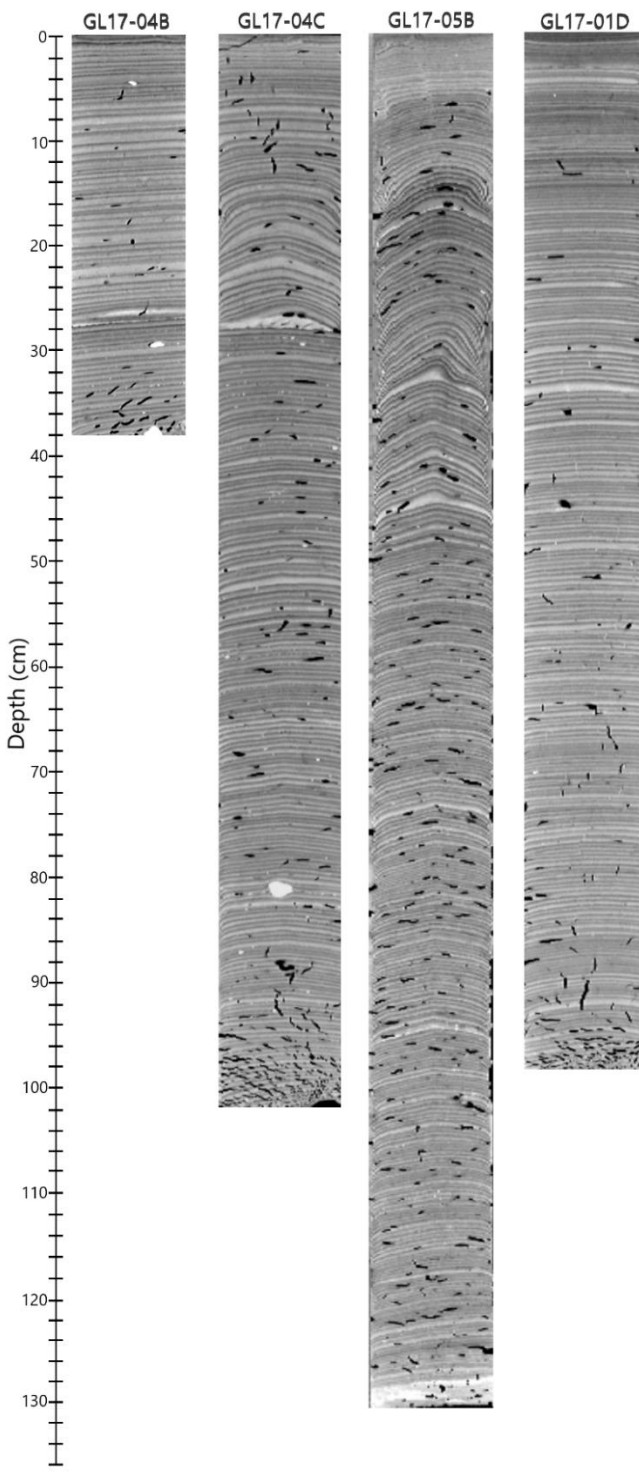

**Figure 2: CT scan images of the four sediment cores GL17-04B, GL17-04C, GL17-01D and GL17-05B used in this study.**



## 2.2 Measurement of varve thickness on thin sections

The making of thin sections was done in several steps. Firstly, the half sediment cores were subsampled using aluminum boxes, which were positioned with an overlap of approximately 1 cm to retrieve a continuous sequence along the sediment core (Francus & Asikainen, 2001).

The sediment was then frozen with liquid nitrogen to avoid the formation of large hexagonal ice crystals which would deform the sediment. Next, the sediment was freeze-dried for 48 hours to eliminate the interstitial water from the pores without modifying the sediment structure (Normandeau et al., 2019). This step can also be problematic because freezing too quickly can cause cracks in the sediment block (Normandeau et al., 2019).

The sediment was subsequently impregnated using a low viscosity Spurr epoxy resin and hardened by thermal curing for 48

90 hours. The final step consisted of cutting the impregnated block of sediments to be used to manufacture thin sections (Lapointe et al., 2019).

These thin sections were first digitized and analyzed under a petrographic microscope, then images in backscattered mode were acquired using the SEM (Zeiss Evo 50) at a voltage of 20 kV (Francus et al., 2008; Gagnon-Poiré et al., 2021; Lapointe et al., 2019) with a resolution of 1 µm/pixel (Francus et al., 2002; Gagnon-Poiré et al., 2021). These 8-bits high resolution

images thus made it possible to count varves, providing good contrast for features which very often measure less than 0.5 mm and are difficult to measure otherwise (Francus, 1998; Lapointe et al., 2019). Varve thicknesses are manually acquired with a custom-made image analysis software (Francus and Nobert, 2007): varve boundaries are manually marked along a vertical line chosen by the user and the software records varve thicknesses using vertical coordinates of each boundaries. If sediment disturbances prevent reliable counts, additional vertical lines can be used for thickness measurements, but the software does

not correct for the dip of the layers.

## 2.3 Experimental setting: µCT acquisition and reconstruction

The scans were made using a TESCAN CoreTOM µCT at INRS-ETE in Québec city with the Aquila software (version 1.2.1) (Dewanckele et al., 2020). The halfcores swere scanned in a vertical position with custom-designed holder in acrylic, maintaining the half sediment core in position during the scans, using a half foam core of the same size.Several instrumental

acquisition parameters were tested to identify the best possible configuration to obtain high-resolution images with easily detectable varve borders.

The best settings were a tube voltage of 140 kV, with an exposure time of 150 ms, with an SDD (Source Detector Distance) of 500 mm and an SOD (Source Object Distance) of 150 mm, resulting in a voxel size of 45 µm. A 1.5 mm thick aluminum filter was used to reduce the impact of beam hardening on image quality (Rana et al., 2015; Ay et al., 2013). Smaller subsets

of the scans, or Volumes of Interest (VOIs), were defined along the sediment cores where varves were clearly distinct. Individual VOIs (c.a. 30 mm wide, 450 mm thick, 40 - 60 mm long) (Fig. 3) were overlapping each other over 10-20 mm to



ensure the continuity of the sedimentary sequence, and to deal with the cone beam artefacts (see below). Image processing and analysis of these individual VOIs were also made easier for a regular laptop computer.

The scan time depended on the length of the sediment core and varied between 2h30 and 4h30. Subsequently, these scanned
areas were reconstructed with the Panthera software (version 1.5.0.21), with an average reconstruction time of 23 minutes per reconstructed VOI (Table 1).

**Table 1: Summary of the four sediment cores with the µCT acquisition and reconstruction time.**

| Sediment cores | Correspond to (Gagnon-Poiré et al., 2021) | Location | | Length (cm) | µCT Acquisition time | µCT reconstruction time | µCT total file size | VOIs number | Thin section total file size |
|---|---|---|---|---|---|---|---|---|---|
| | | Latitude | Longitude | | | | | | |
| GL17-04B | NAS-1A | 53,749279 | -60,820009 | 38 | 2h30 | 3h | 45 GB | 5 | 526 MB |
| GL17-01D | BEA-1 | 53,737257 | -60,904032 | 98 | 3h30 | 5h | 164 GB | 13 | 975 MB |
| GL17-04C | NAS-1B | 53,749279 | -60,820009 | 102 | 3h30 | 5h30 | 122 GB | 12 | 582 MB |
| GL17-05B | NAS-2 | 53,74438 | -60,798786 | 131 | 4h30 | 6h30 | 135 GB | 10 | 629 MB |

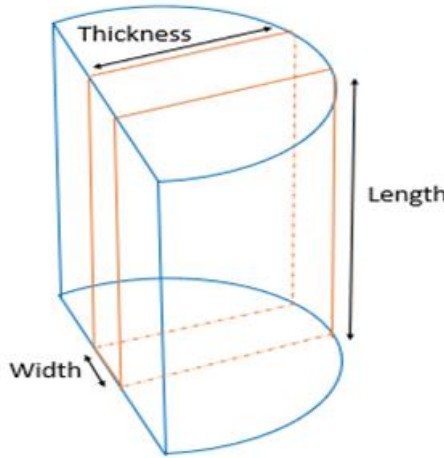

**Figure 3: Orange box showing the shape of Volumes of Interest (VOIs) in the half sediment cores (in blue).**

The scans were performed using the STAMINA mode (Fig. 3), i.e., projections are acquired over a certain high of the sample keeping the source and the detector steading while the sample is rotating over 360 ˚. Since the beam is conical in STAMINA scans, the parts of the VOI outside the central plane of the beam are seen at a certain angle, which causes blurred parts at the limit zones of the VOI when horizontal features are imaged (Fig. 4) (Laeveren D. , 2020; Sheppard et al., 2014). Yet, the
reconstructed VOIs presented blurred zones at their borders due to the beam cone artefacts which prevents to clearly distinguish the limits of the laminae (Laeveren D. , 2020; Sheppard et al., 2014). However, having overlapping VOIs allowed to easily distinguish laminas which were not visible on a single VOI. This allowed to numerically reconstruct each sediment core.





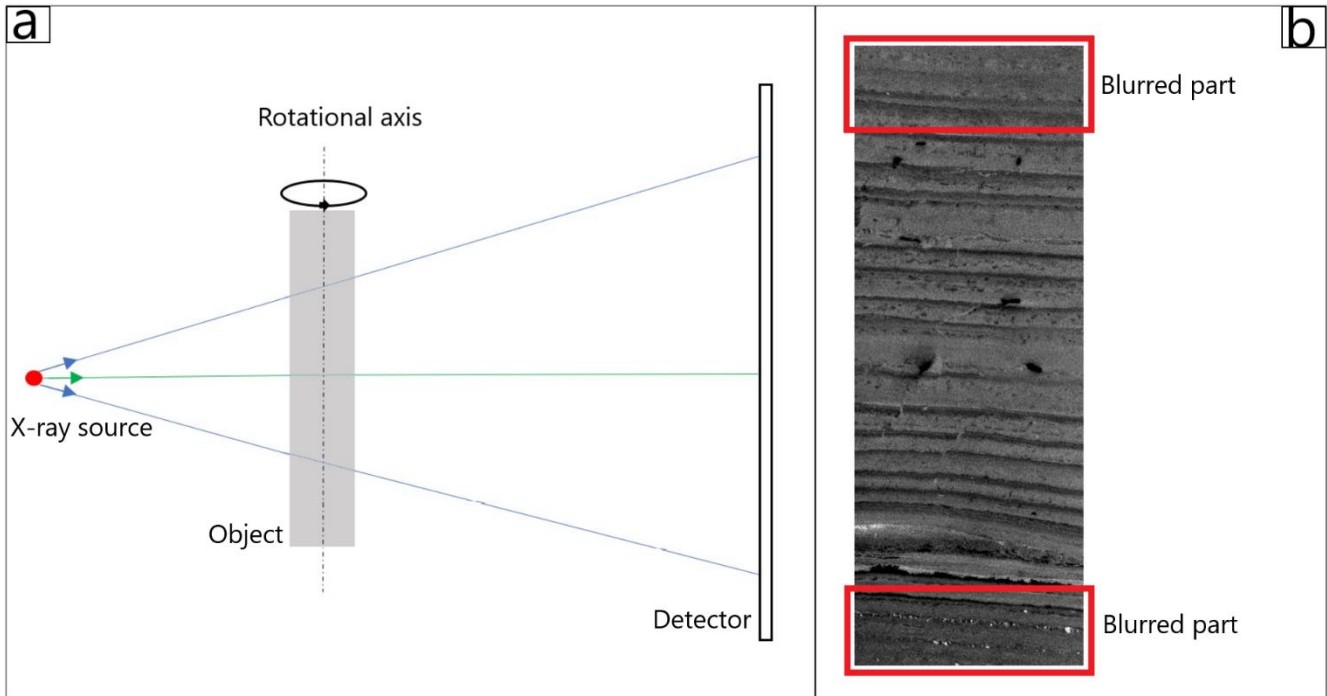

**Figure 4: The formation of the beam cone artefact. (a) Presentation in STAMINA mode of the image acquisition mode. The conical**

**X-rays from the source are absorbed by the object and some of them are transmitted to the detector. The green x-ray follows the**

**shortest distance to reach the detector while the blue x-rays reach the detector at an angle which causes a mix of information in the**

**vertical direction. (b) VOI reconstructed with the presence of beam cone artifact at its boundary parts.**

## 2.4 Data analysis

The sediment core images were analyzed with the Dragonfly software (version 2022.1). Varve counting and their thickness

measurements were made in two (2) different and perpendicular directions, one of which corresponds to the thin section cutting

plane. In each direction three (3) different counts were made on three (3) different cross sections (Fig. 5a) to ensure to have

three (3) counts of varve per direction and to define the average thicknesses of the laminae in each of the directions (Fig. 5a).

After this, the μCT counts were repeated two (2) times in each of the two directions to check the repeatability of the

measurement method. The thicknesses were all measured manually by measuring the length of the shortest line segment which

separates two parallel varve borders, and which were perpendicular to them (Fig. 5b). The next step was to create a profile of

variation of thicknesses as a function of depth. Each varve was identified by the depth of its top and bottom boundary, and its

thickness was calculated by the depth difference (Fig. 6). Thickness profiles were assembled by the addition of these values.



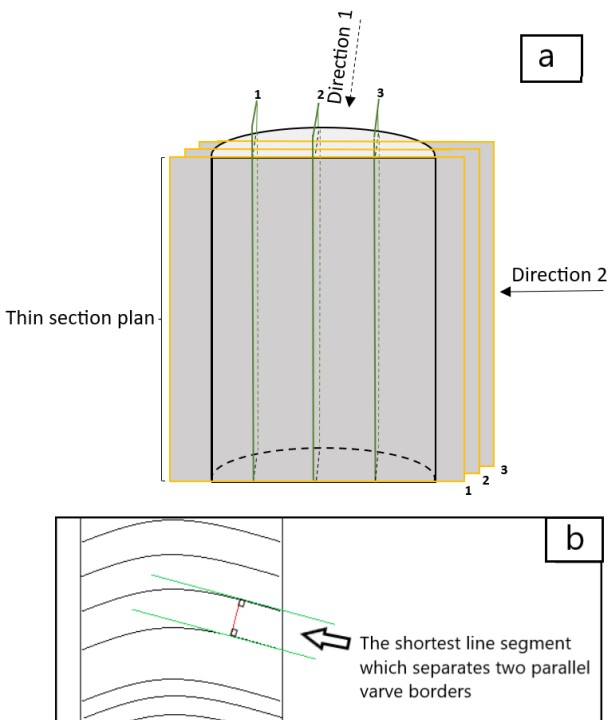

**Figure 5: (a) The Fig. shows a half sediment core delimited by black outlines and the directions in which the thicknesses are measured. Measurements with μCT are made in two different directions, with three (3) different measurements on each direction. The green outlines show the three cross sections selected in the 1st direction to perform the counting and the thickness measurements of varves, and orange outlines show the three cross sections selected in the 2nd direction. The plane of the thin sections corresponds to the first plane of measurements in the direction 2. (b) This Fig. shows how the thickness of the varve is measured. The measurement is made by measuring the length of the shortest line segment that separates two parallel varve borders.**

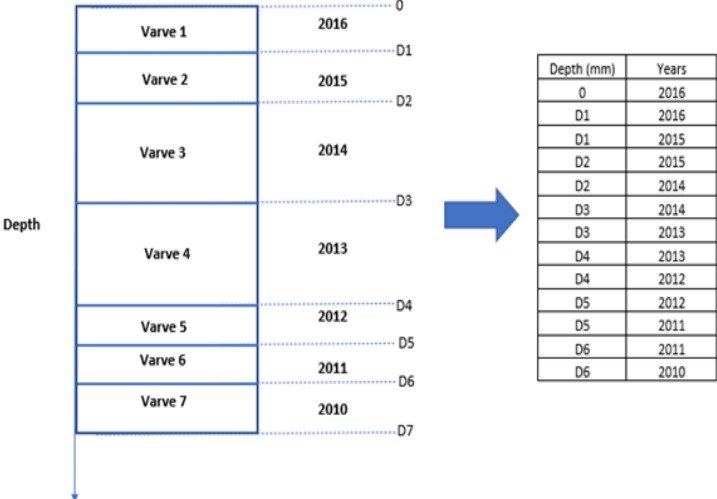

**Figure 6: Depth profile calculation model considering that each varve is characterized by two distinct depths.**



## 2.5 Comparing the different thickness measurements

The results obtained were compared with those of Gagnon-Poiré et al (2019). Thickness comparisons are made by locating the stratigraphic marker beds to ensure the same varves are compared. The average of the thicknesses of each of the three counts

155  per direction obtained with the µCT is compared to that of the three counts obtained with the thin sections. This comparison was made using linear regression , the percentage of average variation of thicknesses compared to the average value and the agreement study to evaluate the relationships that exist between the results obtained by µCT and those of thin sections. The percentage of average variation of thicknesses compared to the average is obtain with the following formula:

$$Percentage\ of\ average\ variation\ of\ thicknesses = \frac{mean\ standard\ deviation}{mean\ thickness}\ x\ 100\ ,$$

The measurements with the µCT were all made along the four sediment cores. However, comparisons with thin sections were only made at locations where thin sections were sampled and previously analysed. Comparisons with thin section were therefore made throughout cores GL17-04B and GL17-01D and on portions of sediment cores of GL17-04C and GL17-05B. The study of the differences between the thickness measurements on thin sections and those obtained from µCT image was made using the agreement method of Bland and Altman (Altman and Bland, 1983; Grenier et al., 2000; Ranganathan P, 2017).

The agreement is a notion which refers to the fact that two or more independent measurements of the same quantity are equal (Ranganathan P, 2017). This method compares the difference observed between the values obtained for the same measurement by two different methods. To do so, it calculates the bias and the limits of the confidence interval at 95% for each measurement and defines the agreement between two series. The bias (mean of differences) and the limits of agreement represent the deviations of the values of one method from another. The difference between the two measurements methods is expressed as

a function of the mean values obtained with each of the two methods (Fig. 7) (Altman, 1983; Grenier, 2000). For this paper, we consider $A_i$ the thickness measured by method 1 for the varve $i$ and $B_i$ the thickness measured by method 2 for the same varve.

$$Difference\ between\ the\ two\ measurement\ methods\ for\ the\ varve\ i = A_i - B_i\ ,$$

$$Mean\ of\ the\ two\ measurement\ methods\ for\ the\ varve\ i = \frac{A_i + B_i}{2}$$

$$Bias\ (or\ mean\ of\ differences) = \frac{\sum_{i=1}^{n}(A_i - B_i)}{n}\ ,\ where\ n\ is\ the\ number\ of\ varves.$$

$$Confidence\ interval\ at\ 95\% = [Bias - 1.96\ x\ STD; Bias + 1.96\ x\ STD]\ ,$$ where STD is the standard deviation of the differences.

The agreement will be considered as good between the two methods when two conditions are fulfilled: there are few points outside the confidence interval, and when the points outside the confidence interval are close to the limits of this interval. In

this case study, we will define the percentage of agreement that is for each sediment core the percentage of points outside the confidence interval.

$$Percentage\ of\ agreement = \frac{number\ of\ points\ outside\ the\ confidence\ interval}{number\ total\ of\ points}\ x100,$$





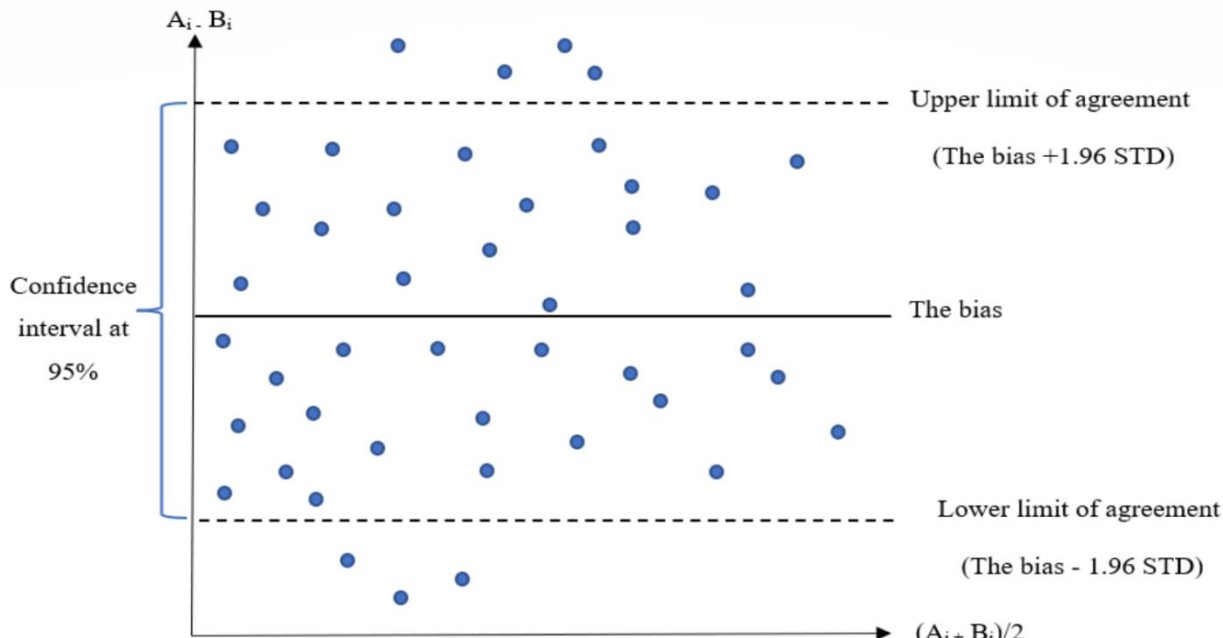

**Figure 7: This image shows the interval in which 95% of the differences between two methods are included. Each blue point represents the result of the difference Ai- Bi between two measurements. The Upper and Lower limits of agreement are the limits of the confidence intervalley at 95%. The difference Ai- Bi between the two measurement methods is expressed as a function of the mean values (Ai-Bi)/2 of results of the two methods.**

**3 Results**

**3.1 Varve counting**

Overall, the number of varves counted with the μCT method is very similar to that one made on thin sections, except that of the sediment core GL1704C where the number of varves is little higher than that of thin section (Table 2).

**Table 2:Summary of varve counting.**

| Sediment  cores | μCT counts | Thin section counts |
|---|---|---|
| GL1704B | 68-69 | 69 |
| GL1701D | 245-246 | 240-247 |
| GL1704C | 206-209 | 198 |
| GL1705B | 303-306 | 303 |



## 3.2 Thickness measurements

The mean varve thicknesses measured using the µCT and the thin sections in direction 1 and 2 and thin section are showed in
Fig. 8. The thickness differences are greater in direction 1 (one) with GL17-04B (Fig. 8a), GL17-04C (Fig. 8c) and GL17-05B
(Fig. 8d and 8e) than in direction 2 (two). As for GL17-01D (Fig. 8b), the measurement differences are less important and
quite similar in both directions. Thickness measurements made on thin sections (Fig. 8a-e) are overall less variable than those
made on µCT images.

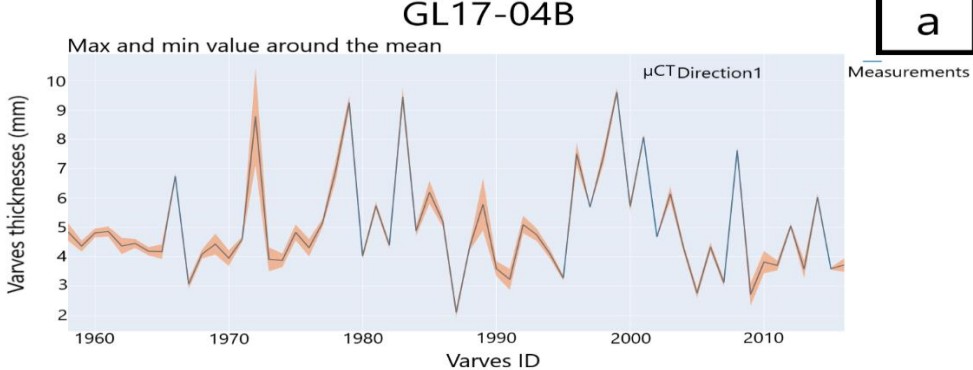

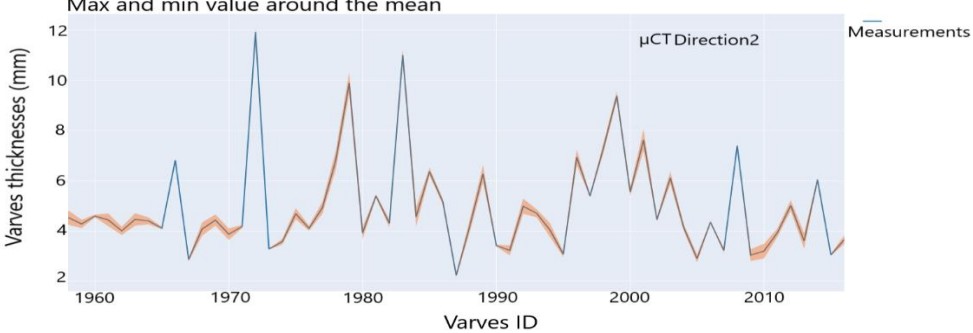

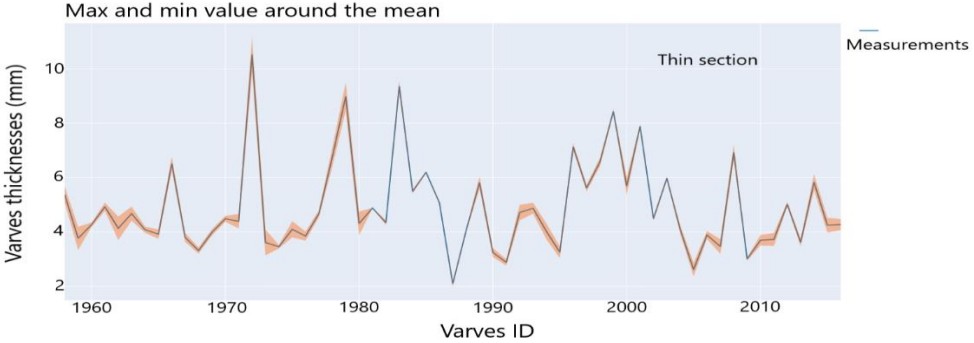



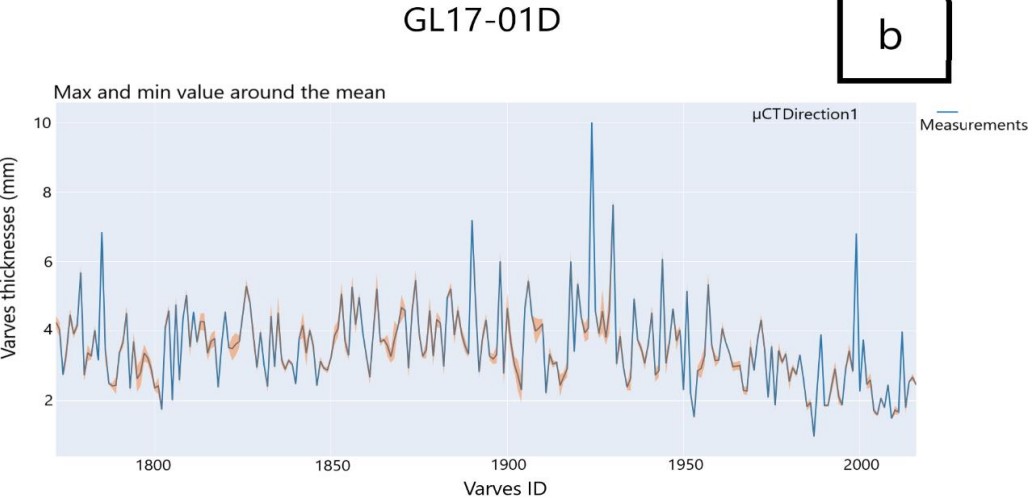

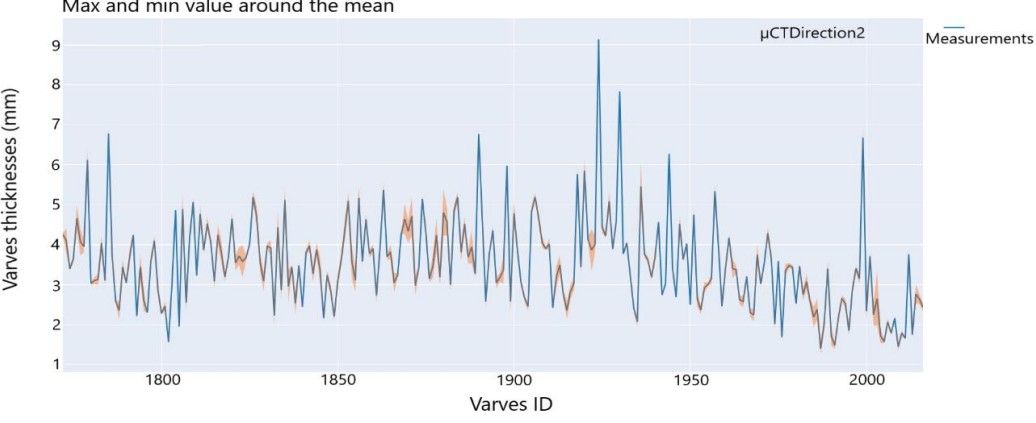

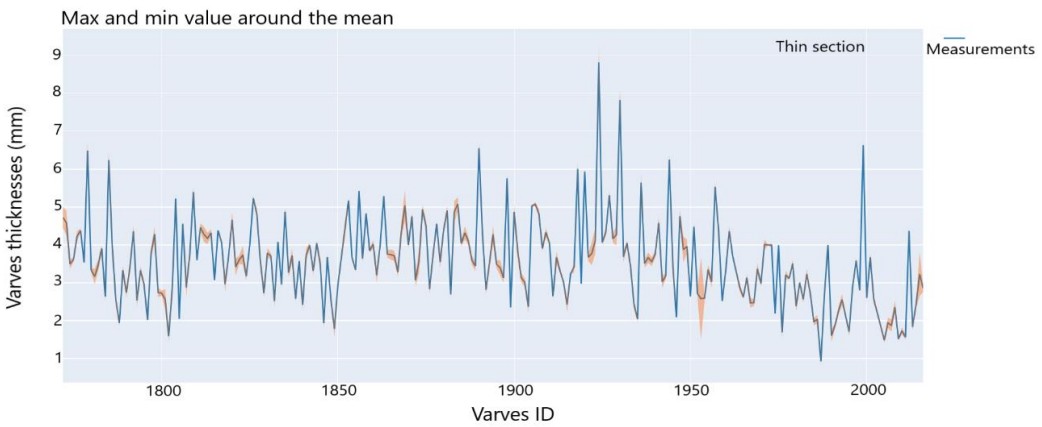



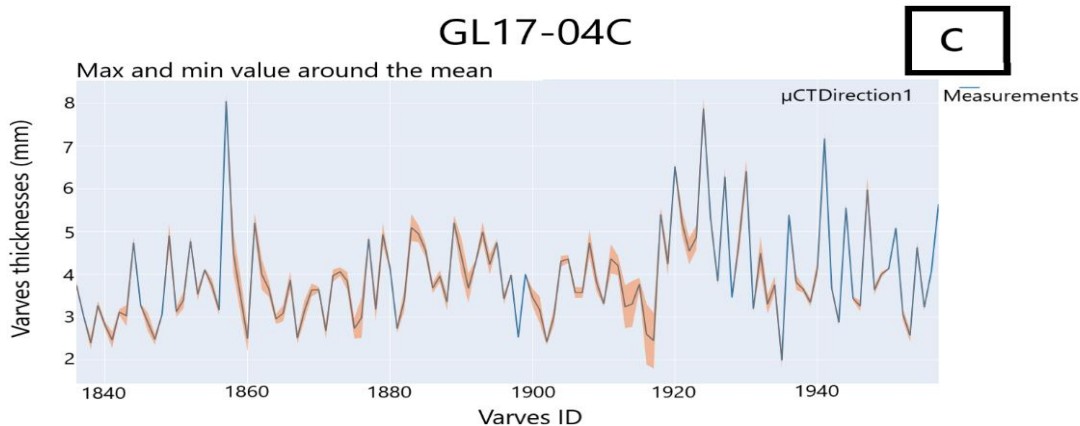

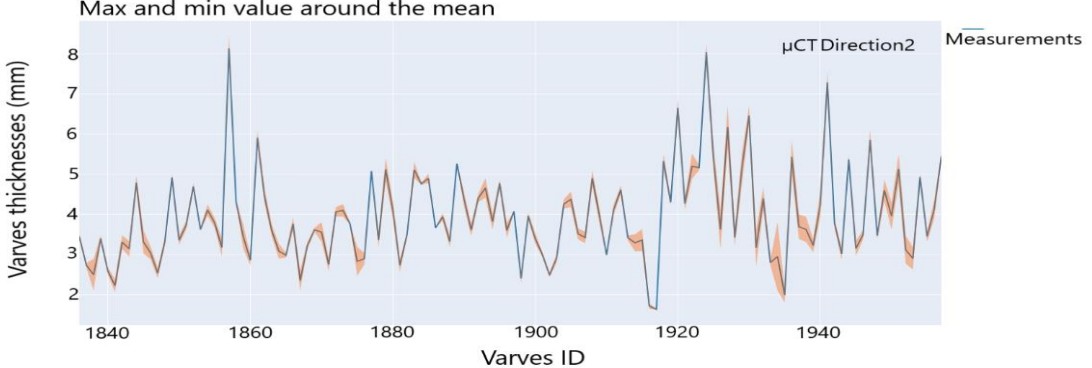

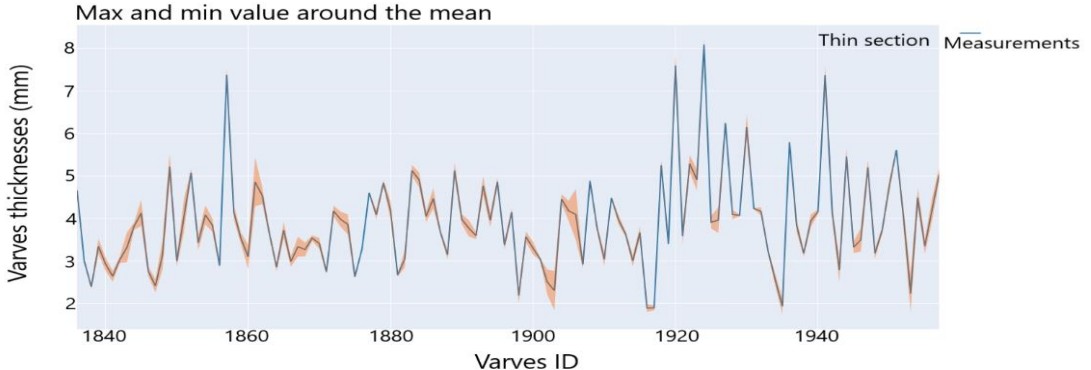



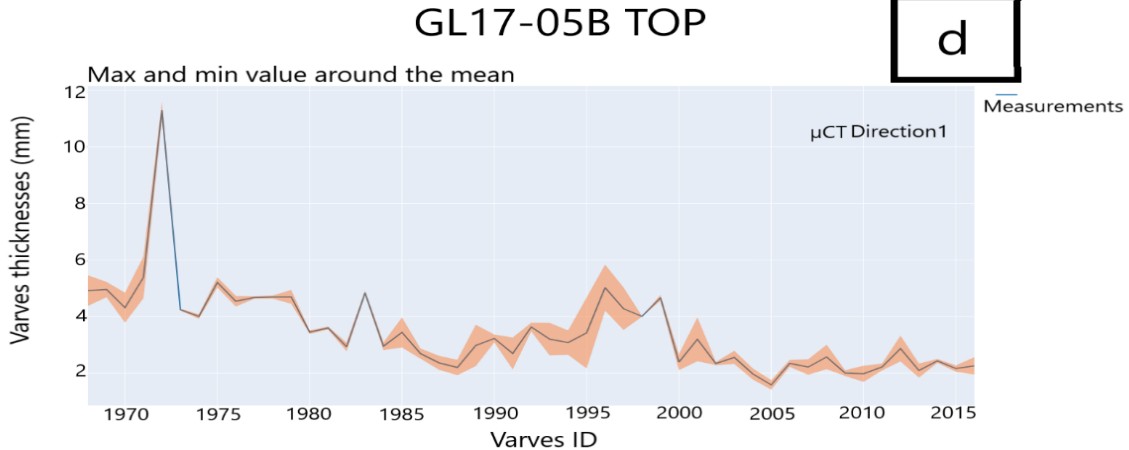

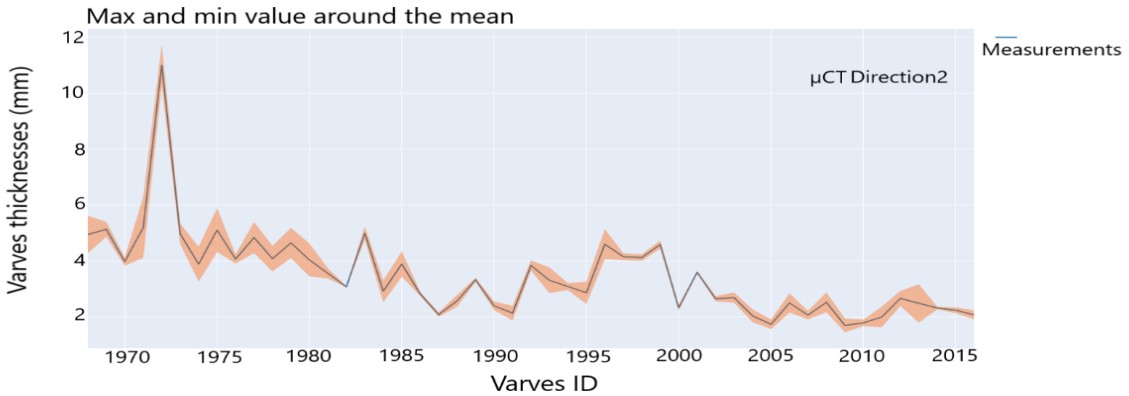

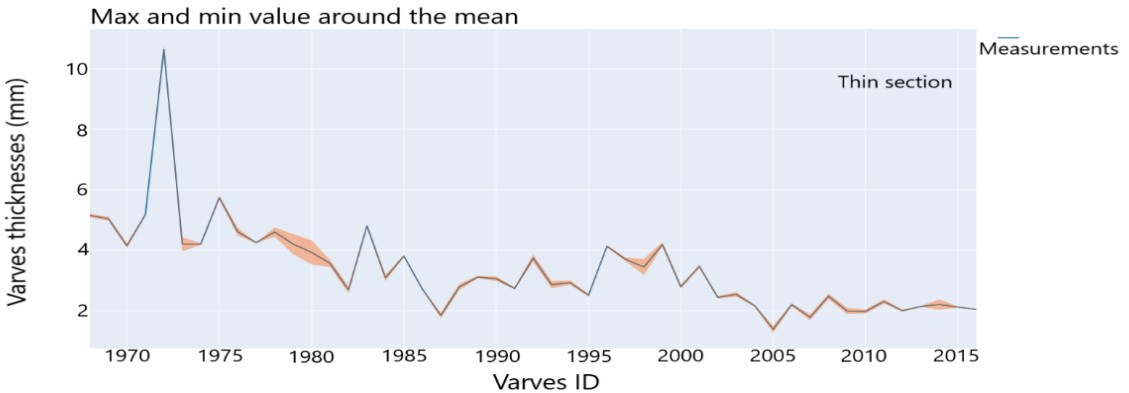



**Figure 8: Mean varve thicknesses of the three measurements made in directions 1 (top panels) and 2 (middle panels) using the µCT and on thin sections (lower panels) for cores GL17-04B (a), GL17-01D (b), GL17-04C (c), GL17-05B (d-e). The max and the min value around the mean relating to the measurements are the orange shaded area and the varves ID represents the identification of**

**the varve over an interval.**





### 3.3 Repeatability the µCT thicknesses measurements

To evaluate the robustness of the thickness measurements made using the µCT, the linear regressions and their determination coefficients $R^2$ have been calculated for the measures made on the same direction. They are all greater than 0.95 with maximum

coefficients of 0.9915 and 0.9972 for the GL17-04B sediment core in directions 1 and 2 respectively. Core GL17-04C shows the next best correlations and GL17-01D with GL17-05B are those with the weakest, but still very high values (Fig. 9).





**Figure 9: comparison of the mean varve thickness measurements results with µCT for the same sediment cores analyzed: GL17-04B, GL17-01D, GL17-04C and GL17-05B (all measurements are in appendices).**





## 225    3.4 Comparison of measurements between µCT and thin sections

The mean thickness measurements with µCT in the two perpendicular directions are compared by linear regression with those obtained by the thin section method (Fig. 10). There is a good correlation between µCT, and thin sections measurements as shown in Fig. 10, with minimum correlation coefficients varying from 0.7688 to 0.8287, and maximum values between 0.9461 and 0.95. In general, the correlations are better with the direction two (2).





**Figure 10: linear regression between mean thickness measurements of µCT and thin sections respectively on the sediment cores GL17-04B, GL17-01D, GL17-04C and GL17-05B.**



## 3.5 Percentage of average variation of thicknesses

Overall, the average percentage variation of thicknesses compared to the average value is the lowest with thin sections except

with GL17-04B. Direction 1 is the one with the highest percentage of thickness variation, followed by the second direction (Table 3).

**Table 3: Percentage of average variation of thicknesses compared to the average value.**

| Sediments cores | GL17-04B | GL17-01D | GL17-04C | GL170-5B TOP | GL170-5B BOTTOM |
|---|---|---|---|---|---|
| Direction 1 | 4.89 % | 4.82 % | 4.9 % | 9.28 % | 7.82 % |
| Direction 2 | 3.81 % | 4.56 % | 4.69 % | 8.76 % | 5.41 % |
| Thin section | 4.32 % | 3.79 % | 4.29 % | 2.87 % | 3.59 % |

## 3.6 Agreement study between the two methods

The output of the Bland and Altman agreement method is presented in Fig. 11 between thickness measurements obtained using the µCT direction #2 and the thin sections. The results in direction #1 are presented in appendix.

The thickness differences observed with a 95% confidence interval are the following: for core GL140B between -0.9615 and 0.7292 mm, with a mean difference (bias) of -0.12 mm; for GL17-01D, between -0.4760 and 0.5598 mm with a mean difference of 0.04 mmGL17-01D; for GL17-04C between -0.9483 and 0.8540 mm with a mean difference of -0.05 mm; and

for GL17-05B, between –0.9347and 0.7721 mm with a mean difference of -0.08 mm (Fig. 11). Overall, with a 95% confidence interval the maximum mean difference obtained is 0.12 mm. The percentages of points outside the interval of confidence for GL17-04B, GL17-01D, GL17-04C and GL17-05B are respectively 3.33 %, 3.27%, 5.83% and 8.05 %. Then the percentage of agreement is respectively for GL17-04B, GL17-01D, GL17-04C and GL17-05B: 96.67%, 96.73%, 94.17% and 91.95%.

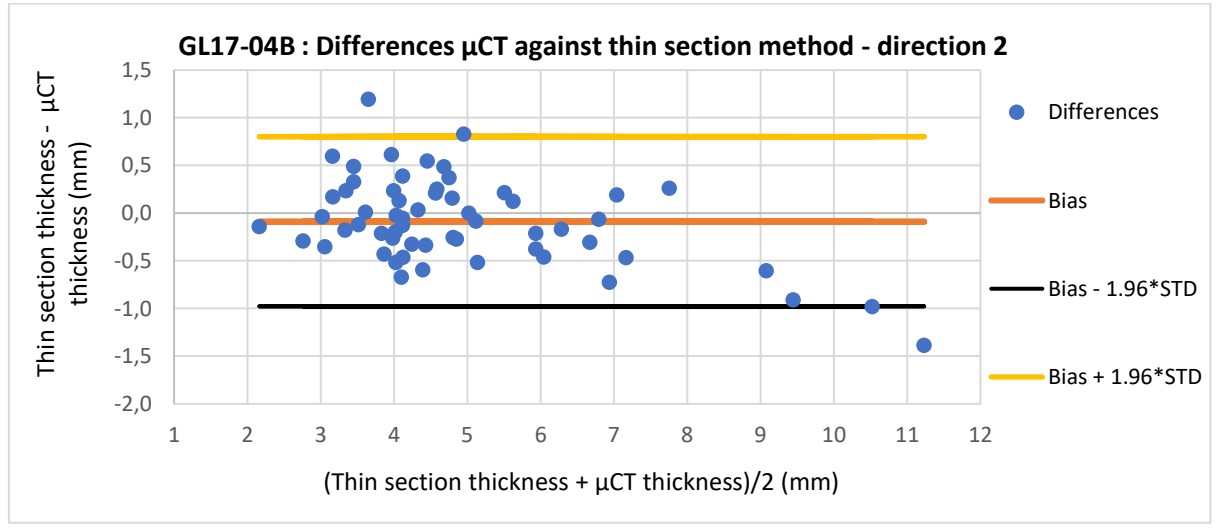




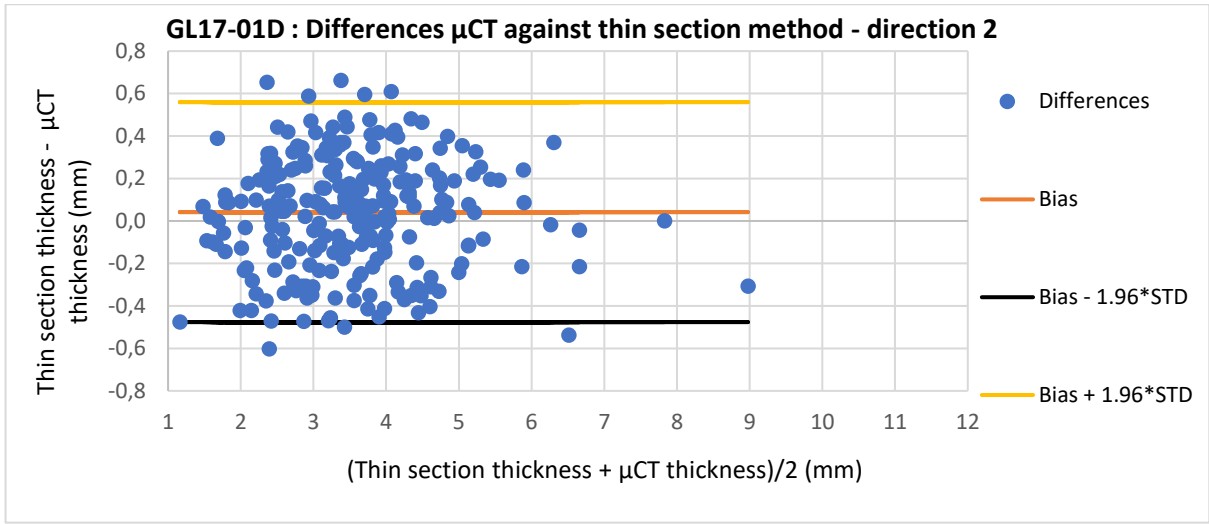

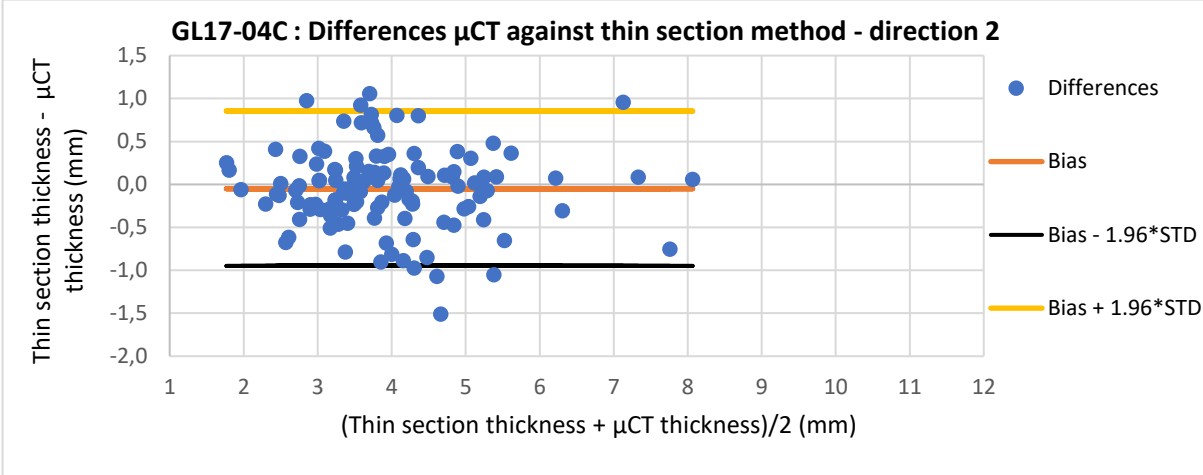

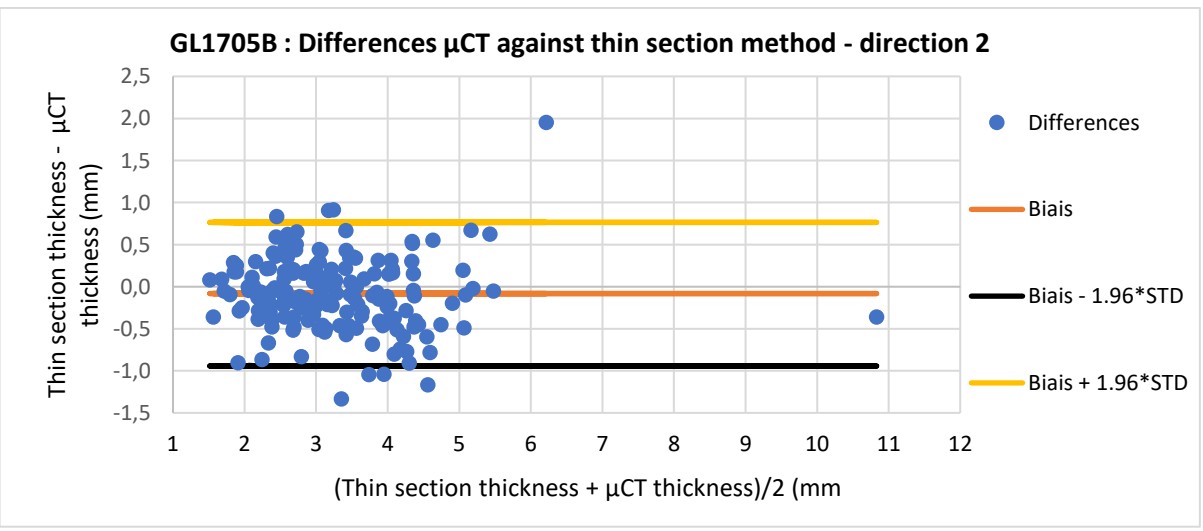





**Figure 11: Agreement study in the direction #2 between the two methods, respectively on the sediment cores GL17-04B, GL17-01D, GL17-04C and GL17-05B. The orange line represents the bias, and the two yellow and gray lines are the limits of the 95% confidence interval. Note that the vertical axes don't have similar ranges.**

The table below presents a summary of the results obtained.

**Table 4: summary table of results.**

| | | GL17-04B | GL17-01D | GL17-04C | GL17-05B |
|---|---|---|---|---|---|
| Sediment core features (figue 1) | | Varves slightly deformed | Varves slightly deformed and its upper part was broken | Varves deformed | Varves strongly deformed |
| Varve counting | | µCT counts similar to thin sections ones | µCT counts similar to thin sections ones | More varves counted with µCT scans | More varves counted with µCT scans |
| Varves thicknesses variations | µCT | Small variations in direction 2 | Similar variations in both directions | Small variations in direction 2 | Small variations in direction 2 |
| | Thin sections | Like µCT results in the direction 2 | Like µCT results in the direction 2 | Differents from µCT results | Differents from µCT results |
| $R^2$ (µCT vs µCT) | Direction 1 | 0,9915 | 0,9678 | 0,9856 | 0,9567 |
| | Direction 2 | 0,9972 | 0,9604 | 0,987 | 0,9759 |
| $R^2$ (µCT vs thin sections) | Direction 1 | 0,9288 | 0,8801 | 0,8287 | 0,7688 |
| | Direction 2 | 0,95 | 0,9461 | 0,8382 | 0,8483 |
| Percentage of agreement | | 96.67% | 96.73% | 94.17% | 91.95% |

## 3.7 Thicknesses depth profiles

Sediment cores GL17-04C, GL17-04B show almost identical depth profiles in their common parts ((Fig. 12). GL17-05B and GL17-01D cores show profiles with much lower thickness peak amplitudes. However, GL17-05B and GL17-01D cores profiles are also different. Nevertheless, stratigraphic correlations between the cores can be performed.



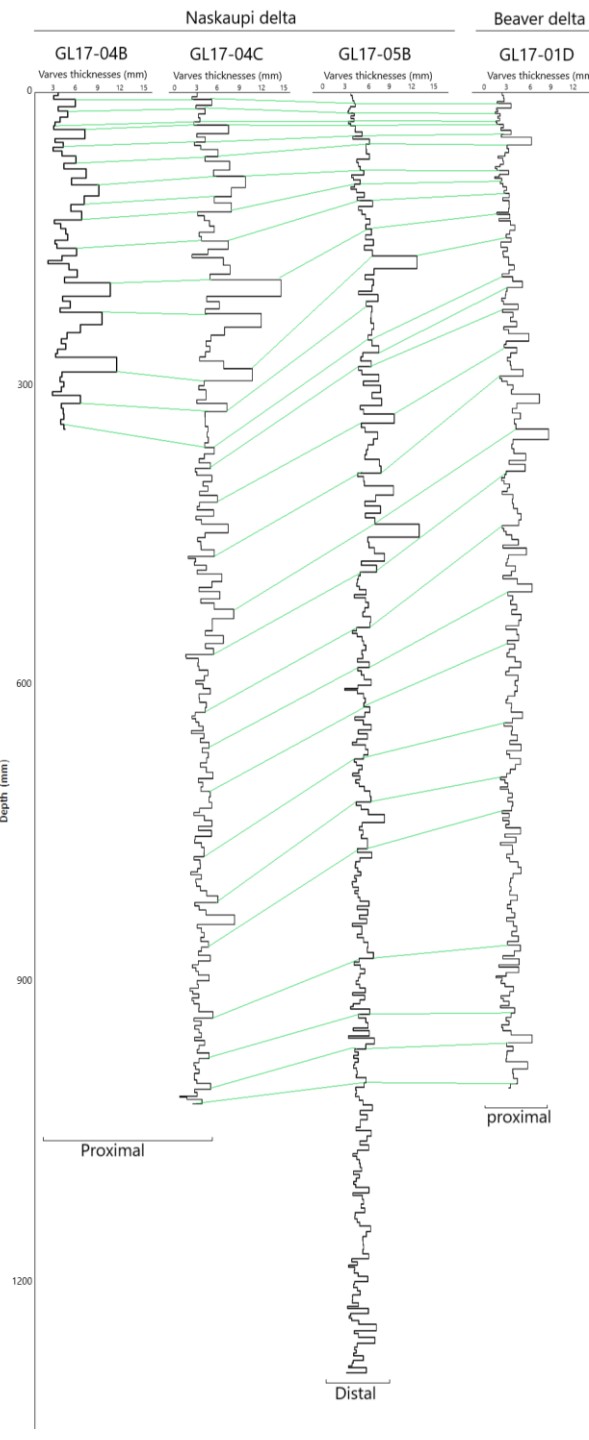

**Figure 12: Depth profile of the four (4) sediment cores. GL17-04B, GL17-04C and GL17-05B were sampled in front of the Naskaupi River delta, and GL17-04B and GL17-04C come from the same drilling site. GL17-01D was sampled from the front of the Beaver River delta. The stratigraphic correlations have been made based on images of sediment cores.**





## 4 Discussion

**4.1 Varve counting**

The difference in the number of varves observed with counting to µCT can be explained by several facts. Firstly, 3D analysis with the µCT allows to access laminations that are not visible on the thin section plane (Fig. 13). Secondly, there are thin laminations which have not been counted as varves in thin sections unlike on µCT 3D images (Fig. 14). However, in this case, the thin sections counting is better because petrographic and SEM observations allows obtaining additional information
regarding the features of a lamina (Lapointe et al., 2019; Francus, 2006), which is not the case with µCT in this study. Finally, the presence of deformations in the thin section plane of the thin section can have an impact on the counting of the varves (Fig. 15). Yet, the µCT offers the advantage of 3D vision and wider choice of a region without disturbances.

**4.2 Thickness measurements**

Direction #1 at µCT display larger thicknesses variations from their mean than direction #2. This can be explained by the fact
that this direction presents laminae that are more disturbed. Thus, in the same direction, from one cross section to another, the measured thicknesses vary. Also, GL17-04C and GL17-05B are the sediments cores with the most variations in thickness. These two cores are the most deformed of the four (Fig. 16b and 16c): a deformation of the varve can induce a simple shift of the measurement point in the same cross section, hence leading to a countable difference in the measured thicknesses. Measuring the thickness of varves at the hinge of a folded boundary is then significantly different from measuring the thickness
on a perfectly horizontal varve. With the 3D vision of µCT, it is possible to perform scans on larger areas at low resolution in order to select the least distorted areas where to perform high resolution scans (Fig. 17).

The measurements on the thin sections, for their part, present small variations with an average percentage variation of thicknesses of less than 5% (Table 3). This is explained by the fact that with high resolution images of thin sections, it is possible to see micro-variations in the thickness of the varve.

**4.3 Repeatability of the µCT method**

Multiple thickness measurements using µCT show that it is possible to repeat measurements with almost identical values when the laminations are slightly or not deformed. This is the case for the less disturbed core, GL17-04B, that has correlation coefficient closest to one (1) GL17-04B in both directions. It is much easier to reproduce varve thickness measurements almost identically on slightly disturbed sediment cores. More disturbed cores, such as GL17-01D, have somehow lower correlation
coefficients, but they are still quite close to 1. Nevertheless, this shows that even slight deformations impact the thickness measurements (Fig. 16a).





### 4.4 Comparison of measurements between µCT and thin sections

The best correlations between µCT and thin sections thickness measurements are obtained in direction #2. This is mainly because this direction corresponds to the cutting plane of the sediment core, and therefore to the same direction as that of the
thin sections. However, the fact that the measurements in direction#2 show slight variations compared to that of the thin sections, shows that the method of measure with µCT scanned images and that on thin sections have differences.

As for the agreement study between the two methods, the GL17-04B and GL17-01D sediment core show the best percentage of agreement above 95% followed by GL17-04C with 94.17% (Table 4). Then with these sediment cores, we are more confident to say that the µCT method can be used as substitutes for the thin section method to do these measures. We are also
confident that using µCT with deformed sediment cores like GL17-05B would effectively select the least deformed locations to perform varve thickness measurements.

### 4.5 Using µCT to study varved sediments

The use of thin sections is key to study varved sediments (Bendle et al., 2015; Normandeau et al., 2019). Thin section represents a source of detailed information in one of its directions, thus facilitating the identification of microstructures and making it
possible to define the nature of the sediments (Bendle et al., 2015; Brauer and Casanova, 2001; Gagnon-Poiré et al., 2021). Its use to classify and measure the constituents of varves at short and specific time intervals has enabled environmental reconstructions of high temporal resolution (Bendle et al., 2015; Brauer and Casanova, 2001; Brauer et al., 2008; Gagnon-Poiré et al., 2021; Lauterbach et al., 2011). However, the process of analyzing sediments using thin sections is very often long and tedious (Normandeau et al., 2019; Francus and Asikainen, 2001; Lotter and Lemcke, 2008).
Measuring varves with µCT can save considerable amount of time time (Ballo et al., 2023; Emmanouilidis et al., 2020). In fact, just manufacturing thin sections take more than ten days while the µCT which can immediately be used to scans the cores without prior preparation. For a sediment core of 1.5 m height, it only takes around one day for data acquisition including the time to test the parameters of acquisition and reconstruction of the scan to ensure that quality images are obtained.

In addition, since repeated counts are important to obtain a good varved chronology and allows estimating varve counting
errors (Roop et al., 2016; Ballo et al., 2023; Ojala et al., 2012; Zolitschka et al., 2015; Martin-Puertas et al., 2021; Żarczyński et al., 2018), the µCT offers the possibility of making several counts in different directions. In this paper, twelve (12) different counts including six (6) in two perpendicular planes of the sediment core have been made. This makes it possible to better understand the three-dimensional structure of varves, to more easily detect missing layers or erosional features, steer the counts in parts of the volume that are less affected by disturbances, to make multiple measurements of layers with thickness lateral
variations. Hence, µCT allows obtaining robust varve counts and thickness measurements. Our paper outlines the need to conduct multiple counts in at least two perpendicular directions. The µCT also offers the possibility of easily obtaining variations in sediment thicknesses along several sediment cores. This is more practical for performing stratigraphic correlations (Fig. 12).



However, the reconstructed µCT data are large files (Table 1), so it is necessary to have high-capacity computers to analyze

them. Yet, a careful planning needs to be conducted prior to analysing an entire sequence. For our study, 170 GB were needed

for a sediment core 1.5 m long with a resolution of 45 µm. If a higher resolution is needed because laminations are thinner,

files to be handled will be larger. For example, to scan GL17-01D at 30 µm, the file size will be 223 GB, so 36 % larger than

those at 45 µm.  Also, image quality can be affected by artefacts like the beam hardening and the beam cone artefact which

can prevent the boundaries of the varves from being clearly distinguished (Laeveren D. , 2020; Sheppard et al., 2014; Meganck

et al., 2009; Davis, 2022).

A compromise is therefore to be considered to have sufficient resolution to obtain accurate thickness measurements and to

have quality images with reduced acquisition and reconstruction times (Du Plessis et al., 2017; Chatzinikolaou and

Keklikoglou, 2021; Dewanckele et al., 2020). It is therefore necessary to find the minimum resolution allowing the laminae to

be easily distinguished in the shortest possible scanning and reconstruction times. In our case study, it was possible to go well

below the resolution of 45 µm (voxel size) obtained for the analysis of the four sediment cores, but this did not provide

additional information. Indeed, at 45 µm we can easily distinguish the smallest lamination which is 0.92 mm thick. These

limits are visible up to 63 microns. Beyond that, the limits of the lamina become blurred (Fig. 18).

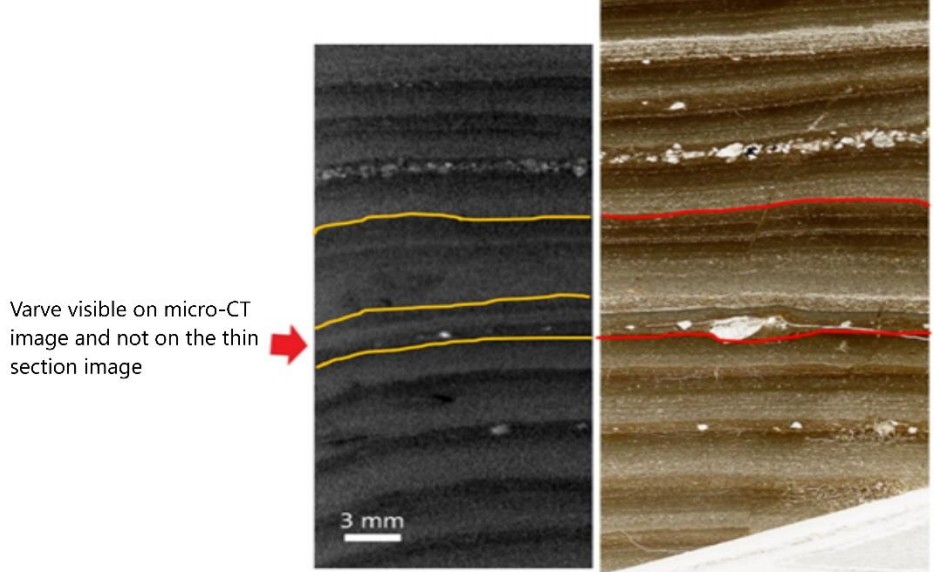

**Figure 13: The sediment core GL17-04C shows a lamination that is not visible on the thin section plane. In orange**

**lamination boundaries observed with µCT scan and in red lamination boundaries defined with thin sections.**



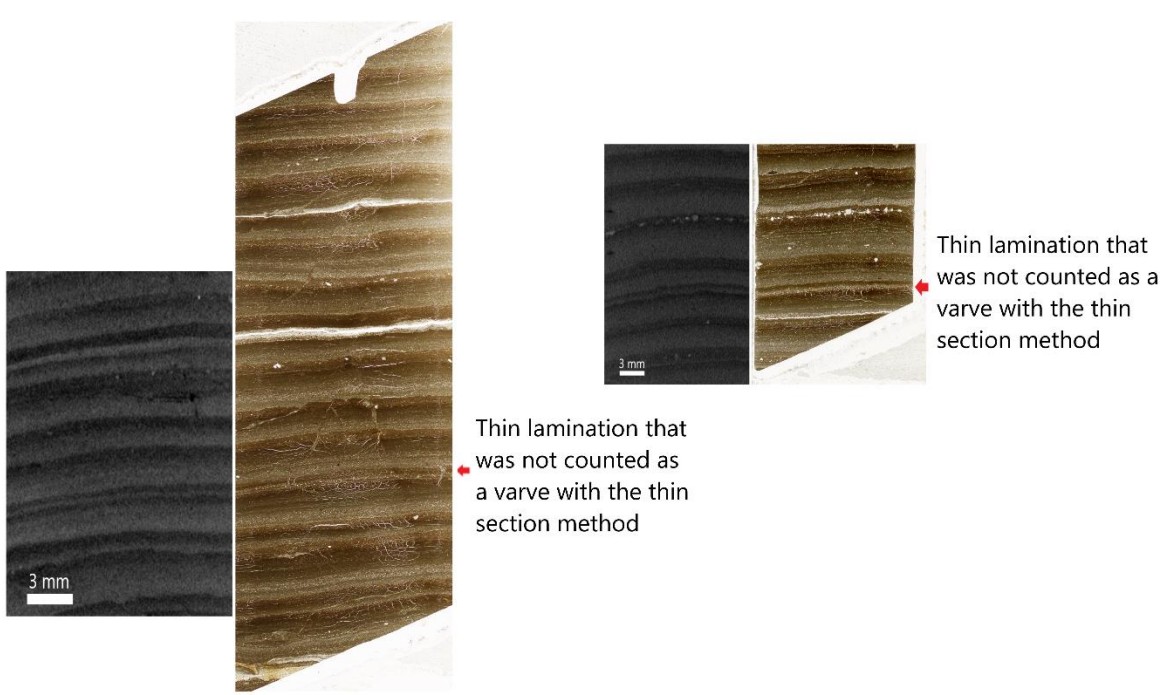

**Figure 14: Sediment core GL17-04C shows laminations that are not counted as varve with the thin sections. These laminations are counted as varves with µCT scans. But there is no sedimentological interpretation with µCT scans to confirm that they are actually varves.**

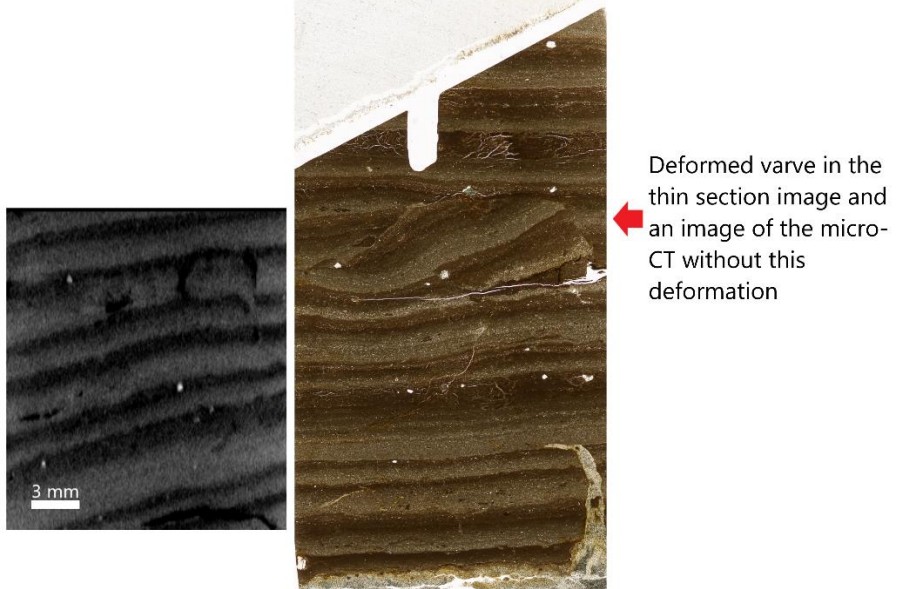

**Fig. 15: The deformations of varve in the thin section plane and the corresponding varve with µCT image in the sediment core GL17-05B.**



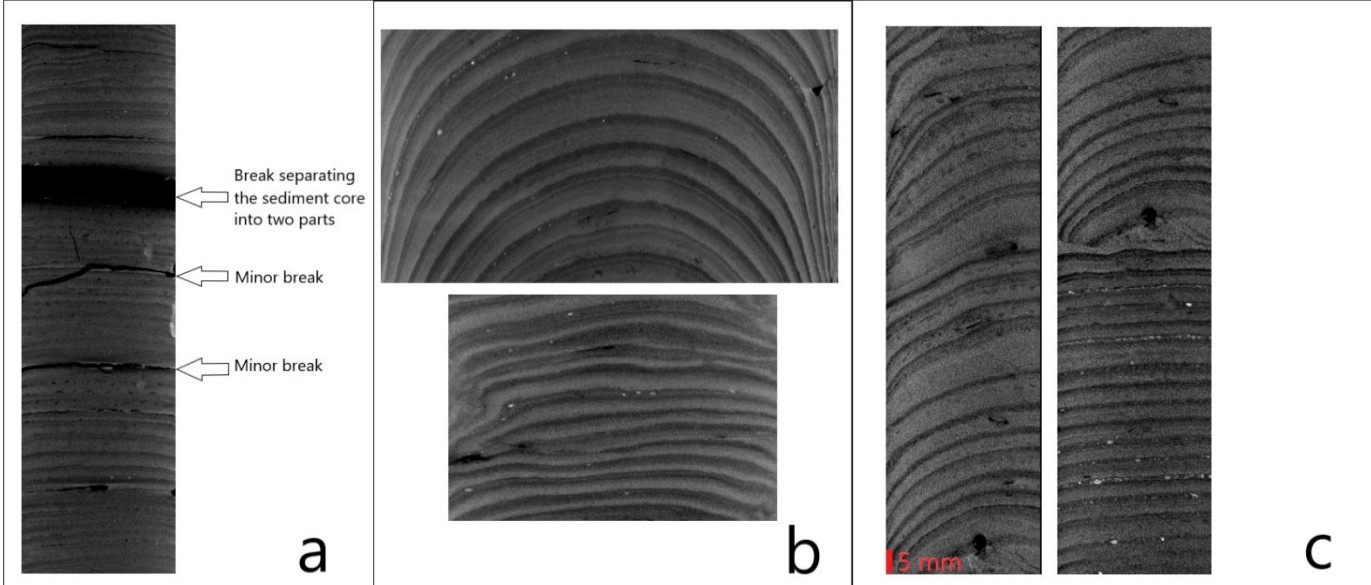

**Figure 16: (a) The sediment core GL17-01D broken parts. (b) deformations in the sediment core GL17-05B.(c) the sediment core GL17-04C shows oblique laminations with variable thickness on flat and horizontal laminations with constant thicknesses.**

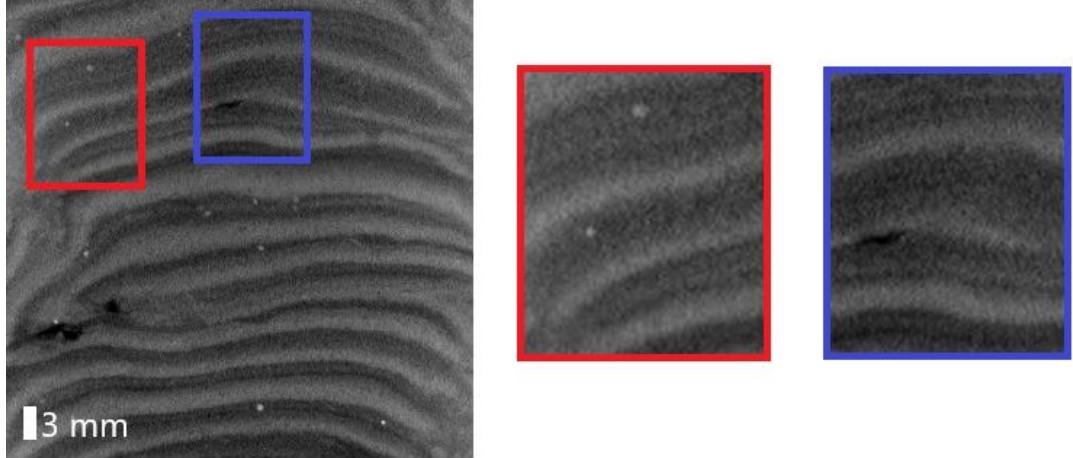

**Figure 17: Large VOIs which allows you to choose where to do more precise scans with in small VOIs. The images with red and purple outlines show how for the same varve, the thickness can vary.**



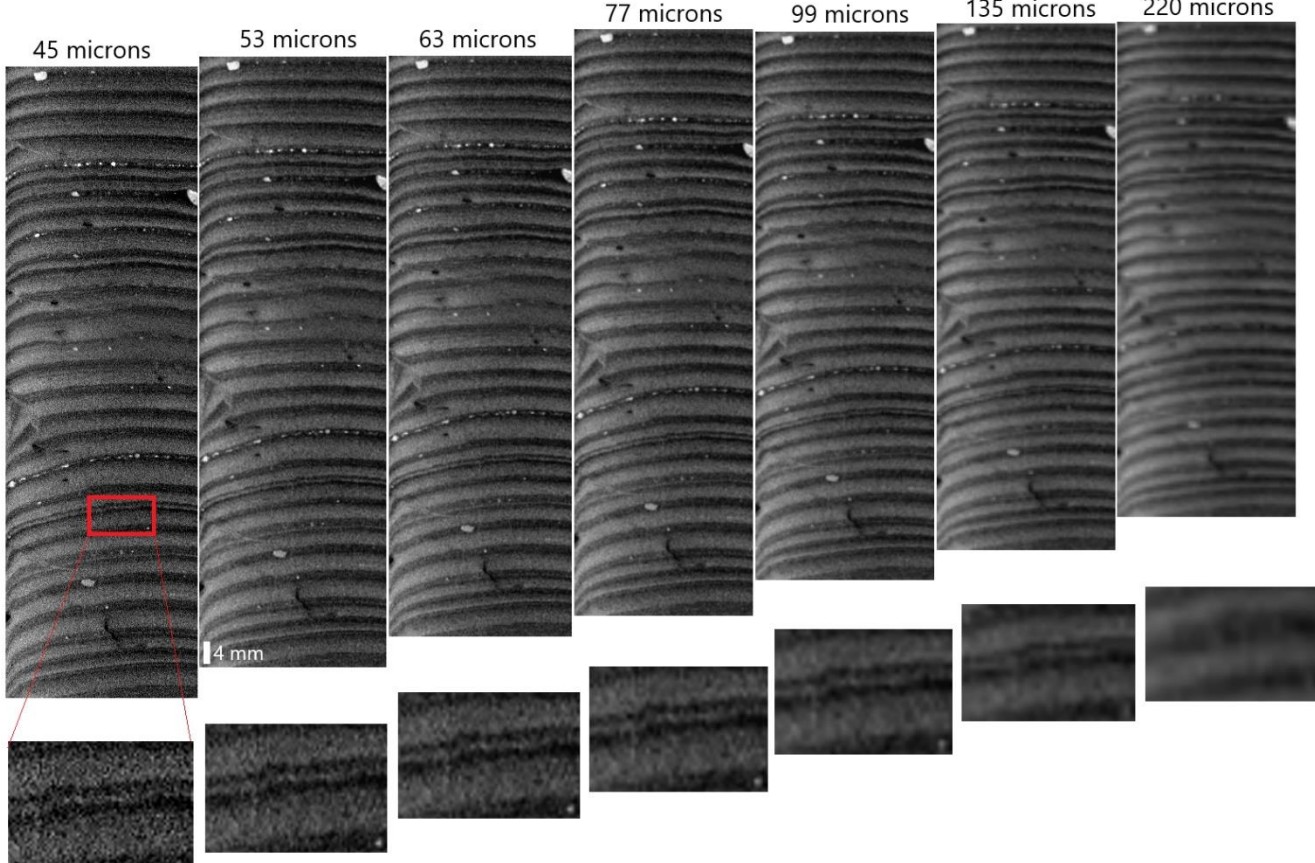

**Figure 18: This Fig. shows the ability to observe the limits of the smallest lamination depending on the resolution of the μCT images.**

## 5 Conclusion

The well-preserved sequence of varves of Grand Lake, the fact that varves are easy to recognize, thick, and not very disturbed has allowed to easily use μCT to perform varve counts and thickness measurements. By doing varve counts with μCT scans, we observed that using only μCT is not always enough to distinguish varve features, the use of thin sections remaining indispensable in some cases. Yet multiple counts in at least two perpendicular directions allows obtaining robust varve counts. Also, for thickness measurements, there is added value to use the μCT. The key results are the following:

- The μCT better reveals the three-dimensional structure of varves, facilitating the detection of missing layers or erosional features:

- The μCT allows making multiple measurements of layers in multiple directions allowing obtaining robust thickness measurements in varves with thickness lateral variations.

 

- In case of deformed sediment cores, µCT allows to locate the best area to perform the measurements.

Finally, thin sections are still necessary to determine the nature of the varved sedimentary facies and remain necessary for contentious cases. However, µCT offers the possibility of no longer making continuous thin sections along a sedimentary sequence for counting and constitutes a powerful tool to improve the quality of the thickness measurements through the access

of a 3D view allowing choosing the most representative part of varved record.

**Appendices**

**Appendix A: Additional information on the agreement study in the direction 1**

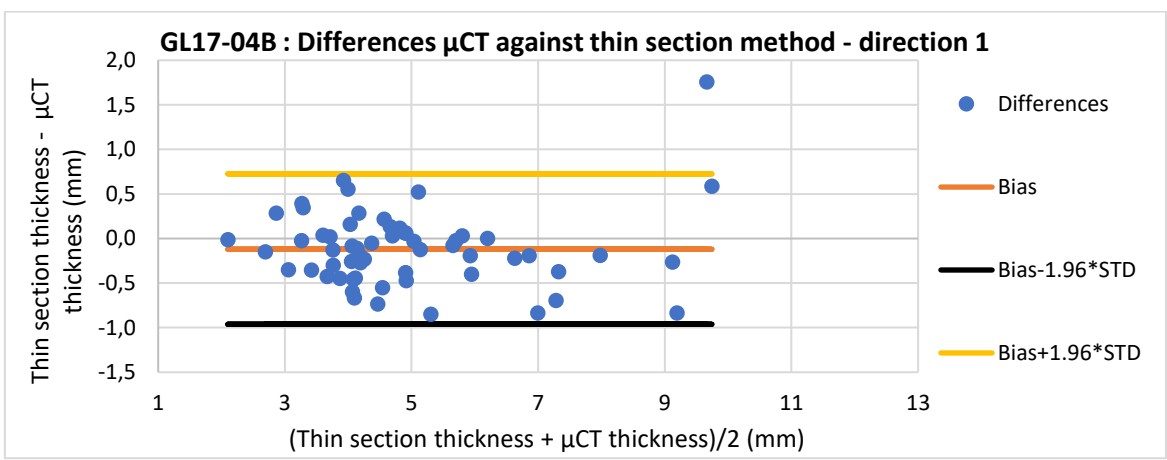

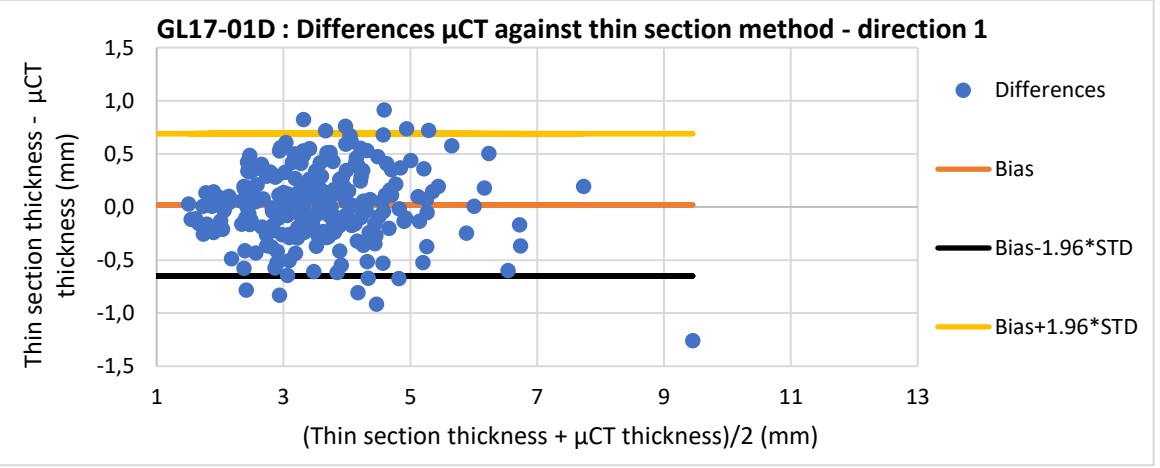





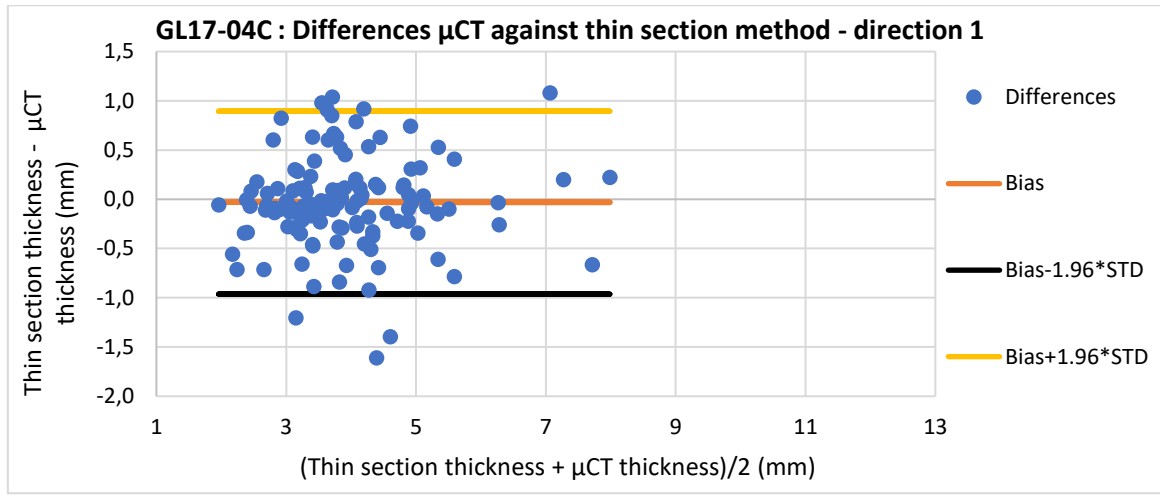

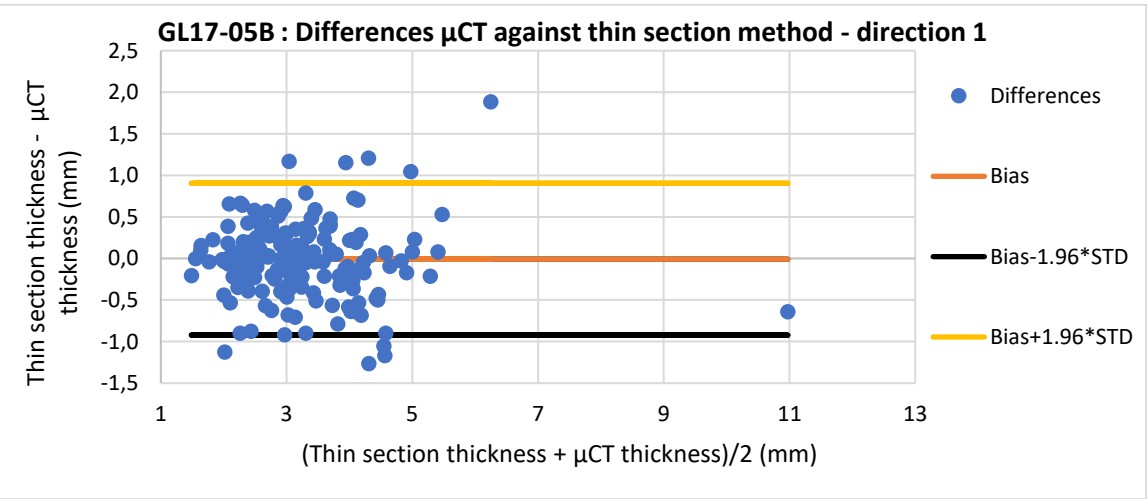

**Figure B1** Agreement study in the direction 1 of the two (2) methods respectively on the sediment cores GL17-04B, GL17-01D, GL17-04C and GL17-05B. The orange line represents the bias, and the two yellow and gray lines are the limits of the 390 95% confidence interval.



## Appendix B: Thickness measurements with μCT and thin sections

Table B1 The thickness measurements with μCT and thin sections

| | GL17-04B | | | | | | | | | | | |
|---|---|---|---|---|---|---|---|---|---|---|---|---|
| | μCT measurements direction 1 | | | | μCT measurements direction 2 | | | | Thin section measurements | | | |
| Years | 1st Count | 2nd Count | 3rd Count | Mean | 1st Count | 2nd Count | 3rd Count | Mean | 1st Count | 2nd Count | 3rd Count | Mean |
| 2016 | 3,46 | 3,8 | 3,9 | 3,72 | 3,46 | 3,75 | 3,77 | 3,66 | 4,49 | 4,24 | 4,09 | 4,27 |
| 2015 | 3,58 | 3,64 | 3,58 | 3,60 | 3,06 | 3,09 | 3,02 | 3,06 | 4,23 | 4,53 | 3,99 | 4,25 |
| 2014 | 6,12 | 6,06 | 5,89 | 6,02 | 5,98 | 6,1 | 6,05 | 6,04 | 5,80 | 5,56 | 6,14 | 5,83 |
| 2013 | 3,81 | 3,69 | 3,25 | 3,58 | 3,95 | 3,34 | 3,54 | 3,61 | 3,56 | 3,51 | 3,78 | 3,62 |
| 2012 | 5,05 | 5,14 | 4,97 | 5,05 | 4,79 | 5,2 | 5,08 | 5,02 | 4,94 | 5,11 | 5,01 | 5,02 |
| 2011 | 3,87 | 3,71 | 3,54 | 3,71 | 3,89 | 3,8 | 4,12 | 3,94 | 3,54 | 3,65 | 3,99 | 3,72 |
| 2010 | 4,23 | 3,76 | 3,48 | 3,82 | 3,31 | 2,88 | 3,43 | 3,21 | 3,76 | 3,85 | 3,48 | 3,70 |
| 2009 | 2,28 | 2,93 | 2,95 | 2,72 | 2,98 | 3,3 | 2,85 | 3,04 | 2,95 | 3,00 | 3,07 | 3,01 |
| 2008 | 7,64 | 7,73 | 7,52 | 7,63 | 7,45 | 7,37 | 7,38 | 7,40 | 7,25 | 6,70 | 6,85 | 6,93 |
| 2007 | 3,12 | 3 | 3,24 | 3,12 | 3,16 | 3,33 | 3,2 | 3,23 | 3,16 | 3,55 | 3,68 | 3,46 |
| 2006 | 4,28 | 4,23 | 4,5 | 4,34 | 4,31 | 4,4 | 4,36 | 4,36 | 3,96 | 3,73 | 3,99 | 3,89 |
| 2005 | 2,74 | 2,93 | 2,63 | 2,77 | 2,86 | 2,79 | 3,08 | 2,91 | 2,40 | 2,89 | 2,56 | 2,62 |
| 2004 | 4,2 | 4,23 | 4,44 | 4,29 | 4,03 | 4,3 | 4,11 | 4,15 | 4,28 | 3,98 | 4,02 | 4,09 |
| 2003 | 6,39 | 5,9 | 6,15 | 6,15 | 6,08 | 6,39 | 5,9 | 6,12 | 5,98 | 5,22 | 6,04 | 5,75 |
| 2002 | 4,72 | 4,68 | 4,65 | 4,68 | 4,46 | 4,39 | 4,54 | 4,46 | 4,56 | 5,18 | 4,40 | 4,71 |
| 2001 | 8,15 | 8,1 | 7,98 | 8,08 | 7,14 | 7,89 | 7,85 | 7,63 | 7,95 | 7,93 | 7,78 | 7,89 |
| 2000 | 5,78 | 5,52 | 5,84 | 5,71 | 5,67 | 5,66 | 5,37 | 5,57 | 5,94 | 5,80 | 5,33 | 5,69 |
| 1999 | 9,73 | 9,45 | 9,66 | 9,61 | 9,44 | 9,53 | 9,18 | 9,38 | 8,44 | 8,37 | 9,52 | 8,78 |
| 1998 | 7,54 | 7,11 | 7,59 | 7,41 | 7,53 | 7,16 | 7,23 | 7,31 | 6,49 | 6,49 | 6,76 | 6,58 |
| 1997 | 5,69 | 5,67 | 5,73 | 5,70 | 5,46 | 5,39 | 5,36 | 5,40 | 5,49 | 5,62 | 5,74 | 5,62 |
| 1996 | 7,08 | 7,67 | 7,78 | 7,51 | 6,78 | 6,75 | 7,31 | 6,95 | 7,05 | 7,29 | 7,07 | 7,14 |
| 1995 | 3,35 | 3,18 | 3,3 | 3,28 | 3,2 | 2,99 | 3,06 | 3,08 | 3,49 | 3,09 | 3,18 | 3,25 |
| 1994 | 4,27 | 4,09 | 3,96 | 4,11 | 4 | 3,8 | 4,34 | 4,05 | 4,22 | 4,15 | 3,69 | 4,02 |
| 1993 | 5 | 4,61 | 4,66 | 4,76 | 4,65 | 4,9 | 4,6 | 4,72 | 4,65 | 4,95 | 5,02 | 4,87 |
| 1992 | 5,43 | 5,01 | 4,86 | 5,10 | 4,92 | 5,32 | 4,73 | 4,99 | 4,61 | 4,50 | 5,04 | 4,72 |





| 1991 | 3,62 | 3,16 | 2,92 | 3,23 | 3,39 | 3,01 | 3,3 | 3,23 | 2,86 | 3,01 | 2,78 | 2,88 |
|---|---|---|---|---|---|---|---|---|---|---|---|---|
| 1990 | 3,33 | 3,65 | 3,82 | 3,60 | 3,4 | 3,46 | 3,41 | 3,42 | 3,42 | 3,23 | 3,09 | 3,24 |
| 1989 | 4,77 | 6,38 | 6,21 | 5,79 | 6,24 | 6,63 | 5,96 | 6,28 | 5,66 | 5,72 | 6,07 | 5,82 |
| 1988 | 4,46 | 4,26 | 4,27 | 4,33 | 3,95 | 4,16 | 4,45 | 4,19 | 4,04 | 4,01 | 4,12 | 4,06 |
| 1987 | 1,92 | 2,17 | 2,23 | 2,11 | 2,2 | 2,19 | 2,32 | 2,24 | 2,14 | 1,98 | 2,16 | 2,09 |
| 1986 | 5,41 | 5,04 | 5,16 | 5,20 | 5,28 | 5,06 | 5,15 | 5,16 | 5,06 | 5,13 | 5,04 | 5,08 |
| 1985 | 5,87 | 6,1 | 6,63 | 6,20 | 6,24 | 6,54 | 6,33 | 6,37 | 6,20 | 6,22 | 6,17 | 6,20 |
| 1984 | 4,94 | 5,02 | 4,69 | 4,88 | 4,74 | 4,83 | 4,13 | 4,57 | 5,57 | 4,83 | 4,42 | 4,94 |
| 1983 | 9,32 | 9,23 | 9,8 | 9,45 | 11,15 | 11,14 | 10,76 | 11,02 | 9,25 | 10,26 | 10,60 | 10,03 |
| 1982 | 4,45 | 4,28 | 4,47 | 4,40 | 4,28 | 4,17 | 4,49 | 4,31 | 4,36 | 4,25 | 4,42 | 4,35 |
| 1981 | 5,59 | 5,76 | 5,85 | 5,73 | 5,45 | 5,33 | 5,42 | 5,40 | 4,89 | 4,88 | 4,88 | 4,88 |
| 1980 | 4,06 | 4,13 | 3,9 | 4,03 | 4,05 | 3,65 | 4,08 | 3,93 | 4,82 | 4,02 | 4,10 | 4,31 |
| 1979 | 8,99 | 9,43 | 9,35 | 9,26 | 10,34 | 9,88 | 9,48 | 9,90 | 8,41 | 9,12 | 9,43 | 8,99 |
| 1978 | 7,36 | 6,73 | 6,78 | 6,96 | 7,25 | 6,54 | 6,7 | 6,83 | 7,11 | 6,41 | 6,77 | 6,76 |
| 1977 | 5,16 | 5,01 | 5,29 | 5,15 | 4,72 | 4,89 | 5,2 | 4,94 | 4,54 | 4,78 | 4,72 | 4,68 |
| 1976 | 4,64 | 4,21 | 4,08 | 4,31 | 4,04 | 4,23 | 4,07 | 4,11 | 3,84 | 4,02 | 3,69 | 3,85 |
| 1975 | 5,12 | 4,59 | 4,79 | 4,83 | 4,95 | 4,55 | 4,58 | 4,69 | 4,06 | 3,82 | 4,41 | 4,10 |
| 1974 | 3,85 | 4,13 | 3,67 | 3,88 | 3,64 | 3,45 | 3,65 | 3,58 | 3,39 | 3,47 | 3,51 | 3,46 |
| 1973 | 4,23 | 4,06 | 3,46 | 3,92 | 3,29 | 3,3 | 3,27 | 3,29 | 3,82 | 3,95 | 3,07 | 3,61 |
| 1972 | 7,34 | 8,43 | 10,58 | 8,78 | 11,73 | 12,03 | 12,02 | 11,93 | 10,13 | 10,21 | 11,27 | 10,54 |
| 1971 | 4,53 | 4,69 | 4,57 | 4,60 | 4,14 | 4,26 | 4,15 | 4,18 | 4,60 | 5,48 | 4,10 | 4,73 |
| 1970 | 4,25 | 3,89 | 3,73 | 3,96 | 3,8 | 4,14 | 3,7 | 3,88 | 4,41 | 3,31 | 4,61 | 4,11 |
| 1969 | 4,58 | 4,69 | 4,03 | 4,43 | 4,32 | 4,3 | 4,7 | 4,44 | 4,06 | 3,85 | 3,38 | 3,77 |
| 1968 | 4,17 | 3,92 | 4,2 | 4,10 | 4,01 | 3,86 | 4,37 | 4,08 | 3,19 | 3,36 | 4,41 | 3,65 |
| 1967 | 3,11 | 3,21 | 2,9 | 3,07 | 2,92 | 2,88 | 2,8 | 2,87 | 3,75 | 3,98 | 2,67 | 3,46 |
| 1966 | 6,84 | 6,72 | 6,67 | 6,74 | 6,84 | 6,8 | 6,84 | 6,83 | 6,74 | 6,26 | 6,56 | 6,52 |
| 1965 | 4,26 | 4,38 | 3,9 | 4,18 | 4,09 | 4,1 | 4,17 | 4,12 | 3,76 | 3,91 | 4,10 | 3,92 |
| 1964 | 4,31 | 4,23 | 4,04 | 4,19 | 4,53 | 4,44 | 4,26 | 4,41 | 3,98 | 4,06 | 4,20 | 4,08 |
| 1963 | 4,62 | 4,32 | 4,44 | 4,46 | 4,23 | 4,45 | 4,72 | 4,47 | 4,92 | 4,70 | 4,41 | 4,68 |
| 1962 | 4,67 | 4,31 | 4,13 | 4,37 | 3,9 | 3,93 | 4,2 | 4,01 | 3,77 | 4,03 | 4,61 | 4,14 |
| 1961 | 4,82 | 5,05 | 4,72 | 4,86 | 4,42 | 4,71 | 4,2 | 4,44 | 4,75 | 5,01 | 5,02 | 4,93 |
| 1960 | 4,67 | 4,97 | 4,81 | 4,82 | 4,55 | 4,6 | 4,66 | 4,60 | 4,36 | 4,23 | 4,20 | 4,26 |





| | | | | | | | | | | | |
|---|---|---|---|---|---|---|---|---|---|---|---|
| 1959 | 4,57 | 4,31 | 4,22 | 4,37 | 4,1 | 4,34 | 4,42 | 4,29 | 4,03 | 3,99 | 3,28 | 3,77 |
| 1958 | 5,14 | 4,85 | 4,55 | 4,85 | 4,86 | 4,45 | 4,31 | 4,54 | 5,05 | 5,42 | 5,64 | 5,37 |

| GL17-01D | | | | | | | | | | | |
|---|---|---|---|---|---|---|---|---|---|---|---|
| | µCT measurements direction 1 | | | | µCT measurements direction 2 | | | | Thin section measurements | | | |
| Years | 1st Count | 2nd Count | 3rd Count | Mean | 1st Count | 2nd Count | 3rd Count | Mean | 1st Count | 2nd Count | 3rd Count | Mean |
| 2016 | 2,46 | 2,43 | 2,48 | 2,46 | 2,46 | 2,32 | 2,53 | 2,44 | 2,97 | 2,82 | 2,78 | 2,86 |
| 2015 | 2,79 | 2,59 | 2,64 | 2,67 | 2,63 | 2,77 | 2,52 | 2,64 | 2,82 | 2,98 | 3,89 | 3,23 |
| 2014 | 2,57 | 2,53 | 2,56 | 2,55 | 2,6 | 2,66 | 3,04 | 2,77 | 2,53 | 2,39 | 2,36 | 2,43 |
| 2013 | 1,88 | 1,9 | 1,6 | 1,79 | 1,72 | 1,7 | 1,83 | 1,75 | 1,93 | 1,73 | 1,86 | 1,84 |
| 2012 | 3,96 | 4,01 | 4 | 3,99 | 3,91 | 3,66 | 3,72 | 3,76 | 4,30 | 4,34 | 4,48 | 4,37 |
| 2011 | 1,66 | 1,61 | 1,75 | 1,67 | 1,67 | 1,67 | 1,64 | 1,66 | 1,63 | 1,52 | 1,53 | 1,56 |
| 2010 | 1,66 | 1,66 | 1,87 | 1,73 | 1,75 | 1,87 | 1,75 | 1,79 | 1,64 | 1,82 | 1,74 | 1,73 |
| 2009 | 1,5 | 1,5 | 1,47 | 1,49 | 1,37 | 1,54 | 1,43 | 1,45 | 1,48 | 1,56 | 1,51 | 1,52 |
| 2008 | 2,48 | 2,48 | 2,41 | 2,46 | 2,16 | 2,17 | 2,15 | 2,16 | 2,53 | 2,33 | 2,20 | 2,35 |
| 2007 | 1,82 | 1,76 | 1,82 | 1,80 | 1,72 | 1,84 | 1,81 | 1,79 | 1,78 | 1,79 | 2,06 | 1,88 |
| 2006 | 2,12 | 2,08 | 2,03 | 2,08 | 2,02 | 2,03 | 2,16 | 2,07 | 2,07 | 1,80 | 1,95 | 1,94 |
| 2005 | 1,6 | 1,55 | 1,66 | 1,60 | 1,54 | 1,57 | 1,62 | 1,58 | 1,48 | 1,56 | 1,41 | 1,48 |
| 2004 | 1,76 | 1,61 | 1,76 | 1,71 | 1,89 | 1,73 | 1,54 | 1,72 | 1,78 | 1,84 | 1,91 | 1,84 |
| 2003 | 2,48 | 2,48 | 2,83 | 2,60 | 2,02 | 2,58 | 3,36 | 2,65 | 2,16 | 2,17 | 2,22 | 2,18 |
| 2002 | 2,62 | 2,38 | 2,4 | 2,47 | 2,38 | 2,44 | 1,93 | 2,25 | 2,45 | 2,70 | 2,56 | 2,57 |
| 2001 | 3,8 | 3,72 | 3,75 | 3,76 | 3,79 | 3,76 | 3,58 | 3,71 | 3,78 | 3,66 | 3,59 | 3,68 |
| 2000 | 2,31 | 2,25 | 2,25 | 2,27 | 2,62 | 2,15 | 2,23 | 2,33 | 2,47 | 2,69 | 2,66 | 2,61 |
| 1999 | 6,76 | 6,8 | 6,86 | 6,81 | 6,8 | 6,84 | 6,4 | 6,68 | 6,57 | 6,62 | 6,72 | 6,64 |
| 1998 | 2,73 | 2,85 | 2,96 | 2,85 | 3,25 | 3,21 | 2,99 | 3,15 | 2,81 | 2,85 | 2,75 | 2,80 |
| 1997 | 3,2 | 3,57 | 3,54 | 3,44 | 3,47 | 3,33 | 3,42 | 3,41 | 3,55 | 3,65 | 3,55 | 3,58 |
| 1996 | 3,02 | 2,91 | 2,83 | 2,92 | 2,9 | 2,99 | 2,73 | 2,87 | 2,91 | 2,81 | 2,97 | 2,90 |
| 1995 | 1,86 | 1,91 | 1,85 | 1,87 | 1,87 | 1,98 | 1,71 | 1,85 | 1,60 | 1,72 | 1,81 | 1,71 |
| 1994 | 2,06 | 2,36 | 1,94 | 2,12 | 2,47 | 2,67 | 2,46 | 2,53 | 2,17 | 2,11 | 2,19 | 2,16 |
| 1993 | 2,88 | 2,74 | 3,14 | 2,92 | 2,65 | 2,73 | 2,59 | 2,66 | 2,70 | 2,41 | 2,55 | 2,55 |
| 1992 | 2,36 | 2,26 | 2,65 | 2,42 | 2,1 | 2,3 | 2,08 | 2,16 | 2,14 | 2,30 | 2,34 | 2,26 |



| 1991 | 1,86 | 1,83 | 1,92 | 1,87 | 1,4 | 1,47 | 1,58 | 1,48 | 1,81 | 1,88 | 1,93 | 1,87 |
| 1990 | 1,82 | 1,89 | 1,88 | 1,86 | 1,97 | 1,61 | 1,56 | 1,71 | 1,75 | 1,42 | 1,65 | 1,60 |
| 1989 | 3,89 | 3,93 | 3,9 | 3,91 | 3,47 | 3,55 | 3,19 | 3,40 | 3,92 | 3,99 | 4,09 | 4,00 |
| 1988 | 2,56 | 2,4 | 2,48 | 2,48 | 1,82 | 2,15 | 2,13 | 2,03 | 2,55 | 2,73 | 2,78 | 2,69 |
| 1987 | 1,03 | 0,9 | 0,99 | 0,97 | 1,4 | 1,54 | 1,26 | 1,40 | 0,95 | 1,00 | 0,82 | 0,92 |
| 1986 | 2 | 2,03 | 1,83 | 1,95 | 2,38 | 2,65 | 2,12 | 2,38 | 2,17 | 1,99 | 1,96 | 2,04 |
| 1985 | 1,87 | 1,92 | 1,7 | 1,83 | 2,47 | 1,98 | 2,12 | 2,19 | 1,88 | 2,07 | 1,95 | 1,97 |
| 1984 | 2,7 | 2,6 | 2,63 | 2,64 | 2,7 | 2,51 | 2,51 | 2,57 | 2,70 | 2,65 | 2,80 | 2,72 |
| 1983 | 3,33 | 3,35 | 3,28 | 3,32 | 3 | 2,93 | 3,3 | 3,08 | 3,13 | 3,37 | 3,19 | 3,23 |
| 1982 | 2,66 | 2,79 | 2,81 | 2,75 | 2,64 | 2,7 | 2,92 | 2,75 | 2,65 | 2,57 | 2,47 | 2,56 |
| 1981 | 2,86 | 2,95 | 3,05 | 2,95 | 3,38 | 3,52 | 3,48 | 3,46 | 3,00 | 3,06 | 2,96 | 3,00 |
| 1980 | 2,35 | 2,5 | 2,81 | 2,55 | 2,56 | 2,52 | 2,5 | 2,53 | 2,32 | 2,27 | 2,57 | 2,39 |
| 1979 | 3,4 | 3,4 | 3,24 | 3,35 | 3,5 | 3,41 | 3,4 | 3,44 | 3,39 | 3,53 | 3,60 | 3,51 |
| 1978 | 3,09 | 3,04 | 3,19 | 3,11 | 3,51 | 3,41 | 3,52 | 3,48 | 3,12 | 3,09 | 3,14 | 3,12 |
| 1977 | 3,22 | 3,46 | 3,66 | 3,45 | 3,5 | 3,18 | 3,38 | 3,35 | 3,33 | 3,09 | 3,19 | 3,20 |
| 1976 | 1,87 | 1,76 | 1,97 | 1,87 | 1,58 | 1,7 | 1,79 | 1,69 | 1,54 | 1,86 | 1,67 | 1,69 |
| 1975 | 3,65 | 3,67 | 3,38 | 3,57 | 3,51 | 3,6 | 3,65 | 3,59 | 4,00 | 3,94 | 4,04 | 3,99 |
| 1974 | 2,15 | 2 | 2,13 | 2,09 | 1,99 | 2,13 | 1,92 | 2,01 | 2,17 | 2,23 | 2,17 | 2,19 |
| 1973 | 3,56 | 3,59 | 3,29 | 3,48 | 3,58 | 3,77 | 3,58 | 3,64 | 3,94 | 4,04 | 4,00 | 3,99 |
| 1972 | 4,14 | 4,47 | 4,36 | 4,32 | 4,42 | 4,33 | 4,12 | 4,29 | 4,01 | 4,00 | 3,99 | 4,00 |
| 1971 | 3,77 | 3,83 | 3,67 | 3,76 | 3,58 | 3,51 | 3,52 | 3,54 | 3,87 | 4,13 | 4,04 | 4,01 |
| 1970 | 2,93 | 2,83 | 2,85 | 2,87 | 3,08 | 2,95 | 3,04 | 3,02 | 3,05 | 2,98 | 2,91 | 2,98 |
| 1969 | 3,44 | 3,35 | 3,83 | 3,54 | 3,6 | 4,07 | 3,57 | 3,75 | 3,38 | 3,23 | 3,51 | 3,37 |
| 1968 | 2,15 | 2,3 | 2,4 | 2,28 | 2,42 | 2,19 | 2,12 | 2,24 | 2,48 | 2,56 | 2,39 | 2,48 |
| 1967 | 2,42 | 2,26 | 2,22 | 2,30 | 2,32 | 2,4 | 2,19 | 2,30 | 2,48 | 2,59 | 2,34 | 2,47 |
| 1966 | 3,09 | 3,1 | 2,85 | 3,01 | 3,21 | 3,14 | 3,26 | 3,20 | 3,23 | 3,03 | 3,14 | 3,13 |
| 1965 | 2,99 | 2,89 | 3,12 | 3,00 | 2,64 | 2,67 | 2,42 | 2,58 | 2,68 | 2,58 | 2,61 | 2,62 |
| 1964 | 2,94 | 2,87 | 3,12 | 2,98 | 2,63 | 2,51 | 2,73 | 2,62 | 2,79 | 2,99 | 2,83 | 2,87 |
| 1963 | 3,35 | 3,43 | 3,35 | 3,38 | 3,27 | 3,39 | 3,47 | 3,38 | 3,36 | 3,29 | 3,27 | 3,31 |
| 1962 | 3,68 | 3,68 | 3,64 | 3,67 | 3,41 | 3,64 | 3,16 | 3,40 | 3,69 | 3,73 | 3,67 | 3,70 |
| 1961 | 3,96 | 3,99 | 4,29 | 4,08 | 4,11 | 4,21 | 4,2 | 4,17 | 4,51 | 4,26 | 4,33 | 4,37 |
| 1960 | 3,16 | 3,11 | 3,24 | 3,17 | 3,58 | 3,3 | 3,36 | 3,41 | 3,31 | 3,25 | 3,36 | 3,31 |





| 1959 | 3,35 | 2,99 | 3,12 | 3,15 | 2,55 | 2,35 | 2,48 | 2,46 | 2,48 | 2,53 | 2,55 | 2,52 |
| 1958 | 3,6 | 3,68 | 3,51 | 3,60 | 3,96 | 3,95 | 3,97 | 3,96 | 4,46 | 4,40 | 4,21 | 4,36 |
| 1957 | 5 | 5,44 | 5,59 | 5,34 | 5,41 | 5,47 | 5,13 | 5,34 | 5,46 | 5,68 | 5,46 | 5,53 |
| 1956 | 3,46 | 3,44 | 2,98 | 3,29 | 3,19 | 2,99 | 3,25 | 3,14 | 3,19 | 3,02 | 2,88 | 3,03 |
| 1955 | 2,74 | 2,8 | 3,25 | 2,93 | 2,99 | 3,05 | 2,97 | 3,00 | 3,33 | 3,48 | 3,23 | 3,35 |
| 1954 | 2,82 | 2,75 | 3,01 | 2,86 | 2,77 | 3,05 | 2,95 | 2,92 | 2,68 | 2,61 | 2,50 | 2,60 |
| 1953 | 1,59 | 1,53 | 1,48 | 1,53 | 2,31 | 2,48 | 2,32 | 2,37 | 1,33 | 3,22 | 3,19 | 2,58 |
| 1952 | 2,26 | 2,29 | 2,13 | 2,23 | 2,46 | 2,84 | 2,62 | 2,64 | 2,60 | 2,82 | 2,71 | 2,71 |
| 1951 | 5,39 | 5,09 | 4,99 | 5,16 | 4,81 | 4,85 | 4,58 | 4,75 | 4,58 | 4,45 | 4,41 | 4,48 |
| 1950 | 2,2 | 2,42 | 2,31 | 2,31 | 2,56 | 2,49 | 2,45 | 2,50 | 2,60 | 2,67 | 2,64 | 2,64 |
| 1949 | 4,13 | 3,92 | 4,05 | 4,03 | 4,01 | 4,07 | 4 | 4,03 | 4,14 | 3,97 | 3,77 | 3,96 |
| 1948 | 3,78 | 3,96 | 3,41 | 3,72 | 3,69 | 3,64 | 3,56 | 3,63 | 3,87 | 4,22 | 3,54 | 3,88 |
| 1947 | 4,58 | 4,66 | 4,7 | 4,65 | 4,43 | 4,75 | 4,37 | 4,52 | 4,58 | 4,91 | 4,78 | 4,76 |
| 1946 | 3,55 | 3,68 | 3,78 | 3,67 | 2,71 | 2,68 | 2,69 | 2,69 | 2,10 | 2,11 | 2,06 | 2,09 |
| 1945 | 3,29 | 2,69 | 3,24 | 3,07 | 3,36 | 3,47 | 3,34 | 3,39 | 3,76 | 3,24 | 3,44 | 3,48 |
| 1944 | 5,8 | 6,27 | 6,16 | 6,08 | 6,3 | 6,25 | 6,26 | 6,27 | 6,16 | 6,40 | 6,20 | 6,25 |
| 1943 | 3,08 | 2,78 | 2,73 | 2,86 | 3,02 | 3,04 | 3,03 | 3,03 | 3,25 | 3,22 | 3,09 | 3,19 |
| 1942 | 2,62 | 2,96 | 2,63 | 2,74 | 2,73 | 2,78 | 2,71 | 2,74 | 2,85 | 3,13 | 3,10 | 3,03 |
| 1941 | 4,6 | 4,68 | 4,31 | 4,53 | 4,55 | 4,54 | 4,6 | 4,56 | 4,51 | 4,75 | 4,48 | 4,58 |
| 1940 | 3,57 | 3,71 | 3,58 | 3,62 | 3,69 | 3,85 | 3,5 | 3,68 | 3,82 | 3,64 | 3,79 | 3,75 |
| 1939 | 2,99 | 3,28 | 2,95 | 3,07 | 3,26 | 3,12 | 3,19 | 3,19 | 3,43 | 3,65 | 3,60 | 3,56 |
| 1938 | 3,59 | 3,55 | 3,36 | 3,50 | 3,54 | 3,61 | 3,69 | 3,61 | 3,50 | 3,74 | 3,75 | 3,66 |
| 1937 | 3,74 | 3,82 | 3,66 | 3,74 | 3,85 | 3,71 | 3,73 | 3,76 | 3,62 | 3,40 | 3,50 | 3,51 |
| 1936 | 5,04 | 4,96 | 4,79 | 4,93 | 5,68 | 5,91 | 4,78 | 5,46 | 5,92 | 5,45 | 5,58 | 5,65 |
| 1935 | 2,93 | 2,77 | 2,24 | 2,65 | 2,01 | 2 | 2,22 | 2,08 | 2,10 | 2,09 | 1,95 | 2,05 |
| 1934 | 2,47 | 2,3 | 2,39 | 2,39 | 2,35 | 2,45 | 2,41 | 2,40 | 2,48 | 2,29 | 2,49 | 2,42 |
| 1933 | 2,91 | 2,88 | 3,04 | 2,94 | 3,26 | 3,22 | 3,04 | 3,17 | 3,31 | 3,45 | 3,55 | 3,44 |
| 1932 | 3,62 | 4,19 | 3,77 | 3,86 | 4,02 | 3,99 | 4,11 | 4,04 | 4,06 | 3,98 | 4,10 | 4,05 |
| 1931 | 2,91 | 3,16 | 3,11 | 3,06 | 3,76 | 3,77 | 3,78 | 3,77 | 3,66 | 3,56 | 3,84 | 3,69 |
| 1930 | 7,43 | 7,67 | 7,81 | 7,64 | 7,85 | 7,85 | 7,78 | 7,83 | 7,50 | 8,04 | 7,94 | 7,83 |
| 1929 | 3,97 | 4,58 | 5,3 | 4,62 | 4,65 | 4,56 | 4,52 | 4,58 | 4,58 | 4,11 | 4,11 | 4,27 |
| 1928 | 3,64 | 4,05 | 3,77 | 3,82 | 3,98 | 3,87 | 3,83 | 3,89 | 4,27 | 4,05 | 4,17 | 4,16 |





| 1927 | 3,9 | 4,71 | 5,11 | 4,57 | 5,15 | 5,29 | 4,82 | 5,09 | 5,33 | 5,19 | 5,40 | 5,31 |
| 1926 | 4,06 | 3,95 | 3,78 | 3,93 | 4,29 | 4,23 | 4,16 | 4,23 | 4,27 | 4,46 | 4,31 | 4,35 |
| 1925 | 4,47 | 4,65 | 4,62 | 4,58 | 4,28 | 4,5 | 4,52 | 4,43 | 4,00 | 4,12 | 4,07 | 4,06 |
| 1924 | 9,97 | 9,97 | 10,07 | 10,00 | 9,13 | 9,04 | 9,22 | 9,13 | 9,03 | 8,34 | 9,10 | 8,82 |
| 1923 | 4,38 | 3,74 | 4,16 | 4,09 | 3,97 | 4,23 | 3,84 | 4,01 | 3,82 | 4,01 | 4,48 | 4,10 |
| 1922 | 3,77 | 3,83 | 4,25 | 3,95 | 4,33 | 3,96 | 3,3 | 3,86 | 3,94 | 3,42 | 3,96 | 3,77 |
| 1921 | 4,44 | 4,39 | 4,35 | 4,39 | 4,13 | 4,08 | 4,18 | 4,13 | 3,76 | 3,71 | 3,56 | 3,68 |
| 1920 | 5,21 | 5,53 | 5,35 | 5,36 | 5,5 | 6 | 6,05 | 5,85 | 6,01 | 5,88 | 5,92 | 5,94 |
| 1919 | 3,34 | 3,5 | 3,39 | 3,41 | 3,61 | 3,32 | 3,39 | 3,44 | 2,98 | 2,99 | 2,94 | 2,97 |
| 1918 | 5,94 | 6,32 | 5,75 | 6,00 | 5,79 | 5,88 | 5,63 | 5,77 | 5,98 | 6,18 | 5,86 | 6,01 |
| 1917 | 3,15 | 2,95 | 2,66 | 2,92 | 2,9 | 3,2 | 2,97 | 3,02 | 3,25 | 3,52 | 3,48 | 3,42 |
| 1916 | 2,83 | 2,67 | 2,56 | 2,69 | 2,8 | 3,01 | 2,65 | 2,82 | 3,24 | 3,23 | 3,24 | 3,24 |
| 1915 | 2,34 | 2,79 | 2,19 | 2,44 | 2,15 | 2,33 | 2,59 | 2,36 | 2,22 | 2,60 | 2,46 | 2,43 |
| 1914 | 3,13 | 3,06 | 3,11 | 3,10 | 2,64 | 2,82 | 2,81 | 2,76 | 3,06 | 3,01 | 2,98 | 3,02 |
| 1913 | 3,08 | 3,15 | 2,93 | 3,05 | 3,3 | 3,44 | 3,75 | 3,50 | 3,24 | 3,36 | 3,36 | 3,32 |
| 1912 | 3,48 | 3,43 | 3,11 | 3,34 | 3,45 | 3,2 | 2,91 | 3,19 | 3,55 | 3,86 | 3,62 | 3,68 |
| 1911 | 2,17 | 2,14 | 2,36 | 2,22 | 2,42 | 2,44 | 2,41 | 2,42 | 2,54 | 2,68 | 2,71 | 2,64 |
| 1910 | 4,04 | 4,27 | 4,32 | 4,21 | 4,02 | 3,87 | 4,16 | 4,02 | 4,00 | 4,04 | 4,11 | 4,05 |
| 1909 | 4,26 | 3,79 | 4,22 | 4,09 | 3,89 | 3,93 | 3,9 | 3,91 | 4,20 | 4,42 | 4,38 | 4,33 |
| 1908 | 4,2 | 4,24 | 3,56 | 4,00 | 3,92 | 4,2 | 4,04 | 4,05 | 3,95 | 3,95 | 3,82 | 3,91 |
| 1907 | 4,45 | 4,46 | 4,37 | 4,43 | 4,56 | 4,63 | 4,8 | 4,66 | 4,76 | 4,96 | 4,77 | 4,83 |
| 1906 | 5,49 | 5,6 | 5,25 | 5,45 | 5,16 | 5,11 | 5,3 | 5,19 | 5,07 | 5,13 | 5,02 | 5,07 |
| 1905 | 4,63 | 4,75 | 4,61 | 4,66 | 4,88 | 4,7 | 4,94 | 4,84 | 5,02 | 5,06 | 5,01 | 5,03 |
| 1904 | 2,73 | 2,11 | 2,09 | 2,31 | 2,51 | 2,33 | 2,52 | 2,45 | 2,23 | 2,54 | 2,32 | 2,36 |
| 1903 | 2,45 | 3,24 | 2,51 | 2,73 | 2,6 | 2,74 | 2,66 | 2,67 | 3,05 | 2,89 | 3,10 | 3,01 |
| 1902 | 2,83 | 3,02 | 3,29 | 3,05 | 3,05 | 2,98 | 3,13 | 3,05 | 3,05 | 3,08 | 3,27 | 3,13 |
| 1901 | 3,68 | 3,99 | 3,42 | 3,70 | 3,91 | 3,85 | 3,76 | 3,84 | 3,95 | 3,62 | 3,94 | 3,84 |
| 1900 | 5,04 | 4,76 | 4,19 | 4,66 | 4,43 | 4,96 | 4,97 | 4,79 | 4,83 | 4,72 | 5,08 | 4,88 |
| 1899 | 2,77 | 3,1 | 2,48 | 2,78 | 2,29 | 2,62 | 2,83 | 2,58 | 2,30 | 2,42 | 2,33 | 2,35 |
| 1898 | 5,87 | 5,88 | 6,27 | 6,01 | 6,05 | 5,89 | 5,97 | 5,97 | 5,92 | 5,60 | 5,76 | 5,76 |
| 1897 | 3,09 | 3,29 | 3,59 | 3,32 | 3,67 | 3,01 | 3,41 | 3,36 | 3,25 | 3,12 | 3,01 | 3,13 |
| 1896 | 3,32 | 3,04 | 3,24 | 3,20 | 3,29 | 3,07 | 3,19 | 3,18 | 3,18 | 3,44 | 3,58 | 3,40 |




| 1895 | 3,41 | 3,23 | 3,2 | 3,28 | 2,9 | 3,08 | 3,16 | 3,05 | 3,38 | 3,57 | 3,52 | 3,49 |
| 1894 | 4,14 | 4,33 | 4,5 | 4,32 | 4,38 | 4,28 | 4,41 | 4,36 | 4,51 | 4,05 | 4,29 | 4,28 |
| 1893 | 3,94 | 3,91 | 3,61 | 3,82 | 3,69 | 3,87 | 3,77 | 3,78 | 3,63 | 3,56 | 3,40 | 3,53 |
| 1892 | 3,01 | 2,71 | 2,76 | 2,83 | 2,46 | 2,68 | 2,59 | 2,58 | 2,92 | 2,70 | 2,83 | 2,82 |
| 1891 | 4,96 | 4,59 | 4,95 | 4,83 | 4,68 | 4,81 | 4,47 | 4,65 | 4,46 | 4,29 | 4,16 | 4,30 |
| 1890 | 7,29 | 7,15 | 7,14 | 7,19 | 6,83 | 6,78 | 6,69 | 6,77 | 6,60 | 6,63 | 6,42 | 6,55 |
| 1889 | 3,31 | 3,46 | 3,25 | 3,34 | 3,25 | 3,4 | 3,18 | 3,28 | 3,55 | 3,49 | 3,29 | 3,44 |
| 1888 | 3,78 | 3,54 | 3,42 | 3,58 | 4,26 | 3,49 | 4,09 | 3,95 | 3,62 | 3,70 | 3,47 | 3,60 |
| 1887 | 3,78 | 4,12 | 3,96 | 3,95 | 3,79 | 3,77 | 3,5 | 3,69 | 4,20 | 3,99 | 4,12 | 4,10 |
| 1886 | 4,8 | 4,47 | 4,51 | 4,59 | 4,53 | 4,34 | 4,68 | 4,52 | 4,13 | 4,36 | 4,47 | 4,32 |
| 1885 | 3,65 | 3,85 | 4,15 | 3,88 | 3,88 | 3,58 | 3,98 | 3,81 | 3,95 | 4,12 | 4,06 | 4,04 |
| 1884 | 5,34 | 4,99 | 5,31 | 5,21 | 5,27 | 5,24 | 5,06 | 5,19 | 5,27 | 5,04 | 4,92 | 5,08 |
| 1883 | 4,99 | 4,91 | 5,03 | 4,98 | 4,71 | 4,81 | 5,02 | 4,85 | 4,58 | 4,87 | 5,16 | 4,87 |
| 1882 | 2,76 | 2,92 | 3,27 | 2,98 | 3,16 | 2,84 | 2,99 | 3,00 | 2,74 | 2,76 | 2,57 | 2,69 |
| 1881 | 4,24 | 4,38 | 4,09 | 4,24 | 4,17 | 4,81 | 4,73 | 4,57 | 4,76 | 5,02 | 4,96 | 4,91 |
| 1880 | 4,33 | 4,12 | 4,55 | 4,33 | 5,14 | 4,14 | 5,13 | 4,80 | 4,39 | 4,32 | 4,49 | 4,40 |
| 1879 | 3,57 | 3,01 | 3,27 | 3,28 | 3,07 | 3,41 | 3,08 | 3,19 | 3,68 | 3,37 | 3,58 | 3,54 |
| 1878 | 4,42 | 4,35 | 5,03 | 4,60 | 4 | 4,71 | 4,01 | 4,24 | 4,58 | 4,50 | 4,59 | 4,56 |
| 1877 | 3,67 | 3,2 | 3,5 | 3,46 | 3,62 | 3,13 | 3,64 | 3,46 | 3,62 | 3,88 | 3,73 | 3,74 |
| 1876 | 3,21 | 3,34 | 3,28 | 3,28 | 3,23 | 3,05 | 3,16 | 3,15 | 2,74 | 2,96 | 2,81 | 2,84 |
| 1875 | 3,91 | 3,78 | 4,14 | 3,94 | 4,29 | 4,42 | 4,2 | 4,30 | 4,38 | 4,65 | 4,45 | 4,49 |
| 1874 | 5,2 | 5,45 | 5,74 | 5,46 | 5,1 | 5,22 | 5,11 | 5,14 | 5,00 | 5,01 | 4,81 | 4,94 |
| 1873 | 4,79 | 4,39 | 4,51 | 4,56 | 3,39 | 3,51 | 3,35 | 3,42 | 3,81 | 3,21 | 3,68 | 3,57 |
| 1872 | 3,24 | 2,8 | 2,77 | 2,94 | 2,87 | 2,82 | 3,25 | 2,98 | 2,99 | 3,27 | 2,96 | 3,07 |
| 1871 | 4,66 | 4,47 | 4,66 | 4,60 | 5,17 | 4,77 | 4,21 | 4,72 | 4,65 | 4,76 | 4,86 | 4,76 |
| 1870 | 5,06 | 4,3 | 4,66 | 4,67 | 4,02 | 4,59 | 4,39 | 4,33 | 3,94 | 4,08 | 3,98 | 4,00 |
| 1869 | 4,12 | 4,12 | 4,15 | 4,13 | 4,29 | 5,03 | 4,61 | 4,64 | 5,03 | 4,64 | 5,46 | 5,04 |
| 1868 | 3,17 | 3,89 | 4,22 | 3,76 | 4,26 | 4,25 | 4,26 | 4,26 | 4,20 | 4,60 | 4,31 | 4,37 |
| 1867 | 3,09 | 3,44 | 3,26 | 3,26 | 3,19 | 3,26 | 3,27 | 3,24 | 3,25 | 3,42 | 3,18 | 3,28 |
| 1866 | 3,89 | 3,61 | 3,26 | 3,59 | 3,23 | 2,77 | 3,14 | 3,05 | 3,56 | 3,80 | 3,77 | 3,71 |
| 1865 | 3,79 | 3,73 | 3,74 | 3,75 | 3,61 | 3,87 | 3,97 | 3,82 | 3,88 | 3,74 | 3,61 | 3,74 |
| 1864 | 3,81 | 3,57 | 3,68 | 3,69 | 3,63 | 3,7 | 3,77 | 3,70 | 3,68 | 3,86 | 3,73 | 3,76 |



| 1863 | 5,73 | 5    | 4,93 | 5,22 | 5,58 | 5,31 | 5,23 | 5,37 | 5,35 | 5,29 | 5,23 | 5,29 |
|------|------|------|------|------|------|------|------|------|------|------|------|------|
| 1862 | 4,15 | 3,92 | 3,84 | 3,97 | 3,86 | 4    | 3,9  | 3,92 | 4,04 | 3,99 | 4,01 | 4,01 |
| 1861 | 2,51 | 2,89 | 2,62 | 2,67 | 2,54 | 2,76 | 2,88 | 2,73 | 2,98 | 3,34 | 3,27 | 3,20 |
| 1860 | 3,34 | 3,22 | 3,36 | 3,31 | 3,8  | 3,9  | 4,01 | 3,90 | 3,95 | 3,98 | 4,14 | 4,02 |
| 1859 | 4    | 3,99 | 3,93 | 3,97 | 3,86 | 3,67 | 3,8  | 3,78 | 3,87 | 3,78 | 3,89 | 3,85 |
| 1858 | 4,83 | 5,07 | 5,01 | 4,97 | 4,75 | 4,52 | 4,62 | 4,63 | 4,84 | 4,78 | 4,88 | 4,83 |
| 1857 | 4,33 | 3,91 | 4,32 | 4,19 | 3,64 | 3,48 | 3,62 | 3,58 | 3,62 | 3,55 | 3,74 | 3,64 |
| 1856 | 4,98 | 5,61 | 5,25 | 5,28 | 5,33 | 5,31 | 4,87 | 5,17 | 5,47 | 5,45 | 5,35 | 5,42 |
| 1855 | 3,41 | 3,16 | 3,35 | 3,31 | 2,79 | 3,29 | 3,25 | 3,11 | 3,37 | 3,27 | 3,39 | 3,34 |
| 1854 | 3,49 | 3,94 | 3,66 | 3,70 | 4,03 | 3,55 | 3,17 | 3,58 | 3,63 | 3,70 | 3,67 | 3,67 |
| 1853 | 5,37 | 5,13 | 4,73 | 5,08 | 5,19 | 5,21 | 4,88 | 5,09 | 5,15 | 5,12 | 5,24 | 5,17 |
| 1852 | 3,85 | 4,29 | 4,09 | 4,08 | 4,63 | 4,28 | 4,13 | 4,35 | 4,57 | 4,19 | 4,49 | 4,42 |
| 1851 | 4,13 | 3,89 | 3,59 | 3,87 | 3,75 | 3,66 | 3,44 | 3,62 | 3,63 | 3,78 | 3,66 | 3,69 |
| 1850 | 3,33 | 3,24 | 3,15 | 3,24 | 2,93 | 3,29 | 3,04 | 3,09 | 2,98 | 2,87 | 2,99 | 2,95 |
| 1849 | 2,94 | 2,92 | 2,78 | 2,88 | 2,3  | 2,18 | 2,12 | 2,20 | 1,91 | 1,56 | 1,87 | 1,78 |
| 1848 | 2,92 | 2,88 | 3    | 2,93 | 2,82 | 2,75 | 3,01 | 2,86 | 2,54 | 2,67 | 2,51 | 2,57 |
| 1847 | 3,14 | 3,26 | 3,01 | 3,14 | 3,12 | 3,36 | 3,24 | 3,24 | 3,63 | 3,68 | 3,74 | 3,68 |
| 1846 | 2,41 | 2,4  | 2,5  | 2,44 | 2,23 | 2,18 | 2,08 | 2,16 | 1,84 | 2,06 | 1,89 | 1,93 |
| 1845 | 3,68 | 3,63 | 3,36 | 3,56 | 3,73 | 3,12 | 3,27 | 3,37 | 3,31 | 3,63 | 3,51 | 3,48 |
| 1844 | 4,1  | 3,93 | 4,05 | 4,03 | 3,81 | 3,76 | 4,06 | 3,88 | 4,00 | 4,10 | 4,04 | 4,05 |
| 1843 | 3,37 | 3,72 | 3,01 | 3,37 | 3,46 | 3,28 | 3,07 | 3,27 | 3,18 | 3,34 | 3,42 | 3,31 |
| 1842 | 4,02 | 4,59 | 3,89 | 4,17 | 4,11 | 3,94 | 3,86 | 3,97 | 4,01 | 3,92 | 4,04 | 3,99 |
| 1841 | 3,78 | 3,77 | 3,89 | 3,81 | 3,81 | 3,88 | 3,73 | 3,81 | 3,87 | 3,81 | 3,53 | 3,74 |
| 1840 | 2,52 | 2,43 | 2,5  | 2,48 | 2,53 | 2,42 | 2,39 | 2,45 | 2,42 | 2,56 | 2,27 | 2,42 |
| 1839 | 3,05 | 3    | 3,1  | 3,05 | 3,38 | 3,54 | 3,52 | 3,48 | 3,60 | 3,62 | 3,51 | 3,58 |
| 1838 | 3,22 | 3,1  | 3,15 | 3,16 | 2,84 | 2,49 | 2,27 | 2,53 | 2,49 | 2,59 | 2,66 | 2,58 |
| 1837 | 2,81 | 3    | 2,9  | 2,90 | 3,38 | 3,48 | 3,46 | 3,44 | 3,56 | 3,93 | 3,69 | 3,72 |
| 1836 | 3,31 | 3,06 | 3,15 | 3,17 | 2,5  | 3,09 | 3,28 | 2,96 | 3,18 | 3,24 | 3,39 | 3,27 |
| 1835 | 4,84 | 4,75 | 4    | 4,53 | 5,46 | 5,01 | 4,89 | 5,12 | 4,96 | 4,89 | 4,78 | 4,88 |
| 1834 | 2,9  | 3,06 | 2,97 | 2,98 | 2,6  | 2,9  | 3,09 | 2,86 | 2,92 | 2,90 | 3,06 | 2,96 |
| 1833 | 4,18 | 4,84 | 4,3  | 4,44 | 4,78 | 4,01 | 4,52 | 4,44 | 4,06 | 4,07 | 4,10 | 4,08 |
| 1832 | 2,25 | 2,47 | 2,5  | 2,41 | 2,48 | 1,99 | 2,21 | 2,23 | 2,36 | 2,67 | 2,52 | 2,52 |



| 1831 | 3,05 | 3,07 | 3,01 | 3,04 | 4,11 | 3,73 | 3,92 | 3,92 | 3,68 | 3,79 | 3,64 | 3,70 |
| 1830 | 4,03 | 3,83 | 4,04 | 3,97 | 3,97 | 3,88 | 4,04 | 3,96 | 3,88 | 3,83 | 3,64 | 3,78 |
| 1829 | 3,08 | 2,85 | 2,93 | 2,95 | 2,93 | 3,38 | 2,96 | 3,09 | 2,60 | 2,83 | 2,75 | 2,73 |
| 1828 | 3,88 | 3,59 | 4,08 | 3,85 | 3,4 | 3,46 | 3,79 | 3,55 | 3,43 | 3,72 | 3,56 | 3,57 |
| 1827 | 4,89 | 4,72 | 4,89 | 4,83 | 4,42 | 4,84 | 4,88 | 4,71 | 4,96 | 4,80 | 4,68 | 4,81 |
| 1826 | 5,1 | 5,49 | 5,28 | 5,29 | 5,27 | 5,03 | 5,28 | 5,19 | 5,28 | 5,23 | 5,19 | 5,23 |
| 1825 | 4,52 | 4,46 | 4,65 | 4,54 | 3,99 | 4,05 | 3,99 | 4,01 | 4,00 | 3,99 | 4,03 | 4,01 |
| 1824 | 3,82 | 3,59 | | 3,71 | 3,74 | 3,66 | 3,62 | 3,67 | 3,29 | 3,09 | 3,14 | 3,17 |
| 1823 | 3,57 | 3,97 | 3,34 | 3,63 | 3,66 | 3,94 | 3,15 | 3,58 | 3,87 | 3,45 | 3,87 | 3,73 |
| 1822 | 3,12 | 3,59 | 3,79 | 3,50 | 3,48 | 3,77 | 3,9 | 3,72 | 3,50 | 3,72 | 3,60 | 3,61 |
| 1821 | 3,49 | 3,59 | 3,49 | 3,52 | 3,39 | 3,48 | 3,76 | 3,54 | 3,18 | 3,48 | 3,60 | 3,42 |
| 1820 | 4,63 | 4,53 | 4,5 | 4,55 | 4,55 | 4,82 | 4,58 | 4,65 | 4,89 | 4,61 | 4,49 | 4,66 |
| 1819 | 3,65 | 3,68 | 3,59 | 3,64 | 3,76 | 3,64 | 3,65 | 3,68 | 3,69 | 3,77 | 3,69 | 3,72 |
| 1818 | 2,33 | 2,47 | 2,37 | 2,39 | 3,3 | 3,2 | 3,09 | 3,20 | 2,73 | 3,06 | 3,10 | 2,96 |
| 1817 | 4,03 | 3,77 | 3,57 | 3,79 | 3,74 | 4,11 | 3,56 | 3,80 | 4,01 | 4,11 | 4,07 | 4,06 |
| 1816 | 3,61 | 3,96 | 3,57 | 3,71 | 4,52 | 4,09 | 4,14 | 4,25 | 4,36 | 4,41 | 4,38 | 4,38 |
| 1815 | 3,34 | 3,59 | 3,17 | 3,37 | 3,17 | 3,17 | 2,91 | 3,08 | 3,05 | 3,05 | 3,12 | 3,07 |
| 1814 | 4,41 | 3,97 | 4,42 | 4,27 | 4,03 | 4,16 | 3,99 | 4,06 | 4,39 | 4,35 | 4,21 | 4,32 |
| 1813 | 4,1 | 4,54 | 4,18 | 4,27 | 4,71 | 4,44 | 4,42 | 4,52 | 4,32 | 4,02 | 4,18 | 4,17 |
| 1812 | 3,8 | 3,59 | 3,67 | 3,69 | 3,84 | 3,95 | 3,81 | 3,87 | 4,08 | 4,48 | 4,28 | 4,28 |
| 1811 | 4,54 | 4,55 | 4,57 | 4,55 | 4,89 | 4,52 | 4,9 | 4,77 | 4,57 | 4,46 | 4,35 | 4,46 |
| 1810 | 3,85 | 3,3 | 3,47 | 3,54 | 3,16 | 3,27 | 3,26 | 3,23 | 3,63 | 3,55 | 3,62 | 3,60 |
| 1809 | 4,85 | 5,08 | 5,18 | 5,04 | 4,98 | 5,03 | 5,19 | 5,07 | 5,54 | 5,27 | 5,38 | 5,40 |
| 1808 | 4,33 | 4,34 | 4,36 | 4,34 | 4,12 | 4,16 | 4,26 | 4,18 | 3,68 | 3,90 | 3,72 | 3,77 |
| 1807 | 2,46 | 2,73 | 2,58 | 2,59 | 2,29 | 2,67 | 2,7 | 2,55 | 2,61 | 2,98 | 3,04 | 2,88 |
| 1806 | 4,62 | 5,03 | 4,64 | 4,76 | 5,24 | 4,47 | 4,96 | 4,89 | 4,51 | 4,63 | 4,54 | 4,56 |
| 1805 | 1,97 | 1,96 | 2,13 | 2,02 | 2,02 | 1,9 | 1,95 | 1,96 | 2,11 | 1,96 | 2,08 | 2,05 |
| 1804 | 4,55 | 4,77 | 4,45 | 4,59 | 4,89 | 4,84 | 4,87 | 4,87 | 5,14 | 5,36 | 5,17 | 5,22 |
| 1803 | 3,95 | 4,12 | 4,26 | 4,11 | 3,05 | 2,97 | 3,14 | 3,05 | 3,06 | 2,68 | 2,80 | 2,85 |
| 1802 | 1,8 | 1,68 | 1,76 | 1,75 | 1,64 | 1,53 | 1,54 | 1,57 | 1,72 | 1,58 | 1,47 | 1,59 |
| 1801 | 2,17 | 2,47 | 2,64 | 2,43 | 2,33 | 2,52 | 2,55 | 2,47 | 2,86 | 2,50 | 2,34 | 2,57 |
| 1800 | 2,37 | 2,46 | 2,28 | 2,37 | 2,29 | 2,25 | 2,31 | 2,28 | 2,67 | 2,71 | 2,80 | 2,73 |



| 1799 | 3,16 | 2,81 | 2,79 | 2,92 | 2,67 | 3,05 | 2,9  | 2,87 | 2,74 | 2,62 | 2,87 | 2,74 |
| 1798 | 3,38 | 3,09 | 3,23 | 3,23 | 4,21 | 3,98 | 4,11 | 4,10 | 4,26 | 4,10 | 4,50 | 4,29 |
| 1797 | 3,65 | 3,3  | 3,16 | 3,37 | 3,51 | 3,49 | 3,76 | 3,59 | 3,63 | 3,92 | 3,80 | 3,78 |
| 1796 | 2,37 | 3,04 | 3    | 2,80 | 2,3  | 2,3  | 2,31 | 2,30 | 1,97 | 2,09 | 2,00 | 2,02 |
| 1795 | 2,88 | 2,81 | 2,2  | 2,63 | 2,88 | 2,57 | 2,35 | 2,60 | 2,86 | 2,99 | 3,02 | 2,96 |
| 1794 | 3,8  | 3,88 | 3,43 | 3,70 | 3,54 | 3,71 | 3,08 | 3,44 | 3,44 | 3,25 | 3,30 | 3,33 |
| 1793 | 2,23 | 2,45 | 2,39 | 2,36 | 2,24 | 2,18 | 2,23 | 2,22 | 2,39 | 2,70 | 2,51 | 2,53 |
| 1792 | 4,44 | 4,37 | 4,75 | 4,52 | 4,26 | 4,31 | 4,16 | 4,24 | 4,52 | 4,30 | 4,26 | 4,36 |
| 1791 | 3,56 | 3,62 | 3,86 | 3,68 | 3,88 | 3,74 | 3,51 | 3,71 | 3,49 | 3,45 | 3,28 | 3,41 |
| 1790 | 3,45 | 3,4  | 3,31 | 3,39 | 3,06 | 3,01 | 3,07 | 3,05 | 2,86 | 2,76 | 2,60 | 2,74 |
| 1789 | 2,69 | 2,43 | 2,2  | 2,44 | 3,37 | 3,31 | 3,65 | 3,44 | 3,31 | 3,46 | 3,23 | 3,33 |
| 1788 | 2,34 | 2,43 | 2,51 | 2,43 | 2,23 | 2,23 | 2,61 | 2,36 | 1,97 | 1,86 | 1,98 | 1,94 |
| 1787 | 2,54 | 2,48 | 2,47 | 2,50 | 2,57 | 2,56 | 2,64 | 2,59 | 2,54 | 2,69 | 2,42 | 2,55 |
| 1786 | 3,36 | 3,73 | 3,2  | 3,43 | 3,65 | 4,04 | 3,53 | 3,74 | 4,12 | 3,88 | 3,82 | 3,94 |
| 1785 | 6,75 | 6,82 | 6,96 | 6,84 | 6,73 | 6,9  | 6,71 | 6,78 | 6,49 | 6,23 | 6,01 | 6,24 |
| 1784 | 3,16 | 3,13 | 3,19 | 3,16 | 3,26 | 3,04 | 3,01 | 3,10 | 2,73 | 2,52 | 2,64 | 2,63 |
| 1783 | 4,08 | 4,15 | 3,86 | 4,03 | 4,23 | 4,03 | 3,86 | 4,04 | 3,82 | 3,98 | 3,94 | 3,91 |
| 1782 | 3,23 | 3,39 | 3,22 | 3,28 | 3    | 3,11 | 3,28 | 3,13 | 3,29 | 3,55 | 3,56 | 3,47 |
| 1781 | 3,37 | 3,15 | 3,62 | 3,38 | 3,15 | 3,01 | 3,15 | 3,10 | 3,38 | 3,15 | 2,97 | 3,17 |
| 1780 | 2,86 | 2,82 | 2,53 | 2,74 | 3,08 | 2,99 | 3,03 | 3,03 | 3,37 | 3,38 | 3,28 | 3,34 |
| 1779 | 5,78 | 5,8  | 5,48 | 5,69 | 6,03 | 5,9  | 6,43 | 6,12 | 6,68 | 6,23 | 6,56 | 6,49 |
| 1778 | 4,28 | 4,13 | 4,06 | 4,16 | 3,8  | 3,96 | 4,1  | 3,95 | 3,57 | 3,50 | 3,55 | 3,54 |
| 1777 | 3,95 | 4,01 | 3,79 | 3,92 | 4,1  | 3,72 | 4,38 | 4,07 | 4,38 | 4,32 | 4,43 | 4,38 |
| 1776 | 4,27 | 4,59 | 4,55 | 4,47 | 4,32 | 4,9  | 4,76 | 4,66 | 4,08 | 4,34 | 4,27 | 4,23 |
| 1775 | 3,48 | 3,14 | 3,66 | 3,43 | 3,74 | 3,55 | 3,64 | 3,64 | 3,68 | 3,62 | 3,55 | 3,62 |
| 1774 | 2,78 | 2,73 | 2,72 | 2,74 | 3,42 | 3,36 | 3,41 | 3,40 | 3,38 | 3,49 | 3,62 | 3,50 |
| 1773 | 4,29 | 3,9  | 3,97 | 4,05 | 3,86 | 4    | 4,44 | 4,10 | 4,95 | 4,37 | 4,42 | 4,58 |
| 1772 | 4,13 | 4,47 | 4,16 | 4,25 | 4,29 | 4,24 | 4,25 | 4,26 | 4,43 | 4,99 | 4,75 | 4,72 |

| GL17-04C | | | | | | | | | |
|---|---|---|---|---|---|---|---|---|---|
| μCT measurements direction 1 | | | μCT measurements direction 2 | | | Thin section measurements | | | |



| Years | 1st Count | 2nd Count | 3rd Count | Mean | 1st Count | 2nd Count | 3rd Count | Mean | 1st Count | 2nd Count | 3rd Count | Mean |
|---|---|---|---|---|---|---|---|---|---|---|---|---|
| 1957 | 5,65 | 5,62 | 5,66 | 5,64 | 5,47 | 5,52 | 5,35 | 5,45 | 5,20 | 4,89 | 5,02 | 5,04 |
| 1956 | 4,09 | 4,07 | 4,07 | 4,08 | 3,88 | 4,31 | 4,2 | 4,13 | 4,01 | 4,06 | 4,51 | 4,19 |
| 1955 | 3,16 | 3,22 | 3,31 | 3,23 | 3,56 | 3,31 | 3,48 | 3,45 | 3,36 | 3,51 | 3,18 | 3,35 |
| 1954 | 4,6 | 4,53 | 4,75 | 4,63 | 5,03 | 4,85 | 4,9 | 4,93 | 4,70 | 4,25 | 4,51 | 4,49 |
| 1953 | 2,71 | 2,42 | 2,57 | 2,57 | 3,04 | 3,09 | 2,59 | 2,91 | 1,95 | 1,98 | 2,77 | 2,23 |
| 1952 | 3,03 | 2,94 | 3,21 | 3,06 | 3,46 | 2,77 | 3,12 | 3,12 | 4,12 | 4,21 | 3,79 | 4,04 |
| 1951 | 5,13 | 5,08 | 5,03 | 5,08 | 4,72 | 5,31 | 5,36 | 5,13 | 5,60 | 5,59 | 5,64 | 5,61 |
| 1950 | 4,12 | 4,12 | 4,15 | 4,13 | 3,89 | 4,32 | 3,66 | 3,96 | 4,89 | 4,77 | 4,61 | 4,76 |
| 1949 | 3,96 | 3,97 | 4,08 | 4,00 | 4,84 | 4,31 | 4,65 | 4,60 | 3,64 | 3,70 | 3,79 | 3,71 |
| 1948 | 3,74 | 3,51 | 3,66 | 3,64 | 3,39 | 3,55 | 3,48 | 3,47 | 3,10 | 3,11 | 3,32 | 3,18 |
| 1947 | 6,2 | 6,09 | 5,65 | 5,98 | 6,08 | 5,89 | 5,58 | 5,85 | 5,40 | 5,07 | 5,12 | 5,20 |
| 1946 | 3,13 | 3,27 | 3,38 | 3,26 | 3,44 | 3,63 | 3,41 | 3,49 | 3,65 | 3,65 | 3,18 | 3,49 |
| 1945 | 3,42 | 3,39 | 3,49 | 3,43 | 2,99 | 3,36 | 3,11 | 3,15 | 3,28 | 3,47 | 3,21 | 3,32 |
| 1944 | 5,66 | 5,48 | 5,53 | 5,56 | 5,25 | 5,43 | 5,43 | 5,37 | 5,67 | 5,35 | 5,36 | 5,46 |
| 1943 | 2,85 | 2,95 | 2,83 | 2,88 | 3,19 | 2,9 | 2,97 | 3,02 | 2,59 | 2,66 | 3,10 | 2,78 |
| 1942 | 3,65 | 3,76 | 3,63 | 3,68 | 3,91 | 3,71 | 3,73 | 3,78 | 4,19 | 3,98 | 4,23 | 4,13 |
| 1941 | 7,21 | 7,28 | 7,03 | 7,17 | 7,51 | 7,33 | 7,02 | 7,29 | 7,21 | 7,63 | 7,28 | 7,37 |
| 1940 | 4,05 | 4,02 | 4,37 | 4,15 | 4,53 | 4,26 | 3,91 | 4,23 | 4,21 | 4,10 | 4,16 | 4,16 |
| 1939 | 3,43 | 3,37 | 3,23 | 3,34 | 3,2 | 3,08 | 3,4 | 3,23 | 3,86 | 4,12 | 3,86 | 3,95 |
| 1938 | 3,59 | 3,74 | 3,62 | 3,65 | 3,71 | 3,89 | 3,28 | 3,63 | 3,23 | 3,09 | 3,21 | 3,18 |
| 1937 | 3,95 | 3,89 | 3,61 | 3,82 | 3,92 | 3,35 | 3,82 | 3,70 | 3,66 | 3,91 | 3,95 | 3,84 |
| 1936 | 5,34 | 5,45 | 5,37 | 5,39 | 5,89 | 5,34 | 5,06 | 5,43 | 5,77 | 5,78 | 5,84 | 5,80 |
| 1935 | 2,14 | 1,99 | 1,83 | 1,99 | 2,2 | 1,87 | 1,91 | 1,99 | 1,73 | 1,99 | 2,08 | 1,93 |
| 1934 | 3,6 | 3,69 | 3,97 | 3,75 | 2,265 | 2,7 | 3,9 | 2,96 | 2,68 | 2,41 | 2,56 | 2,55 |
| 1933 | 3,49 | 3,34 | 3,06 | 3,30 | 2,88 | 2,75 | 2,78 | 2,80 | 3,28 | 3,22 | 3,18 | 3,23 |
| 1932 | 4,18 | 4,94 | 4,37 | 4,50 | 4,23 | 4,24 | 4,72 | 4,40 | 4,13 | 4,28 | 4,10 | 4,17 |
| 1931 | 3,26 | 3,18 | 3,14 | 3,19 | 2,78 | 3,66 | 3,08 | 3,17 | 4,24 | 4,24 | 4,21 | 4,23 |
| 1930 | 6,22 | 6,32 | 6,69 | 6,41 | 6,7 | 6,42 | 6,25 | 6,46 | 5,83 | 6,37 | 6,26 | 6,15 |
| 1929 | 4,85 | 4,49 | 4,96 | 4,77 | 5,38 | 5,34 | 4,71 | 5,14 | 4,08 | 4,05 | 4,09 | 4,07 |
| 1928 | 3,45 | 3,45 | 3,47 | 3,46 | 3,19 | 3,54 | 3,55 | 3,43 | 3,98 | 4,20 | 4,09 | 4,09 |





| 1927 | 6,28 | 6,46 | 6,11 | 6,28 | 5,6  | 6,6  | 6,33 | 6,18 | 6,30 | 6,27 | 6,18 | 6,25 |
| 1926 | 3,95 | 3,77 | 3,81 | 3,84 | 3,72 | 3,1  | 4,06 | 3,63 | 4,30 | 3,74 | 3,84 | 3,96 |
| 1925 | 5,37 | 5,37 | 5,17 | 5,30 | 5,41 | 5,71 | 5,13 | 5,42 | 4,08 | 3,86 | 3,79 | 3,91 |
| 1924 | 8,16 | 7,75 | 7,71 | 7,87 | 7,91 | 7,92 | 8,28 | 8,04 | 8,03 | 8,21 | 8,04 | 8,10 |
| 1923 | 5,03 | 5,02 | 4,53 | 4,86 | 5,12 | 5,24 | 5,14 | 5,17 | 5,19 | 4,83 | 4,71 | 4,91 |
| 1922 | 4,34 | 4,39 | 4,9  | 4,54 | 5,01 | 5,56 | 5,03 | 5,20 | 5,20 | 5,51 | 5,15 | 5,29 |
| 1921 | 5,05 | 5    | 5,54 | 5,20 | 4,4  | 4,3  | 4,1  | 4,27 | 3,65 | 3,39 | 3,72 | 3,59 |
| 1920 | 6,52 | 6,45 | 6,6  | 6,52 | 6,44 | 6,63 | 6,88 | 6,65 | 7,63 | 7,87 | 7,31 | 7,61 |
| 1919 | 3,98 | 4,24 | 4,5  | 4,24 | 4,23 | 4,31 | 4,37 | 4,30 | 3,33 | 3,40 | 3,47 | 3,40 |
| 1918 | 5,45 | 5,51 | 5,24 | 5,40 | 5,44 | 5,42 | 5,12 | 5,33 | 5,12 | 5,27 | 5,37 | 5,25 |
| 1917 | 2,97 | 2,66 | 1,72 | 2,45 | 1,63 | 1,66 | 1,63 | 1,64 | 1,85 | 1,96 | 1,87 | 1,89 |
| 1916 | 3,12 | 2,89 | 1,79 | 2,60 | 1,73 | 1,78 | 1,65 | 1,72 | 1,97 | 1,91 | 1,79 | 1,89 |
| 1915 | 3,78 | 3,91 | 3,62 | 3,77 | 3,07 | 3,4  | 3,62 | 3,36 | 3,81 | 3,64 | 3,54 | 3,66 |
| 1914 | 2,76 | 3,83 | 3,33 | 3,31 | 3,41 | 3,05 | 3,42 | 3,29 | 2,86 | 3,10 | 3,05 | 3,00 |
| 1913 | 2,8  | 3,13 | 3,79 | 3,24 | 3,35 | 3,47 | 3,43 | 3,42 | 3,56 | 3,66 | 3,67 | 3,63 |
| 1912 | 4,46 | 4,02 | 4,13 | 4,20 | 4,66 | 4,69 | 4,48 | 4,61 | 4,08 | 3,86 | 3,97 | 3,97 |
| 1911 | 4,01 | 4,69 | 4,39 | 4,36 | 4    | 4,11 | 4,25 | 4,12 | 4,46 | 4,48 | 4,51 | 4,48 |
| 1910 | 3,38 | 3,29 | 3,27 | 3,31 | 3,01 | 3,02 | 2,95 | 2,99 | 2,89 | 3,19 | 3,04 | 3,04 |
| 1909 | 3,93 | 3,56 | 3,94 | 3,81 | 3,86 | 3,86 | 4,19 | 3,97 | 3,77 | 3,69 | 3,84 | 3,77 |
| 1908 | 4,75 | 4,42 | 5,05 | 4,74 | 4,92 | 5,09 | 4,7  | 4,90 | 4,92 | 4,83 | 4,91 | 4,89 |
| 1907 | 3,68 | 3,6  | 3,44 | 3,57 | 3,38 | 3,31 | 3,58 | 3,42 | 2,82 | 3,01 | 2,93 | 2,92 |
| 1906 | 3,44 | 3,69 | 3,6  | 3,58 | 3,37 | 3,54 | 3,65 | 3,52 | 3,76 | 4,79 | 3,74 | 4,09 |
| 1905 | 4,4  | 4,25 | 4,42 | 4,36 | 4,48 | 4,49 | 4,15 | 4,37 | 4,43 | 3,97 | 4,13 | 4,18 |
| 1904 | 4,32 | 4,42 | 4,17 | 4,30 | 4,21 | 4,18 | 4,39 | 4,26 | 4,50 | 4,54 | 4,33 | 4,46 |
| 1903 | 3,14 | 2,78 | 3,11 | 3,01 | 2,82 | 2,9  | 3,02 | 2,91 | 2,82 | 2,16 | 1,92 | 2,30 |
| 1902 | 2,43 | 2,49 | 2,34 | 2,42 | 2,47 | 2,56 | 2,45 | 2,49 | 2,38 | 2,84 | 2,29 | 2,50 |
| 1901 | 3,49 | 3,16 | 2,87 | 3,17 | 2,96 | 2,96 | 3,07 | 3,00 | 3,02 | 3,05 | 3,07 | 3,04 |
| 1900 | 3,66 | 3,47 | 3,26 | 3,46 | 3,38 | 3,54 | 3,27 | 3,40 | 3,44 | 3,13 | 3,32 | 3,30 |
| 1899 | 4    | 4    | 4,01 | 4,00 | 4,01 | 4,01 | 3,86 | 3,96 | 3,45 | 3,57 | 3,69 | 3,57 |
| 1898 | 2,53 | 2,57 | 2,48 | 2,53 | 2,35 | 2,54 | 2,34 | 2,41 | 2,33 | 1,99 | 2,23 | 2,18 |
| 1897 | 3,93 | 4,03 | 3,99 | 3,98 | 4,07 | 4,07 | 4,09 | 4,08 | 4,09 | 4,24 | 4,10 | 4,14 |
| 1896 | 3,56 | 3,3  | 3,41 | 3,42 | 3,68 | 3,74 | 3,39 | 3,60 | 3,33 | 3,29 | 3,49 | 3,37 |



| | | | | | | | | | | | |
|---|---|---|---|---|---|---|---|---|---|---|---|
| 1895 | 4,84 | 4,69 | 4,7 | 4,74 | 4,72 | 4,89 | 4,71 | 4,77 | 4,96 | 4,78 | 4,84 | 4,86 |
| 1894 | 4,04 | 4,15 | 4,49 | 4,23 | 3,91 | 3,59 | 3,97 | 3,82 | 3,77 | 4,05 | 4,05 | 3,96 |
| 1893 | 4,86 | 4,87 | 5,24 | 4,99 | 4,53 | 4,94 | 4,51 | 4,66 | 4,97 | 4,54 | 4,79 | 4,77 |
| 1892 | 4,33 | 4,2 | 4,26 | 4,26 | 4,36 | 4,34 | 4,52 | 4,41 | 3,60 | 3,67 | 3,51 | 3,59 |
| 1891 | 3,53 | 4,06 | 3,42 | 3,67 | 3,79 | 3,52 | 3,53 | 3,61 | 3,98 | 3,61 | 3,71 | 3,77 |
| 1890 | 4,83 | 3,99 | 4,48 | 4,43 | 4,31 | 4,58 | 4,24 | 4,38 | 3,86 | 3,98 | 4,10 | 3,98 |
| 1889 | 5,18 | 5,05 | 5,38 | 5,20 | 5,25 | 5,25 | 5,29 | 5,26 | 4,92 | 5,22 | 5,24 | 5,13 |
| 1888 | 3,51 | 3,41 | 3,14 | 3,35 | 3,24 | 3,25 | 3,49 | 3,33 | 3,19 | 3,23 | 3,02 | 3,14 |
| 1887 | 3,98 | 3,81 | 4,08 | 3,96 | 3,93 | 3,86 | 4,04 | 3,94 | 3,64 | 3,73 | 3,66 | 3,68 |
| 1886 | 3,56 | 3,67 | 3,81 | 3,68 | 3,67 | 3,67 | 3,66 | 3,67 | 4,67 | 4,54 | 4,20 | 4,47 |
| 1885 | 4,54 | 4,73 | 4,4 | 4,56 | 5,03 | 4,86 | 4,81 | 4,90 | 4,02 | 4,19 | 3,94 | 4,05 |
| 1884 | 5,17 | 4,89 | 4,78 | 4,95 | 4,79 | 4,76 | 4,74 | 4,76 | 5,04 | 4,72 | 4,98 | 4,91 |
| 1883 | 4,85 | 5,45 | 4,98 | 5,09 | 4,97 | 5,34 | 5,02 | 5,11 | 5,24 | 5,17 | 4,98 | 5,13 |
| 1882 | 3,33 | 3,18 | 3,67 | 3,39 | 3,43 | 3,56 | 3,53 | 3,51 | 2,97 | 3,27 | 2,89 | 3,04 |
| 1881 | 2,66 | 2,85 | 2,69 | 2,73 | 2,68 | 2,91 | 2,61 | 2,73 | 2,65 | 2,65 | 2,71 | 2,67 |
| 1880 | 4,2 | 4,07 | 4,16 | 4,14 | 4,26 | 4,22 | 3,86 | 4,11 | 4,24 | 3,95 | 4,37 | 4,19 |
| 1879 | 5,18 | 4,67 | 4,95 | 4,93 | 4,8 | 5,2 | 5,36 | 5,12 | 4,76 | 4,88 | 4,87 | 4,84 |
| 1878 | 2,9 | 3,21 | 3,42 | 3,18 | 3,37 | 3,19 | 3,54 | 3,37 | 4,21 | 4,02 | 4,02 | 4,08 |
| 1877 | 4,75 | 4,92 | 4,8 | 4,82 | 5,11 | 5,04 | 5,08 | 5,08 | 4,61 | 4,61 | 4,58 | 4,60 |
| 1876 | 3,49 | 2,9 | 2,56 | 2,98 | 3,02 | 2,93 | 2,74 | 2,90 | 3,28 | 3,29 | 3,28 | 3,28 |
| 1875 | 2,46 | 2,9 | 2,84 | 2,73 | 2,77 | 3,22 | 2,51 | 2,83 | 2,60 | 2,59 | 2,68 | 2,63 |
| 1874 | 4,03 | 3,62 | 3,88 | 3,84 | 3,75 | 3,76 | 3,79 | 3,77 | 4,13 | 3,68 | 3,77 | 3,86 |
| 1873 | 4,15 | 4,08 | 3,95 | 4,06 | 4,2 | 3,92 | 4,18 | 4,10 | 3,85 | 4,18 | 3,90 | 3,98 |
| 1872 | 3,99 | 3,82 | 4,1 | 3,97 | 3,93 | 4,14 | 4,12 | 4,06 | 4,13 | 4,08 | 4,32 | 4,17 |
| 1871 | 2,63 | 2,88 | 2,53 | 2,68 | 2,58 | 2,8 | 2,89 | 2,76 | 2,75 | 2,79 | 2,68 | 2,74 |
| 1870 | 3,57 | 3,69 | 3,66 | 3,64 | 3,73 | 3,67 | 3,28 | 3,56 | 3,56 | 3,32 | 3,35 | 3,41 |
| 1869 | 3,72 | 3,72 | 3,46 | 3,63 | 3,67 | 3,58 | 3,6 | 3,62 | 3,49 | 3,50 | 3,61 | 3,54 |
| 1868 | 3,16 | 2,91 | 3,39 | 3,15 | 3,16 | 3,32 | 3,18 | 3,22 | 3,24 | 3,12 | 3,44 | 3,27 |
| 1867 | 2,65 | 2,47 | 2,41 | 2,51 | 2,08 | 2,57 | 2,43 | 2,36 | 3,33 | 3,57 | 3,10 | 3,33 |
| 1866 | 3,72 | 4,1 | 3,78 | 3,87 | 3,95 | 3,7 | 3,65 | 3,77 | 3,07 | 2,85 | 3,03 | 2,98 |
| 1865 | 3,31 | 3 | 2,96 | 3,09 | 2,94 | 3,05 | 2,96 | 2,98 | 3,61 | 3,94 | 3,62 | 3,72 |
| 1864 | 3,06 | 2,83 | 2,98 | 2,96 | 3,29 | 2,94 | 3,03 | 3,09 | 2,75 | 2,86 | 2,95 | 2,85 |





| | | | | | | | | | | | | |
|---|---|---|---|---|---|---|---|---|---|---|---|---|
| 1863 | 3,47 | 3,84 | 3,66 | 3,66 | 3,43 | 3,69 | 3,69 | 3,60 | 3,72 | 3,56 | 3,65 | 3,64 |
| 1862 | 3,59 | 4,25 | 4,16 | 4,00 | 4,45 | 4,26 | 4,62 | 4,44 | 4,66 | 4,64 | 4,31 | 4,53 |
| 1861 | 4,98 | 5,18 | 5,44 | 5,20 | 5,93 | 6,07 | 5,72 | 5,91 | 4,54 | 4,51 | 5,52 | 4,86 |
| 1860 | 2,42 | 2,25 | 2,82 | 2,50 | 3 | 2,85 | 2,74 | 2,86 | 3,29 | 3,24 | 2,77 | 3,10 |
| 1859 | 3,22 | 4,02 | 3,39 | 3,54 | 3,79 | 3,24 | 3,32 | 3,45 | 3,55 | 3,65 | 3,39 | 3,53 |
| 1858 | 3,97 | 4,87 | 4,72 | 4,52 | 4,29 | 4,37 | 4,3 | 4,32 | 4,14 | 3,98 | 4,33 | 4,15 |
| 1857 | 7,87 | 8,12 | 8,15 | 8,05 | 7,86 | 8,46 | 8,08 | 8,13 | 7,37 | 7,24 | 7,54 | 7,38 |
| 1856 | 3,27 | 3,04 | 3,19 | 3,17 | 3,37 | 2,94 | 3,23 | 3,18 | 2,94 | 2,83 | 2,90 | 2,89 |
| 1855 | 3,9 | 3,74 | 3,6 | 3,75 | 3,66 | 3,79 | 3,91 | 3,79 | 3,66 | 3,87 | 3,97 | 3,84 |
| 1854 | 4,15 | 4,08 | 4,07 | 4,10 | 3,95 | 4,17 | 4,21 | 4,11 | 4,34 | 3,94 | 3,97 | 4,08 |
| 1853 | 3,58 | 3,37 | 3,65 | 3,53 | 3,66 | 3,6 | 3,62 | 3,63 | 3,24 | 3,47 | 3,57 | 3,42 |
| 1852 | 4,83 | 4,83 | 4,65 | 4,77 | 4,74 | 4,74 | 4,6 | 4,69 | 5,13 | 4,99 | 5,11 | 5,08 |
| 1851 | 3,23 | 3,61 | 3,35 | 3,40 | 3,66 | 3,86 | 3,69 | 3,74 | 4,51 | 3,82 | 3,86 | 4,06 |
| 1850 | 3,26 | 3,06 | 3,04 | 3,12 | 3,23 | 3,45 | 3,39 | 3,36 | 2,86 | 3,11 | 3,01 | 2,99 |
| 1849 | 4,85 | 5,21 | 4,65 | 4,90 | 4,96 | 4,98 | 4,81 | 4,92 | 5,57 | 5,08 | 5,02 | 5,22 |
| 1848 | 3,07 | 3,03 | 3,08 | 3,06 | 3,28 | 3,25 | 3,45 | 3,33 | 2,75 | 3,40 | 3,29 | 3,15 |
| 1847 | 2,62 | 2,41 | 2,41 | 2,48 | 2,4 | 2,57 | 2,65 | 2,54 | 2,38 | 2,28 | 2,58 | 2,41 |
| 1846 | 2,67 | 3,11 | 2,88 | 2,89 | 3,11 | 2,86 | 3,15 | 3,04 | 2,76 | 2,64 | 2,86 | 2,75 |
| 1845 | 3,3 | 3,29 | 3,24 | 3,28 | 3,24 | 3,63 | 3,07 | 3,31 | 3,81 | 4,42 | 4,15 | 4,13 |
| 1844 | 4,6 | 4,83 | 4,78 | 4,74 | 4,9 | 4,6 | 4,87 | 4,79 | 3,85 | 3,87 | 3,72 | 3,82 |
| 1843 | 2,95 | 2,84 | 3,31 | 3,03 | 2,95 | 3,33 | 3,15 | 3,14 | 3,44 | 2,92 | 3,59 | 3,32 |
| 1842 | 3,09 | 3,19 | 3,06 | 3,11 | 3,4 | 3,12 | 3,42 | 3,31 | 3,02 | 3,11 | 2,96 | 3,03 |
| 1841 | 2,28 | 2,66 | 2,44 | 2,46 | 2,39 | 2,23 | 2,06 | 2,23 | 2,75 | 2,66 | 2,50 | 2,64 |
| 1840 | 2,84 | 2,72 | 2,88 | 2,81 | 2,69 | 2,52 | 2,57 | 2,59 | 3,02 | 2,74 | 3,01 | 2,92 |
| 1839 | 3,2 | 3,22 | 3,39 | 3,27 | 3,48 | 3,38 | 3,33 | 3,40 | 3,40 | 3,14 | 3,49 | 3,34 |
| 1838 | 2,53 | 2,42 | 2,23 | 2,39 | 2,1 | 2,51 | 2,89 | 2,50 | 2,43 | 2,34 | 2,39 | 2,39 |
| 1837 | 3,02 | 2,98 | 3,05 | 3,02 | 2,74 | 2,79 | 2,67 | 2,73 | 3,02 | 2,96 | 3,02 | 3,00 |
| 1836 | 3,64 | 3,77 | 3,8 | 3,74 | 3,39 | 3,41 | 3,53 | 3,44 | 4,55 | 4,67 | 4,75 | 4,66 |

| GL1705B | | |
|---|---|---|
| µCT measurements direction 1 | µCT measurements direction 2 | Thin section measurements |



| Years | 1st Count | 2nd Count | 3rd Count | Mean | 1st Count | 2nd Count | 3rd Count | Mean | 1st Count | 2nd Count | 3rd Count | Mean |
|---|---|---|---|---|---|---|---|---|---|---|---|---|
| 2016 | 2,42 | 2,45 | 1,91 | 2,26 | 2,03 | 2,26 | 1,95 | 2,08 | 2,03 | 2,06 | 2,02 | 2,04 |
| 2015 | 2,27 | 2,18 | 2,05 | 2,17 | 2,13 | 2,31 | 2,30 | 2,25 | 2,12 | 2,10 | 2,14 | 2,12 |
| 2014 | 2,42 | 2,51 | 2,38 | 2,44 | 2,29 | 2,31 | 2,38 | 2,33 | 2,02 | 2,35 | 2,23 | 2,20 |
| 2013 | 2,36 | 1,87 | 2,06 | 2,10 | 2,85 | 2,92 | 1,70 | 2,49 | 2,12 | 2,15 | 2,15 | 2,14 |
| 2012 | 2,9 | 3,31 | 2,41 | 2,87 | 2,92 | 2,69 | 2,39 | 2,67 | 2,02 | 2,01 | 1,96 | 2,00 |
| 2011 | 2,12 | 2,19 | 2,35 | 2,22 | 1,83 | 2,42 | 1,76 | 2,00 | 2,33 | 2,22 | 2,36 | 2,30 |
| 2010 | 1,66 | 2,19 | 2,1 | 1,98 | 1,7 | 1,92 | 1,76 | 1,79 | 1,91 | 2,04 | 1,97 | 1,97 |
| 2009 | 1,93 | 2,12 | 1,97 | 2,01 | 1,98 | 1,52 | 1,61 | 1,70 | 2,10 | 1,89 | 1,98 | 1,99 |
| 2008 | 2,37 | 3,07 | 2,29 | 2,58 | 2,9 | 2,49 | 2,20 | 2,53 | 2,43 | 2,41 | 2,58 | 2,47 |
| 2007 | 2,18 | 2,51 | 1,97 | 2,22 | 2,12 | 1,9 | 2,19 | 2,07 | 1,67 | 1,85 | 1,82 | 1,78 |
| 2006 | 2,38 | 2,45 | 2,22 | 2,35 | 2,9 | 2,29 | 2,33 | 2,51 | 2,22 | 2,14 | 2,22 | 2,19 |
| 2005 | 1,7 | 1,67 | 1,4 | 1,59 | 1,94 | 1,65 | 1,64 | 1,74 | 1,27 | 1,49 | 1,39 | 1,38 |
| 2004 | 1,98 | 2,17 | 1,78 | 1,98 | 2,13 | 2,22 | 1,79 | 2,05 | 2,19 | 2,13 | 2,16 | 2,16 |
| 2003 | 2,7 | 2,69 | 2,29 | 2,56 | 2,64 | 2,55 | 2,90 | 2,70 | 2,63 | 2,49 | 2,50 | 2,54 |
| 2002 | 2,32 | 2,3 | 2,41 | 2,34 | 2,71 | 2,71 | 2,55 | 2,66 | 2,48 | 2,39 | 2,45 | 2,44 |
| 2001 | 3,28 | 3,93 | 2,39 | 3,20 | 3,56 | 3,65 | 3,60 | 3,60 | 3,43 | 3,52 | 3,44 | 3,46 |
| 2000 | 2,48 | 2,08 | 2,62 | 2,39 | 2,29 | 2,27 | 2,44 | 2,33 | 2,81 | 2,72 | 2,84 | 2,79 |
| 1999 | 4,74 | 4,71 | 4,55 | 4,67 | 4,75 | 4,56 | 4,47 | 4,59 | 4,22 | 4,10 | 4,24 | 4,19 |
| 1998 | 4 | 4,02 | 4,01 | 4,01 | 3,99 | 4,18 | 4,21 | 4,13 | 3,15 | 3,68 | 3,50 | 3,44 |
| 1997 | 4,11 | 5,1 | 3,63 | 4,28 | 4,06 | 4,29 | 4,13 | 4,16 | 3,74 | 3,63 | 3,73 | 3,70 |
| 1996 | 4,75 | 5,94 | 4,39 | 5,03 | 4,32 | 5,22 | 4,28 | 4,61 | 4,11 | 4,17 | 4,11 | 4,13 |
| 1995 | 2,87 | 4,86 | 2,55 | 3,43 | 2,82 | 2,5 | 3,29 | 2,87 | 2,56 | 2,49 | 2,48 | 2,51 |
| 1994 | 2,99 | 3,55 | 2,71 | 3,08 | 3,01 | 3,03 | 3,23 | 3,09 | 2,90 | 2,87 | 3,01 | 2,93 |
| 1993 | 3,84 | 3,06 | 2,72 | 3,21 | 2,91 | 3,23 | 3,81 | 3,32 | 2,99 | 2,76 | 2,83 | 2,86 |
| 1992 | 3,54 | 3,55 | 3,81 | 3,63 | 3,87 | 3,65 | 4,01 | 3,84 | 3,61 | 3,84 | 3,77 | 3,74 |
| 1991 | 3,33 | 2,53 | 2,24 | 2,70 | 2,43 | 1,92 | 2,08 | 2,14 | 2,76 | 2,70 | 2,74 | 2,73 |
| 1990 | 3,36 | 3,09 | 3,24 | 3,23 | 2,5 | 2,48 | 2,23 | 2,40 | 3,06 | 2,98 | 3,13 | 3,06 |
| 1989 | 2,82 | 2,35 | 3,78 | 2,98 | 3,26 | 3,39 | 3,35 | 3,33 | 3,08 | 3,16 | 3,09 | 3,11 |
| 1988 | 2,35 | 1,89 | 2,37 | 2,20 | 2,44 | 2,48 | 2,82 | 2,58 | 2,88 | 2,68 | 2,79 | 2,78 |
| 1987 | 2,62 | 2,36 | 2,13 | 2,37 | 2,12 | 2,15 | 2,00 | 2,09 | 1,94 | 1,78 | 1,80 | 1,84 |





| | | | | | | | | | | | | |
|------|-------|-------|-------|-------|-------|-------|-------|-------|-------|-------|-------|-------|
| 1986 | 2,89 | 2,66 | 2,54 | 2,70 | 2,78 | 2,8 | 2,91 | 2,83 | 2,71 | 2,69 | 2,74 | 2,71 |
| 1985 | 2,96 | 4,01 | 3,36 | 3,44 | 3,77 | 3,53 | 4,40 | 3,90 | 3,84 | 3,77 | 3,81 | 3,81 |
| 1984 | 3,08 | 2,98 | 2,81 | 2,96 | 3,05 | 3,23 | 2,48 | 2,92 | 3,01 | 3,05 | 3,20 | 3,09 |
| 1983 | 4,84 | 4,76 | 4,91 | 4,84 | 4,75 | 5,06 | 5,20 | 5,00 | 4,77 | 4,85 | 4,80 | 4,81 |
| 1982 | 2,79 | 3,09 | 2,92 | 2,93 | 3,07 | 3,1 | | 3,09 | 2,73 | 2,77 | 2,56 | 2,69 |
| 1981 | 3,66 | 3,58 | 3,56 | 3,60 | 3,51 | 3,76 | 3,41 | 3,56 | 3,52 | 3,47 | 3,69 | 3,56 |
| 1980 | 3,49 | 3,49 | 3,38 | 3,45 | 3,49 | 4,66 | 3,98 | 4,04 | 4,38 | 3,77 | 3,63 | 3,93 |
| 1979 | 4,98 | 4,57 | 4,55 | 4,70 | 5,12 | 4,07 | 4,77 | 4,65 | 3,82 | 4,40 | 4,39 | 4,20 |
| 1978 | 4,69 | 4,75 | 4,64 | 4,69 | 3,93 | 3,73 | 4,60 | 4,09 | 4,76 | 4,57 | 4,47 | 4,60 |
| 1977 | 4,69 | 4,63 | 4,72 | 4,68 | 4,21 | 5,08 | 5,24 | 4,84 | 4,27 | 4,24 | 4,24 | 4,25 |
| 1976 | 4,35 | 4,56 | 4,72 | 4,54 | 4,24 | 4,06 | 3,92 | 4,07 | 4,50 | 4,75 | 4,58 | 4,61 |
| 1975 | 5,03 | 5,21 | 5,39 | 5,21 | 4,37 | 5,04 | 5,93 | 5,11 | 5,81 | 5,70 | 5,71 | 5,74 |
| 1974 | 4,09 | 4 | 3,93 | 4,01 | 3,2 | 4,06 | 4,41 | 3,89 | 4,17 | 4,23 | 4,21 | 4,20 |
| 1973 | 4,26 | 4,21 | 4,27 | 4,25 | 4,62 | 4,97 | 5,35 | 4,98 | 3,94 | 4,36 | 4,30 | 4,20 |
| 1972 | 11,6 | 11,03 | 11,25 | 11,29 | 10,18 | 11,48 | 11,37 | 11,01 | 10,80 | 10,63 | 10,53 | 10,65 |
| 1971 | 6,02 | 5,56 | 4,59 | 5,39 | 4,53 | 6,45 | 4,62 | 5,20 | 5,14 | 5,21 | 5,18 | 5,18 |
| 1970 | 4,92 | 4,09 | 3,94 | 4,32 | 4 | 4,12 | 3,83 | 3,98 | 4,23 | 4,09 | 4,12 | 4,15 |
| 1969 | 5,1 | 4,65 | 5,13 | 4,96 | 5,35 | 5,22 | 4,84 | 5,14 | 5,04 | 5,10 | 4,98 | 5,04 |
| 1968 | 4,3 | 5,25 | 5,21 | 4,92 | 5,72 | 4,65 | 4,49 | 4,95 | 5,09 | 5,14 | 5,22 | 5,15 |
| GAP | | | | | | | | | | | | |
| 1835 | 4,17 | 4,18 | 4,55 | 4,30 | 4,55 | 4,21 | 4,38 | 4,38 | 4,29 | 4,36 | 4,36 | 4,34 |
| 1834 | 3,01 | 3,2 | 2,63 | 2,95 | 3,2 | 3,23 | 3,2 | 3,21 | 2,43 | 2,36 | 2,34 | 2,38 |
| 1833 | 4,42 | 4,05 | 3,87 | 4,11 | 4,47 | 3,96 | 4,22 | 4,22 | 3,75 | 3,99 | 3,82 | 3,85 |
| 1832 | 2,88 | 2,54 | 2,34 | 2,59 | 1,95 | 2,55 | 2,27 | 2,26 | 2,11 | 2,19 | 2,29 | 2,20 |
| 1831 | 3,12 | 3,05 | 2,79 | 2,99 | 3,7 | 3,41 | 3,57 | 3,56 | 3,08 | 3,14 | 3,07 | 3,10 |
| 1830 | 3,59 | 3,15 | 3,24 | 3,33 | 3,61 | 3,52 | 3,6 | 3,58 | 3,27 | 3,36 | 3,20 | 3,28 |
| 1829 | 2,78 | 3,39 | 3,54 | 3,24 | 2,85 | 2,99 | 2,92 | 2,92 | 2,69 | 2,84 | 2,80 | 2,77 |
| 1828 | 3,61 | 3,05 | 3,21 | 3,29 | 3,9 | 3,71 | 3,81 | 3,81 | 3,21 | 3,45 | 3,29 | 3,31 |
| 1827 | 4,15 | 3,75 | 3,79 | 3,90 | 4,64 | 4,15 | 4,41 | 4,40 | 4,23 | 3,99 | 4,12 | 4,11 |
| 1826 | 3,66 | 4,21 | 3,25 | 3,71 | 4,94 | 3,61 | 4,29 | 4,28 | 4,41 | 4,42 | 4,46 | 4,43 |
| 1825 | 5,16 | 4,87 | 5,41 | 5,15 | 4,35 | 5,88 | 5,2 | 5,14 | 3,96 | 3,96 | 4,02 | 3,98 |
| 1824 | 3,3 | 2,92 | 2,73 | 2,98 | 3,14 | 3,22 | 3,19 | 3,18 | 2,68 | 3,31 | 3,01 | 3,00 |





| 1823 | 2,62 | 2,62 | 2,71 | 2,65 | 2,49 | 2,49 | 2,49 | 2,49 | 2,92 | 2,91 | 2,96 | 2,93 |
| 1822 | 3,18 | 3,33 | 3,15 | 3,22 | 3,37 | 3,4 | 3,39 | 3,39 | 2,83 | 2,81 | 2,91 | 2,85 |
| 1821 | 4,16 | 3,95 | 4,08 | 4,06 | 3,92 | 3,95 | 3,97 | 3,95 | 3,88 | 3,72 | 3,80 | 3,80 |
| 1820 | 3,3 | 3,26 | 2,9 | 3,15 | 3,14 | 3,45 | 3,3 | 3,30 | 3,63 | 3,76 | 3,54 | 3,64 |
| 1819 | 3,55 | 3,6 | 3,08 | 3,41 | 2,93 | 3,25 | 3,09 | 3,09 | 3,00 | 3,08 | 3,12 | 3,07 |
| 1818 | 2,95 | 2,81 | 2,99 | 2,92 | 2,94 | 2,56 | 2,75 | 2,75 | 2,92 | 2,86 | 2,95 | 2,91 |
| 1817 | 3,1 | 3,32 | 3,21 | 3,21 | 3,23 | 3,25 | 3,24 | 3,24 | 3,19 | 3,40 | 3,16 | 3,25 |
| 1816 | 2,95 | 3,41 | 3,74 | 3,37 | 3,92 | 4,12 | 4,02 | 4,02 | 2,81 | 2,47 | 2,79 | 2,69 |
| 1815 | 3,1 | 2,5 | 2,29 | 2,63 | 2,46 | 2,12 | 2,29 | 2,29 | 2,67 | 3,02 | 3,05 | 2,91 |
| 1814 | 4,57 | 4,15 | 3,94 | 4,22 | 3,92 | 3,96 | 3,94 | 3,94 | 4,33 | 3,99 | 3,95 | 4,09 |
| 1813 | 3,67 | 3,23 | 3,57 | 3,49 | 3,3 | 3,27 | 3,31 | 3,29 | 2,86 | 2,98 | 2,52 | 2,79 |
| 1812 | 1,84 | 2,22 | 3,69 | 2,58 | 2,39 | 2,32 | 2,37 | 2,36 | 1,42 | 1,45 | 1,50 | 1,46 |
| 1811 | 1,74 | 2,07 | 3,96 | 2,59 | 2,07 | 1,99 | 2,03 | 2,03 | 2,86 | 2,87 | 2,86 | 2,86 |
| 1810 | 4,42 | 4,49 | 4,33 | 4,41 | 4,76 | 4,53 | 4,66 | 4,65 | 3,88 | 3,85 | 3,91 | 3,88 |
| 1809 | 3,11 | 3,05 | 3,07 | 3,08 | 3,23 | 2,94 | 3,1 | 3,09 | 2,99 | 3,14 | 3,01 | 3,04 |
| 1808 | 3,62 | 3,81 | 4,54 | 3,99 | 4,08 | 4,14 | 4,11 | 4,11 | 3,78 | 3,85 | 3,99 | 3,87 |
| 1807 | 4,31 | 4,18 | 4,23 | 4,24 | 4,53 | 4,24 | 4,39 | 4,39 | 3,93 | 3,74 | 3,96 | 3,88 |
| 1806 | 2,17 | 1,81 | 2,11 | 2,03 | 2,77 | 1,99 | 2,38 | 2,38 | 2,13 | 1,97 | 1,88 | 1,99 |
| 1805 | 4,44 | 4,59 | 4,33 | 4,45 | 4,23 | 4,88 | 4,53 | 4,55 | 3,79 | 3,77 | 3,86 | 3,81 |
| 1804 | 1,47 | 1,69 | 1,59 | 1,58 | 1,48 | 2 | 1,74 | 1,74 | 1,62 | 1,81 | 1,63 | 1,69 |
| 1803 | 5,03 | 5,43 | 5,45 | 5,30 | 4,4 | 6,07 | 5,24 | 5,24 | 7,15 | 7,23 | 7,19 | 7,19 |
| 1802 | 4,88 | 4,68 | 5,27 | 4,94 | 3,38 | 4,79 | 4,09 | 4,09 | 3,65 | 3,79 | 3,59 | 3,68 |
| 1801 | 2,16 | 2,76 | 2,48 | 2,47 | 2,48 | 2,71 | 2,6 | 2,60 | 2,77 | 2,76 | 2,74 | 2,76 |
| 1800 | 2,52 | 2,48 | 2,83 | 2,61 | 3,07 | 3,02 | 3,05 | 3,05 | 3,14 | 3,14 | 3,08 | 3,12 |
| 1799 | 2,46 | 2,33 | 2,4 | 2,40 | 2,46 | 2,47 | 2,47 | 2,47 | 2,57 | 2,60 | 2,77 | 2,65 |
| 1798 | 3,24 | 2,89 | 3,35 | 3,16 | 3,38 | 2,78 | 3,08 | 3,08 | 3,73 | 3,81 | 3,71 | 3,75 |
| 1797 | 2,2 | 2,45 | 2,69 | 2,45 | 2,61 | 2,62 | 2,63 | 2,62 | 2,11 | 2,06 | 2,26 | 2,14 |
| 1796 | 2,16 | 2,27 | 2,21 | 2,21 | 2,31 | 2,54 | 2,43 | 2,43 | 2,51 | 2,42 | 2,32 | 2,41 |
| 1795 | 2,56 | 2,75 | 2,64 | 2,65 | 2,84 | 2,86 | 2,85 | 2,85 | 3,31 | 3,20 | 3,32 | 3,28 |
| 1794 | 2,69 | 2,75 | 3,19 | 2,88 | 2,87 | 2,96 | 2,93 | 2,92 | 2,83 | 2,63 | 2,56 | 2,67 |
| 1793 | 2,69 | 2,58 | 2,33 | 2,53 | 3,01 | 2,56 | 2,79 | 2,79 | 2,64 | 2,60 | 2,66 | 2,63 |
| 1792 | 3,92 | 4 | 4,15 | 4,02 | 4,17 | 4,08 | 4,13 | 4,13 | 3,96 | 3,89 | 3,93 | 3,93 |



| 1791 | 3,61 | 3,75 | 3,93 | 3,76 | 3,84 | 3,92 | 3,88 | 3,88 | 3,75 | 3,94 | 3,76 | 3,81 |
| 1790 | 2,87 | 2,5 | 2,45 | 2,61 | 2,92 | 2,56 | 2,74 | 2,74 | 2,22 | 2,41 | 2,50 | 2,38 |
| 1789 | 2,65 | 2,54 | 2,68 | 2,62 | 2,84 | 2,8 | 2,82 | 2,82 | 3,33 | 3,17 | 3,28 | 3,26 |
| 1788 | 2,83 | 2,39 | 2,91 | 2,71 | 2,69 | 2,66 | 2,68 | 2,68 | 1,70 | 1,87 | 1,86 | 1,81 |
| 1787 | 3,02 | 2,9 | 2,91 | 2,94 | 3,15 | 3,24 | 3,2 | 3,20 | 2,86 | 3,14 | 3,12 | 3,04 |
| 1786 | 5,07 | 4,86 | 5,3 | 5,08 | 3,61 | 3,81 | 3,71 | 3,71 | 3,97 | 4,05 | 4,06 | 4,02 |
| 1785 | 3,23 | 3,47 | 3,4 | 3,37 | 5,07 | 4,86 | 4,98 | 4,97 | 4,43 | 4,48 | 4,64 | 4,52 |
| 1784 | 4,15 | 4,4 | 4,47 | 4,34 | 3,23 | 3,47 | 3,35 | 3,35 | 3,76 | 3,81 | 3,54 | 3,70 |
| 1783 | 3,54 | 3,23 | 3,69 | 3,49 | 4,15 | 4,4 | 4,28 | 4,28 | 3,98 | 3,65 | 4,08 | 3,90 |
| 1782 | 3,54 | 3,23 | 3,69 | 3,49 | 3,54 | 3,23 | 3,37 | 3,38 | 3,60 | 3,85 | 3,71 | 3,72 |
| 1781 | 3,04 | 3,23 | 3,31 | 3,19 | 3,53 | 3,23 | 3,38 | 3,38 | 2,75 | 2,95 | 2,94 | 2,88 |
| 1780 | 2,56 | 2,19 | 2,95 | 2,57 | 2,68 | 2,86 | 2,77 | 2,77 | 2,92 | 3,13 | 2,80 | 2,95 |
| 1779 | 4,91 | 4,3 | 4,15 | 4,45 | 4,72 | 4,94 | 4,83 | 4,83 | 5,46 | 5,58 | 5,46 | 5,50 |
| 1778 | 3,47 | 3,91 | 3,8 | 3,73 | 4,15 | 4,37 | 4,26 | 4,26 | 3,18 | 3,25 | 3,22 | 3,22 |
| 1777 | 3,21 | 3,09 | 3,3 | 3,20 | 3,09 | 3,51 | 3,3 | 3,30 | 3,65 | 3,23 | 3,68 | 3,52 |
| 1776 | 4,51 | 3,7 | 4,42 | 4,21 | 4,47 | 4,48 | 4,45 | 4,47 | 3,28 | 3,60 | 3,39 | 3,43 |
| 1775 | 3,37 | 3,06 | 2,9 | 3,11 | 2,52 | 3,27 | 3,03 | 2,94 | 2,86 | 2,45 | 2,83 | 2,71 |
| 1774 | 2,32 | 2,18 | 2,4 | 2,30 | 2,24 | 2,57 | 2,24 | 2,35 | 2,13 | 2,18 | 2,20 | 2,17 |
| 1773 | 2,87 | 2,95 | 2,67 | 2,83 | 2,82 | 2,96 | 2,84 | 2,87 | 3,28 | 2,95 | 3,17 | 3,13 |
| 1772 | 3,32 | 3,33 | 2,85 | 3,17 | 2,96 | 3,26 | 3,48 | 3,23 | 2,94 | 3,46 | 3,52 | 3,31 |
| 1771 | 2,81 | 3,12 | 3,31 | 3,08 | 3,25 | 3,36 | 3,34 | 3,32 | 3,24 | 3,11 | 3,37 | 3,24 |
| 1770 | 3,69 | 3,51 | 3,21 | 3,47 | 3,53 | 3,56 | 3,48 | 3,52 | 3,55 | 3,38 | 3,36 | 3,43 |
| 1769 | 2,49 | 2,35 | 2,29 | 2,38 | 2,48 | 2,07 | 2,68 | 2,41 | 2,17 | 2,52 | 2,39 | 2,36 |
| 1768 | 3,18 | 3,11 | 3,16 | 3,15 | 2,96 | 3,27 | 3,38 | 3,20 | 3,66 | 3,58 | 3,66 | 3,63 |
| 1767 | 2,7 | 2,5 | 2,26 | 2,49 | 3,05 | 2,96 | 2,4 | 2,80 | 2,65 | 2,60 | 2,63 | 2,62 |
| 1766 | 2,66 | 2,32 | 2,31 | 2,43 | 2,3 | 2,47 | 2,64 | 2,47 | 2,11 | 2,37 | 2,29 | 2,25 |
| 1765 | 2,4 | 2,66 | 2,31 | 2,46 | 2,49 | 2,87 | 2,8 | 2,72 | 3,45 | 3,71 | 3,71 | 3,62 |
| 1764 | 4,08 | 4,13 | 4,36 | 4,19 | 4,58 | 4,44 | 4,52 | 4,51 | 3,97 | 3,84 | 3,96 | 3,92 |
| 1763 | 2,64 | 2,27 | 2,31 | 2,41 | 2,48 | 2,27 | 2,24 | 2,33 | 2,71 | 2,94 | 2,88 | 2,84 |
| 1762 | 3,63 | 3,74 | 3,64 | 3,67 | 3,53 | 3,75 | 3,59 | 3,62 | 3,87 | 3,61 | 3,67 | 3,72 |
| 1761 | 3,78 | 3,7 | 3,64 | 3,71 | 3,83 | 3,55 | 3,99 | 3,79 | 3,49 | 3,50 | 3,49 | 3,49 |
| 1760 | 3,01 | 3,23 | 3,39 | 3,21 | 3,43 | 3,55 | 3,38 | 3,45 | 3,55 | 3,34 | 3,64 | 3,51 |



| 1759 | 4,02 | 3,78 | 4,01 | 3,94 | 4,1 | 3,75 | 4,02 | 3,96 | 4,09 | 4,37 | 4,04 | 4,17 |
| 1758 | 2,46 | 2,61 | 2,54 | 2,54 | 2,86 | 2,57 | 2,48 | 2,64 | 2,33 | 2,38 | 2,58 | 2,43 |
| 1757 | 2,66 | 2,53 | 2,24 | 2,48 | 2,96 | 2,39 | 2,47 | 2,61 | 2,60 | 2,53 | 2,51 | 2,55 |
| 1756 | 2,63 | 2,51 | 2,74 | 2,63 | 2,76 | 3,06 | 2,87 | 2,90 | 3,05 | 3,42 | 3,11 | 3,19 |
| 1755 | 2,6 | 2,9 | 3,84 | 3,11 | 3,06 | 3,65 | 3,07 | 3,26 | 3,03 | 3,12 | 3,00 | 3,05 |
| 1754 | 4,98 | 4,89 | 3,71 | 4,53 | 4,87 | 4,64 | 4,75 | 4,75 | 3,86 | 3,85 | 3,82 | 3,85 |
| 1753 | 3,43 | 3,83 | 4,09 | 3,78 | 4,49 | 3,75 | 4,31 | 4,18 | 4,51 | 4,31 | 4,64 | 4,49 |
| 1752 | 3,63 | 3,69 | 3,59 | 3,64 | 3,53 | 3,95 | 3,57 | 3,68 | 3,28 | 3,13 | 3,25 | 3,22 |
| 1751 | 3,43 | 2,97 | 3,67 | 3,36 | 3,68 | 3,66 | 3,77 | 3,70 | 3,18 | 3,05 | 3,18 | 3,13 |
| 1750 | 3,25 | 3,48 | 2,56 | 3,10 | 4,1 | 3,66 | 3,66 | 3,81 | 3,27 | 3,63 | 3,47 | 3,46 |
| 1749 | 3,4 | 3,11 | 3,67 | 3,39 | 3,34 | 3,95 | 3,76 | 3,68 | 3,50 | 3,48 | 3,43 | 3,47 |
| 1748 | 3,6 | 4,53 | 3,9 | 4,01 | 4,96 | 4,25 | 4,26 | 4,49 | 3,70 | 3,62 | 3,74 | 3,69 |
| 1747 | 2,1 | 1,49 | 1,69 | 1,76 | 2,1 | 2,28 | 2,23 | 2,20 | 2,22 | 2,58 | 2,45 | 2,42 |
| 1746 | 2,83 | 2,57 | 3,03 | 2,81 | 2,85 | 3,03 | 2,92 | 2,93 | 2,43 | 2,28 | 2,53 | 2,42 |
| 1745 | 1,8 | 1,75 | 1,6 | 1,72 | 1,75 | 1,87 | 1,68 | 1,77 | 1,91 | 2,04 | 1,88 | 1,94 |
| 1744 | 2,58 | 2,31 | 2,61 | 2,50 | 2,59 | 2,56 | 2,64 | 2,60 | 2,55 | 2,51 | 2,43 | 2,50 |
| 1743 | 2,17 | 2,28 | 2,07 | 2,17 | 2,52 | 2,47 | 2,52 | 2,50 | 2,38 | 2,75 | 2,66 | 2,60 |
| 1742 | 3,16 | 2,64 | 2,75 | 2,85 | 2,72 | 3,25 | 3,19 | 3,05 | 3,12 | 3,25 | 3,08 | 3,15 |
| 1741 | 4,07 | 3,47 | 3,56 | 3,70 | 4,4 | 4,44 | 4,23 | 4,36 | 4,93 | 4,91 | 4,88 | 4,91 |
| 1740 | 2,6 | 2,69 | 2,35 | 2,55 | 2,52 | 2,66 | 2,55 | 2,58 | 2,17 | 2,15 | 2,25 | 2,19 |
| 1739 | 2,32 | 2,29 | 2,27 | 2,29 | 2,46 | 2,47 | 2,34 | 2,42 | 2,76 | 2,92 | 2,63 | 2,77 |
| 1738 | 2,8 | 2,68 | 3 | 2,83 | 2,98 | 2,86 | 2,96 | 2,93 | 2,91 | 3,16 | 2,90 | 2,99 |
| 1737 | 3,82 | 3,95 | 3,49 | 3,75 | 3,69 | 3,06 | 3,28 | 3,34 | 2,90 | 2,67 | 3,00 | 2,86 |
| 1736 | 2,41 | 2,23 | 2,35 | 2,33 | 2,39 | 2,37 | 2,15 | 2,30 | 2,75 | 2,89 | 2,86 | 2,83 |
| 1735 | 2,38 | 2,45 | 2,37 | 2,40 | 2,2 | 2,37 | 2,44 | 2,34 | 2,00 | 2,09 | 2,06 | 2,05 |
| 1734 | 1,87 | 2,01 | 2,05 | 1,98 | 2,27 | 2,37 | 2,12 | 2,25 | 2,59 | 2,70 | 2,56 | 2,62 |
| 1733 | 3,46 | 2,87 | 3,37 | 3,23 | 3,36 | 2,86 | 3,1 | 3,11 | 2,91 | 2,76 | 2,95 | 2,87 |
| 1732 | 1,45 | 1,73 | 1,51 | 1,56 | 1,49 | 1,78 | 1,63 | 1,63 | 1,70 | 1,62 | 1,84 | 1,72 |
| 1731 | 3,88 | 4,03 | 4,18 | 4,03 | 4,27 | 4,73 | 4,28 | 4,43 | 4,29 | 4,20 | 4,47 | 4,32 |
| 1730 | 2,17 | 1,94 | 2,23 | 2,11 | 2,33 | 1,78 | 2,04 | 2,05 | 2,13 | 1,94 | 2,08 | 2,05 |
| 1729 | 1,91 | 1,74 | 1,72 | 1,79 | 1,88 | 1,97 | 1,68 | 1,84 | 1,64 | 1,84 | 1,77 | 1,75 |
| 1728 | 2,13 | 1,76 | 1,92 | 1,94 | 1,95 | 2,56 | 2,09 | 2,20 | 2,65 | 2,63 | 2,53 | 2,60 |



| 1727 | 2,95 | 2,98 | 2,84 | 2,92 | 3,1 | 3,06 | 3,22 | 3,13 | 2,71 | 2,81 | 2,87 | 2,80 |
| 1726 | 5,6 | 5,16 | 5,35 | 5,37 | 5,74 | 5,42 | 5,35 | 5,50 | 5,56 | 5,32 | 5,48 | 5,45 |
| 1725 | 2,35 | 2,26 | 2,3 | 2,30 | 2,53 | 2,56 | 2,52 | 2,54 | 2,55 | 2,43 | 2,52 | 2,50 |
| 1724 | 3,08 | 3,15 | 2,67 | 2,97 | 2,98 | 3,25 | 3,1 | 3,11 | 3,18 | 3,31 | 3,47 | 3,32 |
| 1723 | 4,93 | 5,14 | 4,91 | 4,99 | 5,39 | 5,32 | 5,22 | 5,31 | 4,96 | 4,75 | 4,75 | 4,82 |
| 1722 | 3,19 | 2,85 | 3,19 | 3,08 | 3,09 | 2,66 | 3,04 | 2,93 | 2,34 | 2,42 | 2,59 | 2,45 |
| 1721 | 2,66 | 2,26 | 2,43 | 2,45 | 2,52 | 2,56 | 2,66 | 2,58 | 2,65 | 2,86 | 2,81 | 2,77 |
| 1720 | 2,69 | 3,16 | 2,88 | 2,91 | 2,53 | 2,86 | 2,96 | 2,78 | 3,61 | 3,84 | 3,65 | 3,70 |
| 1719 | 2,5 | 2,49 | 2,23 | 2,41 | 2,18 | 2,66 | 2,57 | 2,47 | 3,02 | 2,97 | 2,94 | 2,98 |
| 1718 | 3,96 | 3,99 | 2,55 | 3,50 | 3,7 | 3,78 | 3,73 | 3,74 | 3,81 | 3,92 | 3,94 | 3,89 |
| 1717 | 1,91 | 1,94 | 1,77 | 1,87 | 2,65 | 2,3 | 2,24 | 2,40 | 2,13 | 2,39 | 2,26 | 2,26 |
| 1716 | 1,9 | 3 | 2,5 | 2,47 | 1,92 | 2,37 | 2,42 | 2,24 | 2,54 | 2,23 | 2,59 | 2,45 |
| 1715 | 2,55 | 1,76 | 1,79 | 2,03 | 1,53 | 1,82 | 1,92 | 1,76 | 2,01 | 2,05 | 1,97 | 2,01 |
| 1714 | 4,23 | 3,67 | 3,93 | 3,94 | 3,93 | 4,2 | 4,39 | 4,17 | 3,72 | 3,54 | 3,95 | 3,73 |
| 1713 | 1,67 | 1,66 | 1,34 | 1,56 | 1,42 | 1,34 | 1,66 | 1,47 | 1,64 | 1,46 | 1,57 | 1,56 |
| 1712 | 1,75 | 2,28 | 2,55 | 2,19 | 1,89 | 2,28 | 2,18 | 2,12 | 2,17 | 1,98 | 2,23 | 2,13 |
| 1711 | 2,45 | 2,42 | 2,21 | 2,36 | 2,67 | 2,42 | 2,42 | 2,50 | 2,23 | 2,15 | 2,23 | 2,20 |

**Author contributions**

Marie-Eugenie Jamba and Pierre Francus conceptualized the project. Marie-Eugenie Jamba performed the formal data analysis. Pierre Francus acquired funding. Marie-Eugenie Jamba and Antoine Gagnon-Poiré performed the investigation. Marie-Eugenie Jamba developed the methodology. Pierre Francus provided supervision. Marie-Eugenie Jamba wrote the original draft. Marie-Eugenie Jamba, Pierre Francus and Guillaume St-Onge reviewed and edited the paper.

**Competing interests**

The authors declare that they have no conflict of interest.

**Disclaimer**

**Acknowledgements**

We would like to thank Mathieu Des Roches for the design of the sample holder. We are grateful to Philippe Letellier for the help provided during the scans of our samples on the micro-CT and for his continuous assistance. We would like to thank too

Margherita Martini for his advice and assistance in analyzing the micro-CT scans. We greatly thank Wanda and Dave Blake from North west River for their guiding experience and accommodation at Grand Lake. We thank the Labrador Institute at North West River for the use of their facility during fieldwork.





**Financial support**

The fieldwork campaigns have been supported by the POLAR Knowledge Canada through the Northern Scientific Training
Program grant to Antoine Gagnon-Poiré and Pierre Francus.

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
