# Peer review of "Measuring varve thickness using $\mu$ CT: a comparison with thin section"

_EGUsphere, 2024_

## Author Response (AR2)

**Author's response**

Dear reviewers and Associate editor,

Thank you very much for your valuable suggestions and comments on our manuscript. We appreciate the time and effort that you have dedicated to providing your valuable feedback on my manuscript. We will incorporate most of the suggestions you made for the final submission. Here is a point-by-point response to your comments and concerns.

**Reviewer 1**

Reviewer' Comments to the Authors:

Reviewer comment 1:

There are instances of, for lack of better expression, sloppy editing, repetitions, and overall lack of care. I had an impression that the manuscript's submission was rushed. See some of the specific comments.

Author response:  Thank you for your frankness. We will work to correct these errors for the final submission.

Reviewer comment 2:

The powerful μCT equipment used in the study is not readily available to numerous laboratories, and I'd appreciate some general perspective on this.

Author response: Micro-CT is starting to be used in many laboratories today. Also, it is well documented on the university's website and on the merchant's website. But indeed, we could improve this part of the article.

Reviewer comment 3:

Conclusions could be improved, at the moment it is a little too expected, consider improving with something really specific to your study. The use of CT is, in principle, well known.

Author response: We will provide an improved version during the final submission.

Reviewer comment 4:

A methodological paper like this would be extremely beneficial for the wider scientific community if, at some point, the Authors provided a simple chart or another way of following the "best practices." Introduce it, even if it is in supplementary materials. In other words, condense the methods section into a streamlined workflow with short explanations. This way, your paper will become a cornerstone for future works more quickly.

Author response: We think this is an excellent suggestion.

Reviewer comment 5:

A crucial part, that is not discussed, and I was actually expecting at least some indication of it: using the proposed approach one could more confidently measure the thickness of the varves – this is great. What about actual varve chronology development and transfer of these thicknesses onto the age-depth model? I understand that this is not an aim of this paper, but something that is a natural

follow up. Authors discuss missing varves/laminations on different planes. Combining counts from one plane is tedious task, but routinely done. What about now?

Author response: This is indeed not the purpose of our article, but I indeed it will have an impact on the chronology of varves by adding or removing one or more varves.

Reviewer comment 6:

consider adding a discussion paragraph, where you explain whether it would be possible and cost/time effective to scan the core before opening and then finding the best plane for splitting it. The power of µCT is its non-destructive nature, would all of us benefit from knowing how to split the core before deciding on thin sections? Does your setup allow the scanning of non-halves?

Author response: We agree with you, this is a section that could be included in the article.

Reviewer comment 7:

Generally, figures could use some more care and a unified approach. For example, multiple fonts are used, text is bolded and not, and size varies too much. Finally, there might be too many in the main body. Consider moving some into the appendix.

Author response: We agree with you. We will work to correct these errors for the final submission.

**Reviewer 2**

Reviewer' Comments to the Authors:

Reviewer comment 1: The writing is not always clear and precise and needs to be improved. There are many grammatical errors, incorrect expressions, and repetitions.

Author response: we acknowledge it was not internally reviewed by an English native speaker. That will be done for the final submission.

Reviewer comment 2: Not all figures are formatted correctly and not all are required in the body text. Consider moving some to the appendix.

Author response: We will work to improve the figures for the final submission and will consider to move some into the appendix.

Reviewer comment 3: Line 50-52: The limitations associated with the production of thin sections, i.e. sediment modification, should be explained and references added.

Author response: Indeed, we could provide more explanation.

Reviewer comment 4: In "2.1 Sample selection" information on the environmental setting is missing and the figure of the study area needs to be improved.

Author response: We will work to improve this part for the final submission.

Reviewer comment 5: In "2.3 Experimental setting: µCT acquisition and reconstruction": Why were the halfcores scanned in STAMINA mode? Helical scan trajectories (available in TESCAN CoreTOM) would avoid blurred parts in your samples.

Author response: Indeed, the helical mode is now accessible in TESCAN CoreTOM.

However, the half-core scans were done well before the helical mode was integrated into the µCT. Also, the results of the tests done on sediment cores and wood cores with the helical mode presented

several problems. First, the size and duration of the reconstruction were far too large, requiring much more powerful computers than those present in our laboratory. Also, even if the problem of the cone beam artifact was solved with the helical mode, there was the presence of new artifacts (which we could not explain) that degraded the quality of the images.

Reviewer comment 6: Line 139: Why is the shortest line segment separating two parallel varve borders measured, as these could potentially be disturbed? What is the reason for this?

Yes, indeed varves can be deformed as shown in Figure 16b-c, and these deformations can occur during the coring process. Mainly for poorly consolidated sediments, the layers are deformed at the edges as the core liner gradually sinks into the sediment. We could include some explanations in the article.

Author response:

Reviewer comment 7: As this is a methodological paper, I would recommend proposing a standardised workflow in the discussion section to simplify its application for the scientific community and to enable future comparisons between studies.

Author response: We agree with you, it's a great idea to propose a standardized workflow.

However, we think it would be better to integrate this workflow into the methodological part.

**Associate editor reviews**

Associate editor ' Comments to the Authors:

Comment 1: Comment by the Review of Fig 1. It could be more graphically improved, with all parts ABC adjusted within the one square, etc.

Maps b and c are a bit dark, too. I can ask for more tips from the Reviewer if you would like me to. Please let me know.

Author response: Thank you for your feedback. We agree with you. We will improve it.

Comment 2: Figure 7 has no units.

Author response:  We will correct it.

Comment 3: line 274: I would replace 'better' counting with 'more accurate' counting

Author response:  We will correct it.

Comment 4: Fig 13. the yellow and red lines are very thick, so it's impossible to see what the laminae looks like. Except for the one that should have been included in the thin section. Similar to Figures 14 and 15, it would be enough to have arrows pointing to the laminae.

Author response:  We will improve this image according to your indications.

**List of all relevant changes made in the manuscript**

1. The manuscript has been corrected by a native speaker.

2. More information about the µCT and the study site has been added in the methodology part.

3. The images have been improved and some have been sent to the appendices section.

4. Methodological workflows and additional explanations have been added for a better understanding of the work presented in the article.

5. An additional explanation to better understand the method of measuring thicknesses has been added.

6. Additional sections have been added to the discussion and conclusion to show the added value of this study.